# Theory-Inspired Path-Regularized Differential Network Architecture Search

**Pan Zhou**    **Caiming Xiong**    **Richard Socher**    **Steven C.H. Hoi**
Salesforce Research
{pzhou, cxiong, rsocher, shoi}@salesforce.com

## Abstract

Despite its high search efficiency, differential architecture search (DARTS) often selects network architectures with dominated skip connections which lead to performance degradation. However, theoretical understandings on this issue remain absent, hindering the development of more advanced methods in a principled way. In this work, we solve this problem by theoretically analyzing the effects of various types of operations, *e.g.* convolution, skip connection and zero operation, to the network optimization. We prove that the architectures with more skip connections can converge faster than the other candidates, and thus are selected by DARTS. This result, for the first time, theoretically and explicitly reveals the impact of skip connections to fast network optimization and its competitive advantage over other types of operations in DARTS. Then we propose a theory-inspired path-regularized DARTS that consists of two key modules: (i) a differential group-structured sparse binary gate introduced for each operation to avoid unfair competition among operations, and (ii) a path-depth-wise regularization used to incite search exploration for deep architectures that often converge slower than shallow ones as shown in our theory and are not well explored during search. Experimental results on image classification tasks validate its advantages.

## 1 Introduction

Network architecture search (NAS) [1] is an effective approach for automating network architecture design, with many successful applications witnessed to image recognition [2–6] and language modeling [1, 6]. The methodology of NAS is to automatically search for a directed graph and its edges from a huge search space. Unlike expert-designed architectures which require substantial efforts from experts by trial and error, the automatic principle in NAS greatly alleviates these design efforts and possible design bias brought by experts which could prohibit achieving better performance. Thanks to these advantages, NAS has been widely devised via reinforcement learning (RL) and evolutionary algorithm (EA), and achieved promising results in many applications, *e.g.* classification [2, 4].

DARTS [6] is a recently developed leading approach. Different from RL and EA based methods [1–4] that discretely optimize architecture parameters, DARTS converts the operation selection for each edge in the directed graph into continuously weighting a fixed set of operations. In this way, it can optimize the architecture parameters via gradient descent and greatly reduces the high search cost in RL and EA approaches. However, as observed in the literatures [7–10] and Fig. 1 (a), this differential NAS family, including DARTS and its variants [11, 12], typically selects many skip connections which dominate over other types of operations in the network graph. Consequently, the searched networks are observed to have unsatisfactory performance. To alleviate this issue, some empirical techniques are developed, *e.g.* operation-level dropout [7], fair operation-competing loss [8]. But no attention has been paid to developing theoretical understandings for why skip connections dominate other types of operations in DARTS. The theoretical answer to this question is important not only for better understanding DARTS, but also for inspiring new insights for DARTS algorithm improvement.

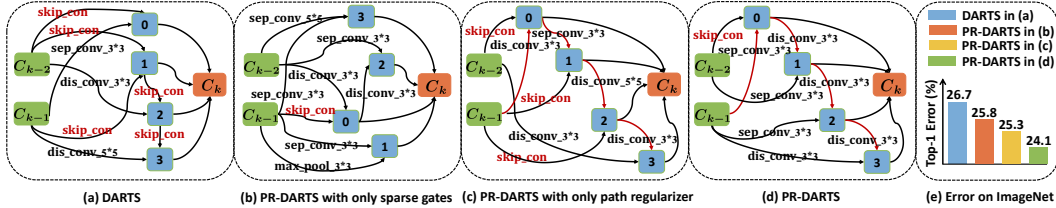

Figure 1: Illustration of selected normal cells by DARTS and PR-DARTS. By comparison, the group-structured sparse gates in PR-DARTS (b) well alleviate unfair operation competition and overcome the dominated-skip-connection issue in DARTS (a); path-depth-wise regularization in PR-DARTS (c) helps rectify cell-selection-bias to shallow cells; PR-DARTS (d) combines these two complementary components and well alleviates the above two issues, testified by the results in (e).

**Contributions.** In this work, we address the above fundamental question and contribute to derive some new results, insights and alternatives for DARTS. Particularly, we provide rigorous theoretical analysis for the dominated skip connections in DARTS. Inspired by our theory, we then propose a new alternative of DARTS which can search networks without dominated skip connections and achieves state-of-the-art classification performance. Our main contributions are highlighted below.

Our first contribution is proving that DARTS prefers to skip connection more than other types of operations, *e.g.* convolution and zero operation, in the search phase, and tends to search favor skip-connection-dominated networks as shown in Fig. 1 (a). Formally, in the search phase, DARTS first fixes architecture parameter $\boldsymbol{\beta}$ which determines the operation weights in the graph to optimize the network parameter $\boldsymbol{W}$ by minimizing training loss $F_{\text{train}}(\boldsymbol{W}, \boldsymbol{\beta})$ via gradient descent, and then uses the validation loss $F_{\text{val}}(\boldsymbol{W}, \boldsymbol{\beta})$ to optimize $\boldsymbol{\beta}$ via gradient descent. We prove that when optimizing $F_{\text{train}}(\boldsymbol{W}, \boldsymbol{\beta})$, the convergence rate at each iteration depends on the weights of skip connections much heavier than other types of operations, *e.g.* convolution, meaning that the more skip connections the faster convergence. Since training and validation data come from the same distribution which means $\mathbb{E}[F_{\text{train}}(\boldsymbol{W}, \boldsymbol{\beta})] = \mathbb{E}[F_{\text{val}}(\boldsymbol{W}, \boldsymbol{\beta})]$, more skip connections can also faster decay $F_{\text{val}}(\boldsymbol{W}, \boldsymbol{\beta})$ in expectation. So when updating architecture parameter $\boldsymbol{\beta}$, DARTS will tune the weights of skip connections larger to faster decay validation loss, and meanwhile, will tune the weights of other operations smaller since all types of operations on one edge share a softmax distribution. Accordingly, skip connections gradually dominate the network graph. To our best knowledge, this is the first theoretical result that explicitly shows heavier dependence of the convergence rate of NAS algorithm on skip connections, explaining the dominated skip connections in DARTS due to their optimization advantages.

Inspired by our theory, we further develop the path-regularized DARTS (PR-DARTS) as a novel alternative to alleviate unfair competition between skip connection and other types of operations in DARTS. To this end, we define a group-structured sparse binary gate implemented by Bernoulli distribution for each operation. These gates independently determine whether their corresponding operations are used in the graph. Then we divide all operations in the graph into skip connection group and non-skip connection group, and independently regularize the gates in these two groups to be sparse via a hard threshold function. This group-structured sparsity penalizes the skip connection group heavier than another group to rectify the competitive advantage of skip connections over other operations as shown in Fig. 1 (b), and globally and gradually prunes unnecessary connections in the search phase to reduce the pruning information loss after searching. More importantly, we introduce a path-depth-wise regularization which encourages large activation probability of gates along the long paths in the network graph and thus incites more search exploration to deep graphs illustrated by Fig. 1 (c). As our theory shows that gradient descent can faster optimize shallow and wide networks than deep and thin ones, this path-depth-wise regularization can rectify the competitive advantage of shallow network over deep one. So PR-DARTS can search performance-oriented networks instead of fast-convergence-oriented networks and achieves better performance testified by Fig. 1 (e).

## 2 Related Work

DARTS [6] has gained much attention recently thanks to its high search efficiency [7–17]. It relaxes a discrete search space to a continuous one via continuously weighting the operations, and then employs gradient descent algorithm to select promising candidates. In this way, it significantly improves

the search efficiency over RL and EA based NAS approaches [1–4]. But the selected networks by DARTS have dominated skip connections which lead to unsatisfactory performance [7–10]. To solve this issue, Chen *et al.* [7] introduced operation-level dropout [18] to regularize skip connection. Chu *et al.* [8] used independent sigmoid function for weighting each operation to avoid operation competition, and designed a new loss to independently push the operation weights to zero or one. In contrast, our PR-DARTS uses binary gate for each operation and then imposes group-structured and path-depth-wise regularizations to alleviate the fast-convergence-oriented search issue in DARTS.

The intrinsic theoretical reasons for the dominated skip connection in DARTS are rarely investigated though heavily desired. Zela *et al.* [9] empirically analyzed the poor generalization performance of the selected architectures by DARTS from the argument of sharp and flat minima. Shu *et al.* [19] studied general NAS and showed that NAS prefers to shallow and wide networks since these networks have more smooth landscape empirically and smaller gradient variance which both boost training speed. But they did not reveal any relation between skip connections and convergence behaviors. Differently, we explicitly show the role of weights of different operations in determining the convergence rate in network optimization, revealing the intrinsic reasons for the dominated skip connections in DARTS.

## 3 Theoretical Analysis for DARTS

In this section, we first recall the formulation of DARTS, and then theoretically analyze the intrinsic reasons for the dominated skip connections in DARTS by analyzing its convergence behaviors.

### 3.1 Formulation of DARTS

DARTS [6] searches cells which are used to stack the full network architecture. A cell is organized as a directed acyclic graph with $h$ nodes $\{X^{(l)}\}_{l=0}^{h-1}$. Typically, the graph contains two input nodes $X^{(0)}$ and $X^{(1)}$ respectively defined as the outputs of two previous cells, and has one output node $X^{(h-1)}$ giving by concatenating all intermediate nodes $X^{(l)}$. Each intermediate node $X^{(l)}$ connects with all previous nodes $X^{(s)}$ $(0 \le s < l)$ via a continuous operation-weighting strategy, namely

$$X^{(l)} = \sum_{0 \le s < l} \sum_{t=1}^{r} \alpha_{s,t}^{(l)} O_t(X^{(s)}) \quad \text{with} \quad \alpha_{s,t}^{(l)} = \exp(\beta_{s,t}^{(l)}) / \sum_{t=1}^{r} \exp(\beta_{s,t}^{(l)}), \quad (1)$$

where the operation $O_t$ comes from the operation set $\mathcal{O} = \{O_t\}_{t=1}^{r}$, including zero operation, skip connection, convolution, etc. In this way, the architecture search problem becomes efficiently learning continuous architecture parameter $\boldsymbol{\beta} = \{\beta_{s,t}^{(l)}\}_{l,s,t}$ via optimizing the following bi-level model

$$\min_{\boldsymbol{\beta}} \ F_{\text{val}}(W^*(\boldsymbol{\beta}), \boldsymbol{\beta}), \qquad \text{s.t.} \ W^*(\boldsymbol{\beta}) = \text{argmin}_{W} \ F_{\text{train}}(W, \boldsymbol{\beta}), \quad (2)$$

where $F_{\text{train}}$ and $F_{\text{val}}$ respectively denote the loss on the training and validation datasets, $W$ is the network parameters in the graph, *e.g.* convolution parameters. Then DARTS optimizes the architecture parameter $\boldsymbol{\beta}$ and the network parameter $W$ by alternating gradient descent. After learning $\boldsymbol{\beta}$, DARTS prunes the dense graph according to the weight $\alpha_{s,t}^{(l)}$ in Eqn. (1) to obtain compact cells.

Despite its much higher search efficiency over RL and EA based methods, DARTS typically selects a cell with dominated skip connections, leading to unsatisfactory performance [7–10]. But there is no rigorously theoretical analysis that explicitly justifies why DARTS tends to favor skip connections. The following section attempts to solve this issue by analyzing the convergence behaviors of DARTS.

### 3.2 Analysis Results for DARTS

For analysis, we detail the cell structures in DARTS. Let input be $X \in \mathbb{R}^{\bar{m} \times \bar{p}}$ where $\bar{m}$ and $\bar{p}$ are respectively the channel number and dimension of input. Typically, one needs to resize the input to a target size $m \times p$ via a convolution layer with parameter $W^{(0)} \in \mathbb{R}^{m \times k_c \bar{m}}$ (kernel size $k_c \times k_c$)

$$X^{(0)} = \text{conv}(W^{(0)}, X) \in \mathbb{R}^{m \times p} \quad \text{with} \quad \text{conv}(W; X) = \tau \sigma(W \Phi(X)), \quad (3)$$

and then feed it into the subsequent layers. The convolution operation conv performs convolution and then nonlinear mapping via activation function $\sigma$. The scaling factor $\tau$ equals to $\frac{1}{\sqrt{\bar{m}}}$ when channel number in conv is $\bar{m}$. It is introduced to simplify the notations in our analysis and does not affect

convergence behaviors of DARTS. For notation simplicity, we assume stride $s_c = 1$ and padding zero $p_c = \frac{k_c-1}{2}$ to make the same sizes of output and input. Given a matrix $\boldsymbol{Z} \in \mathbb{R}^{m \times p}$, $\Phi(\boldsymbol{Z})$ is defined as

$$\Phi(\boldsymbol{Z}) = \begin{bmatrix} \boldsymbol{Z}_{1,-p_c+1:p_c+1}^\top & \boldsymbol{Z}_{1,-p_c+2:p_c+2}^\top & \cdots & \boldsymbol{Z}_{1,p-p_c:p+p_c}^\top \\ \boldsymbol{Z}_{2,-p_c+1:p_c+1}^\top & \boldsymbol{Z}_{2,-p_c+2:p_c+2}^\top & \cdots & \boldsymbol{Z}_{2,p-p_c:p+p_c}^\top \\ \vdots & \vdots & \ddots & \vdots \\ \boldsymbol{Z}_{m,-p_c+1:p_c+1}^\top & \boldsymbol{Z}_{m,-p_c+2:p_c+2}^\top & \cdots & \boldsymbol{Z}_{m,p-p_c:p+p_c}^\top \end{bmatrix} \in \mathbb{R}^{k_c m \times p},$$

where $\boldsymbol{Z}_{i,t} = 0$ ($t \leq 0$ or $t > p$). Then the conventional convolution can be computed as $\boldsymbol{W}\Phi(\boldsymbol{X})$ where each row in $\boldsymbol{W}$ denotes a conventional kernel. Note, for other convolutions, e.g. depth-wise separable convolution, our analysis framework still holds and can derive very similar results. Now we are ready to define the subsequent layers in the cell:

$$\boldsymbol{X}^{(l)} = \sum_{s=0}^{l-1} \left( \boldsymbol{\alpha}_{s,1}^{(l)} \mathsf{zero}(\boldsymbol{X}) + \boldsymbol{\alpha}_{s,2}^{(l)} \mathsf{skip}(\boldsymbol{X}) + \boldsymbol{\alpha}_{s,3}^{(l)} \mathsf{conv}(\boldsymbol{W}_s^{(l)}; \boldsymbol{X}^{(s)}) \right) \in \mathbb{R}^{m \times p} \ (l=1,\cdots,h-1),$$
(4)

where zero operation $\mathsf{zero}(\boldsymbol{X}) = \boldsymbol{0}$ and skip connection $\mathsf{skip}(\boldsymbol{X}) = \boldsymbol{X}$, $\boldsymbol{\alpha}_{s,t}^{(l)}$ is given in (1). In this work, we consider three representative operations, *i.e.* zero, skip connection and convolution, and ignore pooling operation since it reveals the same behaviors as convolution, namely both being dominated by skip connections [7–9]. Next, we feed concatenation of all intermediate nodes into a linear layer to obtain the prediction $u_i$ of the $i$-th sample $\boldsymbol{X}_i$ and then obtain a mean squared loss:

$$F(\boldsymbol{W}, \boldsymbol{\beta}) = \frac{1}{2n} \sum_{i=1}^{n} (u_i - y_i)^2 \quad \text{with} \quad u_i = \sum_{s=0}^{h-1} \langle \boldsymbol{W}_s, \boldsymbol{X}_i^{(s)} \rangle \in \mathbb{R},$$
(5)

where $\boldsymbol{X}_i^{(s)}$ denotes the $s$-th feature node for sample $\boldsymbol{X}_i$, $\{\boldsymbol{W}_s\}_{s=0}^{h-1}$ denote the parameters for the linear layer. $F(\boldsymbol{W}, \boldsymbol{\beta})$ becomes $F_{\text{train}}(\boldsymbol{W}, \boldsymbol{\beta})$ ($F_{\text{val}}(\boldsymbol{W}, \boldsymbol{\beta})$) when samples come from training dataset (validation dataset). Subsequently, we analyze the effects of various types of operations to the convergence behaviors of $F_{\text{train}}(\boldsymbol{W}, \boldsymbol{\beta})$ when optimize the network parameter $\boldsymbol{W}$ via gradient descent:

$$\boldsymbol{W}_s^{(l)}(k{+}1) = \boldsymbol{W}_s^{(l)}(k) - \eta \nabla_{\boldsymbol{W}_s^{(l)}(k)} F_{\text{train}}(\boldsymbol{W}, \boldsymbol{\beta}) \ (\forall l, s), \ \ \boldsymbol{W}_s(k{+}1) = \boldsymbol{W}_s(k) - \eta \nabla_{\boldsymbol{W}_s(k)} F_{\text{train}}(\boldsymbol{W}, \boldsymbol{\beta}) \ (\forall s),$$
(6)

where $\eta$ is the learning rate. We use gradient descent instead of stochastic gradient descent, since gradient descent is expectation version of stochastic one and can reveal similar convergence behaviors. For analysis, we first introduce mild assumptions widely used in stochastic optimization [20–23] and network analysis [24–31].

**Assumption 1.** *Assume the activation function $\sigma$ is $\mu$-Lipschitz and $\rho$-smooth. That is, for $\forall x_1, x_2$, $\sigma$ satisfies $|\sigma(x_1) - \sigma(x_2)| \leq \mu|x_1 - x_2|$ and $|\sigma'(x_1) - \sigma'(x_2)| \leq \rho|x_1 - x_2|$. Moreover, we assume that $\sigma(0)$ can be upper bounded, and $\sigma$ is analytic and is not a polynomial function.*

**Assumption 2.** *Assume the initialization of the convolution parameters ($\boldsymbol{W}_s^{(l)}$) and the linear mapping parameters ($\boldsymbol{W}_s$) are drawn from Gaussian distribution $\mathcal{N}(\boldsymbol{0}, \boldsymbol{I})$.*

**Assumption 3.** *Suppose the samples $\{\boldsymbol{X}_i\}_{i=1}^n$ are normalized such that $\|\boldsymbol{X}_i\|_F = 1$. Moreover, they are not parallel, namely $\mathsf{vec}(\boldsymbol{X}_i) \notin \mathrm{span}(\mathsf{vec}(\boldsymbol{X}_j))$ for all $i \neq j$, where $\mathsf{vec}(\boldsymbol{X}_i)$ vectorizes $\boldsymbol{X}_i$.*

Assumption 1 is mild, since most differential activation functions, *e.g.* softplus and sigmoid, satisfy it. The Gaussian assumption on initial parameters in Assumption 2 is used in practice. We assume Gaussian variance to be one for notation simplicity in analysis, but our technique is applicable to any constant variance. The normalization and non-parallel conditions in Assumption 3 are satisfied in practice, as normalization is a data preprocess and samples in a dataset are often not restrictively parallel. Based on assumptions, we summarize our result in Theorem 1 with proof in Appendix C.1.

**Theorem 1.** *Suppose Assumptions 1, 2 and 3 hold. Let $c = \left(1 + \boldsymbol{\alpha}_2 + 2\boldsymbol{\alpha}_3 \mu \sqrt{k_c} c_{w0}\right)^h$, $\boldsymbol{\alpha}_2 = \max_{s,l} \boldsymbol{\alpha}_{s,2}^{(l)}$ and $\boldsymbol{\alpha}_3 = \max_{s,l} \boldsymbol{\alpha}_{s,3}^{(l)}$. If $m \geq \frac{c_m \mu^2}{\lambda^2} \left[\rho p^2 n^2 \log(n/\delta) + c^2 k_c^2 c_{w0}^2/n\right]$ and $\eta \leq \frac{c_\eta \lambda}{\sqrt{m} \mu^4 h^3 k_c^2 c^4}$, where $c_{w0}$, $c_m$, $c_\eta$ are constants, $\lambda$ is given below. Then when fixing architecture parameterize $\boldsymbol{\alpha}$ in (1) and optimizing network parameter $\boldsymbol{W}$ via gradient descent (6), with probability at least $1 - \delta$ we have*

$$F_{\text{train}}(\boldsymbol{W}(k+1), \boldsymbol{\beta}) \leq (1 - \eta\lambda/4) \, F_{\text{train}}(\boldsymbol{W}(k), \boldsymbol{\beta}) \quad (\forall k \geq 1),$$

*where $\lambda = \frac{3c_\sigma}{4} \lambda_{\min}(\boldsymbol{K}) \sum_{s=0}^{h-2} (\boldsymbol{\alpha}_{s,3}^{(h-1)})^2 \prod_{t=0}^{s-1} (\boldsymbol{\alpha}_{t,2}^{(s)})^2$, the positive constant $c_\sigma$ only depends on $\sigma$ and input data, $\lambda_{\min}(\boldsymbol{K}) = \min_{i,j} \lambda_{\min}(\boldsymbol{K}_{ij})$ is larger than zero in which $\lambda_{\min}(\boldsymbol{K}_{ij})$ is the smallest eigenvalue of $\boldsymbol{K}_{ij} = \left[\boldsymbol{X}_i^\top \boldsymbol{X}_j, \boldsymbol{X}_i^\top \boldsymbol{X}_j; \boldsymbol{X}_j^\top \boldsymbol{X}_i, \boldsymbol{X}_j^\top \boldsymbol{X}_j\right]$.*

Theorem 1 shows that for an architecture-fixed over-parameterized network, when using gradient descent to optimize the network parameter $\boldsymbol{W}$, one can expect the convergence of the algorithm which is consistent with prior deep learning optimization work [24–27]. More importantly, the convergence rate at each iteration depends on the network architectures which is parameterized by $\boldsymbol{\alpha}$.

Specifically, for each factor $\lambda_s = (\boldsymbol{\alpha}_{s,3}^{(h-1)})^2 \prod_{t=0}^{s-1} (\boldsymbol{\alpha}_{t,2}^{(s)})^2$ in the factor $\lambda$, it is induced by the connection path $\boldsymbol{X}^{(0)} \to \boldsymbol{X}^{(1)} \to \cdots \to \boldsymbol{X}^{(s)} \to \boldsymbol{X}^{(h-1)}$. By observing $\lambda_s$, one can find that (1) for the connections before node $\boldsymbol{X}^{(s)}$, it depends on the weights $\boldsymbol{\alpha}_{t,2}^{(s)}$ of skip connections heavier than convolution and zero operation, and (2) for the direct connection between $\boldsymbol{X}^{(s)}$ and $\boldsymbol{X}^{(h-1)}$, it relies on convolution weight $\boldsymbol{\alpha}_{s,3}^{(h)}$ heavier than the weights of other type operations. For observation (1), it can be intuitively understood: as shown in [32–35], skip connection often provides larger gradient flow than the parallel convolution and zero connection and thus greatly benefits faster convergence of networks, since skip connection maintains primary information flow, while convolution only learns the residual information and zero operation does not delivery any information. So convolution and zero operations have negligible contribution to information flow and thus their weights do not occur in $\prod_{t=0}^{s-1} (\boldsymbol{\alpha}_{t,2}^{(s)})^2$ of $\lambda_s$. For observation (2), as the path $\boldsymbol{X}^{(0)} \to \boldsymbol{X}^{(1)} \to \cdots \to \boldsymbol{X}^{(s)}$ is shared for all subsequent layers, it prefers skip connection more to maintain information flow, while for the private connection between $\boldsymbol{X}^{(s)}$ and $\boldsymbol{X}^{(h-1)}$ which is not shared since $\boldsymbol{X}^{(h-1)}$ is the last node, it relies on learnable convolution more heavily than non-parameterized operations, since learnable operations have parameter to learn and can reduce the loss. For the theoretical reasons for observations (1) and (2), the skip connection in the shared path can improve the singularity of network Gram matrix more than other types of operations, where the singularity directly determines the convergence rate, while the learnable convolution in private path can benefit the Gram matrix singularity much more. See details in Appendix C.3. The weight $\boldsymbol{\alpha}_{s,3}^{(l)}$ of zero operation does not occur in $\lambda$, as it does not delivery any information.

Now we analyze why the selected cell has dominated skip connections. The above analysis shows that the convergence rate when optimizing $F_{\text{train}}(\boldsymbol{W}, \boldsymbol{\beta})$ depends on the weights of skip connections heavier than other weights in the shared connection path which dominates the connections of a cell. So larger weights of skip connections often give faster loss decay of $F_{\text{train}}(\boldsymbol{W}, \boldsymbol{\beta})$. Consider the samples for training and validation come from the same distribution which means $\mathbb{E}[F_{\text{train}}(\boldsymbol{W}, \boldsymbol{\beta})] = \mathbb{E}[F_{\text{val}}(\boldsymbol{W}, \boldsymbol{\beta})]$, larger weights of skip connections can also faster reduce $F_{\text{val}}(\boldsymbol{W})$ in expectation, which accords with the empirical observations in Fig. 2 and the observations in [9]. In Fig. 2, we first set all operations in NAS cell (normal and reduction cells, see details in Sec. 5) as convolution

$(3 \times 3)$, and randomly select $0\%$, $37.5\%$ and $62.5\%$ operations as skip connections. Next, we stack 8 NAS cells to build a network and train on CIFAR10 with same settings. Fig. 2 shows that more skip connections gives faster convergence. So when optimizing $\boldsymbol{\alpha}$ via optimizing $\boldsymbol{\beta}$ in $F_{\text{val}}(\boldsymbol{W}, \boldsymbol{\beta})$, DARTS will tune weights of most skip connections larger to faster reduce $F_{\text{val}}(\boldsymbol{W}, \boldsymbol{\beta})$. As the weights of three operations on one edge share a softmax distribution in (1), increasing one operation weight means reducing other operation weights. Thus, skip connections gradually dominate over other types of operations for most connections in the cell. So when pruning operations according to their weights, most of skip connections are preserved while most of other operations are pruned. This explains the dominated skip connections in the cell selected by DARTS.

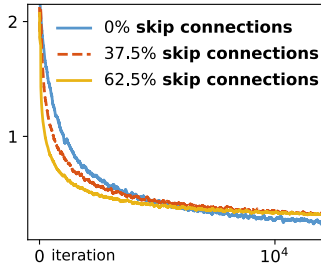

Figure 2: Effects of skip connections to convergence rate of network.

## 4 Path-Regularized Differential Network Architecture Search

The proposed method consists of two main components, *i.e.* group-structured sparse stochastic gate for each operation and path-depth-wise regularization on gates, which are introduced below in turn.

### 4.1 Group-structured Sparse Operation Gates

The analysis in Sec. 3.2 shows that skip connection has superior competing advantages over other types of operations when they share one softmax distribution. To resolve this issue, we introduce independent stochastic gate for each operation between two nodes to avoid the direct competition

between skip connection and other operations. Specifically, we define a stochastic binary gate $\boldsymbol{g}_{s,t}^{(l)}$ for the $t$-th operation between nodes $\boldsymbol{X}^{(s)}$ and $\boldsymbol{X}^{(l)}$, where $\boldsymbol{g}_{s,t}^{(l)} \sim \text{Bernoulli}\big(\exp(\boldsymbol{\beta}_{s,t}^{(l)})/(1+\exp(\boldsymbol{\beta}_{s,t}^{(l)}))\big)$. Then at each iteration, we sample gate $\boldsymbol{g}_{s,t}^{(l)}$ from its Bernoulli distribution and compute each node as

$$\boldsymbol{X}^{(l)} = \sum_{1 \le i < l} \sum_{t=1}^{r} \boldsymbol{g}_{s,t}^{(l)} O_t\big(\boldsymbol{X}^{(i)}\big). \tag{7}$$

Since the discrete sampling of $\boldsymbol{g}_{s,t}^{(l)}$ is not differentiable, we use Gumbel technique [36, 37] to approximate $\boldsymbol{g}_{s,t}^{(l)}$ as $\bar{\boldsymbol{g}}_{s,t}^{(l)} = \Theta\big((\ln \delta - \ln(1 - \delta) + \boldsymbol{\beta}_{s,t}^{(l)})/\tau\big)$ where $\Theta$ denotes sigmoid function, $\delta \sim \text{Uniform}(0,1)$. For temperature $\tau$, when $\tau \to 0$ the approximated distribution $\bar{\boldsymbol{g}}_{s,t}^{(l)}$ recovers Bernoulli distribution and is non-smooth, while when $\tau \to +\infty$, the approximated distribution becomes very smooth. In this way, the gradient can be back-propagated through $\bar{\boldsymbol{g}}_{s,t}^{(l)}$ to the network parameter $\boldsymbol{W}$.

If there is no regularization on the independent gates, then there are two issues. The first one is that the selected cells would have large weights for most operations. This is because (1) as shown in Theorem 1, increasing operation weights can lead to faster convergence rate; (2) increasing weights of any operations can strictly reduce or maintain the loss which is formally stated in Theorem 2. Let $t_{\text{skip}}$ and $t_{\text{conv}}$ respectively be the indexes of skip connection and convolution in the operation set $\mathcal{O}$.

**Theorem 2.** *Assume the weights in DARTS model (2) is replaced with the independent gates $\boldsymbol{g}_{s,t}^{(l)}$.*
*(1) Increasing the value of $\boldsymbol{g}_{s,t}^{(l)}$ of the operations, including zero operation, skip connection, pooling, and convolution with any kernel size, can reduce or maintain the loss $F_{val}(\boldsymbol{W}^*(\boldsymbol{\beta}), \boldsymbol{\beta})$ in (2).*
*(2) Suppose the assumptions in Theorem 1 hold. With probability at least $1 - \delta$, increasing $\boldsymbol{g}_{s,t_{skip}}^{(l)}$ ($0 \le s < l < h - 1$) of skip connection or $\boldsymbol{g}_{s,t_{conv}}^{(h-1)}$ ($0 \le s < h - 1$) of convolution with increment $\epsilon$ can reduce the loss $F_{val}(\boldsymbol{W}^*(\boldsymbol{\beta}), \boldsymbol{\beta})$ in (2) to $F_{val}(\boldsymbol{W}^*(\boldsymbol{\beta}), \boldsymbol{\beta}) - C\epsilon$ in expectation, where $C$ is a positive constant.*

See its proof in Appendix D.1. Theorem 2 shows that DARTS with independent gates would tune the weights of most operations large to obtain faster convergence and smaller loss, leading to dense cells and thus performance degradation when pruning these large weights. The second issue is that independent gates cannot encourage benign competition and cooperation among operations, as Theorem 2 shows most operations tend to increase their weights. Considering the performance degradation caused by pruning dense cells, benign competition and cooperation among operations are necessary for gradually pruning unnecessary operations to obtain relatively sparse selected cells.

To resolve these two issues, we impose group-structured sparsity regularization on the stochastic gates. Following [38] we stretch $\bar{\boldsymbol{g}}_{s,t}^{(l)}$ from the range $[0, 1]$ to $[a, b]$ via rescaling $\tilde{\boldsymbol{g}}_{s,t}^{(l)} = a + (b - a)\bar{\boldsymbol{g}}_{s,t}^{(l)}$, where $a < 0$ and $b > 1$ are two constants. Then we feed $\tilde{\boldsymbol{g}}_{s,t}^{(l)}$ into a hard threshold gate to obtain the gate $\boldsymbol{g}_{s,t}^{(l)} = \min(1, \max(0, \tilde{\boldsymbol{g}}_{s,t}^{(l)}))$. In this way, the gate $\boldsymbol{g}_{s,t}^{(l)}$ enjoys good properties, *e.g.* exact zero values and computable activation probability ($\mathbb{P}(\boldsymbol{g}_{s,t}^{(l)} \ne 0)$, which are formally stated in Theorem 3.

**Theorem 3.** *For each stochastic gate $\boldsymbol{g}_{s,t}^{(l)}$, it satisfies $\boldsymbol{g}_{s,t}^{(l)} = 0$ when $\tilde{\boldsymbol{g}}_{s,t}^{(l)} \in (0, -\frac{a}{b-a}]$; $\boldsymbol{g}_{s,t}^{(l)} = \tilde{\boldsymbol{g}}_{s,t}^{(l)}$ when $\tilde{\boldsymbol{g}}_{s,t}^{(l)} \in (-\frac{a}{b-a}, \frac{1-a}{b-a}]$; $\boldsymbol{g}_{s,t}^{(l)} = 1$ when $\tilde{\boldsymbol{g}}_{s,t}^{(l)} \in (\frac{1-a}{b-a}, 1]$. Moreover, $\mathbb{P}(\boldsymbol{g}_{s,t}^{(l)} \ne 0) = \Theta(\boldsymbol{\beta}_{s,t}^{(l)} - \tau \ln \frac{-a}{b})$.*

See its proof in Appendix D.2. Theorem 3 shows that the gate $\boldsymbol{g}_{s,t}^{(l)}$ can achieve exact zero, which can reduce information loss caused by pruning at the end of search. Next based on the activation probability $\mathbb{P}(\boldsymbol{g}_{s,t}^{(l)} \ne 0)$ in Theorem 3, we design group-structured sparsity regularizations. We collect all skip connections in the cell as a skip-connection group and take the remaining operations into non-skip-connection group. Then we compute the average activation probability of these two groups:

$$\mathcal{L}_{\text{skip}}(\boldsymbol{\beta}) = \zeta \sum_{l=1}^{h-1} \sum_{s=0}^{l-1} \Theta\Big(\boldsymbol{\beta}_{s,t_{\text{skip}}}^{(l)} - \tau \ln \frac{-a}{b}\Big), \quad \mathcal{L}_{\text{non-skip}}(\boldsymbol{\beta}) = \frac{\zeta}{r-1} \sum_{l=1}^{h-1} \sum_{s=0}^{l-1} \sum_{1 \le t \le r, t \ne t_{\text{skip}}} \Theta\Big(\boldsymbol{\beta}_{s,t}^{(l)} - \tau \ln \frac{-a}{b}\Big),$$

where $\zeta = \frac{2}{h(h-1)}$. Then we respectively regularize $\mathcal{L}_{\text{skip}}$ and $\mathcal{L}_{\text{non-skip}}$ by two different regularization constants $\lambda_1$ and $\lambda_2$ ($\lambda_1 > \lambda_2$ in experiments). This group-structured sparsity has three benefits:

(1) penalizing skip connections heavier than other types of operations can rectify the competitive advantage of skip connections over other operations and avoids skip-connection-dominated cell; (2) sparsity regularizer gradually and automatically prunes redundancy and unnecessary connections which reduces the information loss of pruning at the end of search; (3) sparsity regularizer defined on the whole cell can encourage global competition and cooperation of all operations in the cell, which differs from DARTS that only introduces local competition among the operations between two nodes.

## 4.2 Path-depth-wise Regularizer on Operation Gates

Except for the above advantages, independent sparse gates also introduce one issue: they prohibit the method to select deep cells. Without dominated skip connections in the cell, other types of operations, *e.g.* zero operation, become freer and are widely used. Accordingly, the search algorithm can easily transform a deep cell to a shallow cell whose intermediate nodes connect with input nodes via skip connections and whose intermediate neighboring nodes are not connected via zero operations. Meanwhile, gradient descent algorithm prefers shallow cells than deep ones, as shallow cells often have more smooth landscapes and can be faster optimized. So these two factors together lead to a bias of search algorithm to shallow cells. Here we provide an example to prove the faster convergence of shallow cells.

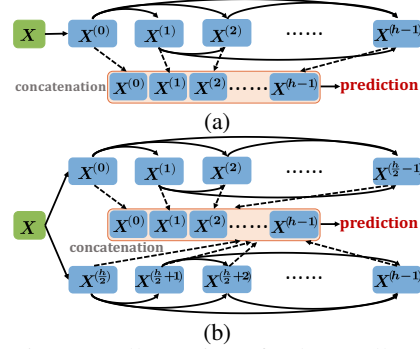

Figure 3: Illustration of a deep cell (a) and a shallow cell (b).

Suppose $X^{(l)}(l = 0, \cdots, h-1)$ are in two branches in Fig. 3 (b): nodes $X^{(0)}$ to $X^{(\frac{h}{2}-1)}$ are in one branch with input $X$ and they are connected via (7), and $X^{(l)}$ $(l = \frac{h}{2}, \cdots, h-1)$ are in another branch with input $X$ and connection (7). Next, similar to DARTS we use all intermediate nodes to obtain a squared loss in (5). Then we show in Theorem 4 that the shallow cell B in Fig. 3 (b) enjoys much faster convergence than the deep cell A in Fig. 3 (a). Note for cell B, when its node $X^{(h/2)}$ connects with node $X^{(l)}(l < h/2 - 1)$, we have very similar results.

**Theorem 4.** *Suppose the assumptions in Theorem 1 hold and for each $g_{s,t}^{(l)}$ $(0 \leq s < l \leq h-1)$ in deep cell A, it has the same value in shallow cell B if it occurs in B. When optimizing $W$ in $F_{train}(W, \beta)$ via gradient descent (6), both losses of cells A and B obey $F_{train}(W(k+1), \beta) \leq (1 - \eta \lambda'/4) F_{train}(W(k), \beta)$, where $\lambda'$ in A is defined as $\lambda_A = \frac{3c_\sigma}{4} \lambda_{\min}(K) \sum_{s=0}^{h-2} (g_{s,3}^{(h-1)})^2 \prod_{t=0}^{s-1} (g_{t,2}^{(s)})^2$, while $\lambda'$ in B becomes $\lambda_B$ and obeys $\lambda_B \geq \lambda_A + \frac{3c_\sigma}{4} \lambda_{\min}(K) \sum_{s=h/2}^{h-1} (g_{s,3}^{(h-1)})^2 \prod_{t=h/2}^{s-1} (g_{t,2}^{(s)})^2 > \lambda_A$.*

See its proof in Appendix D.3. Theorem 4 shows that when using gradient descent to optimize the inner-level loss $F_{train}(W, \beta)$ equipped with independent gates, shallow cells can faster reduce the loss $F_{train}(W, \beta)$ than deep cells. As training and validation data come from the same distribution which means $\mathbb{E}[F_{train}(W, \beta)] = \mathbb{E}[F_{val}(W, \beta)]$, shallow cells reduce $F_{val}(W, \beta)$ faster in expectation which accords with the theoretical and empirical results in [39]. So it is likely that to avoid deep cells, search algorithm would connect intermediate nodes with input nodes and cut the connection between neighboring nodes via zero operation, which is indeed illustrated by Fig. 1 (b). But it leads to cell-selection bias in the search phase, as some cells that fast decay the loss $F_{val}(W, \beta)$ at the current iteration have competitive advantage over other cells that reduce $F_{val}(W, \beta)$ slowly currently but can achieve superior final performance. This prohibits us to search good cells.

To resolve this cell-selection bias, we propose a path-depth-wise regularization to rectify the unfair competition between shallow and deep cells. From Theorem 3, the probability that $X^{(l)}$ and $X^{(l+1)}$ are connected by parameterized operations $\mathcal{O}_p$, *e.g.* various types of convolutions, is $\mathbb{P}_{l,l+1}(\beta) = \sum_{\mathcal{O}_t \in \mathcal{O}_p} \Theta(\beta_{l,t}^{(l+1)} - \tau \ln \frac{-a}{b})$. So the probability that all neighboring nodes $X^{(l)}$ and $X^{(l+1)}$ $(l = 0, \cdots, h-1)$ are connected via operations $\mathcal{O}_p$, namely, the probability of the path of depth $h$, is

$$\mathcal{L}_{path}(\beta) = \prod_{l=1}^{h-1} \mathbb{P}_{l,l+1}(\beta) = \prod_{l=1}^{h-1} \sum_{O_t \in \mathcal{O}_p} \Theta(\beta_{l,t}^{(l+1)} - \tau \ln \frac{-a}{b}). \quad (8)$$

Here we do not consider skip connection, zero and pooling operations, as they indeed make a network shallow. To rectify the competitive advantage of shallow cells over deep ones, we impose path-depth-

Table 1: Classification errors (%) on CIFAR10 (C10) and CIFAR100 (C100).

| Architecture | Test Error (%) C10 | C100 | Params (M) | Search Cost (GPU-days) | Search space #Ops/zero | Search method |
|---|---|---|---|---|---|---|
| DenseNet-BC [42] | 3.46 | 17.18 | 25.6 | — | — | manual |
| NASNet-A + cutout [2] | 2.65 | — | 3.3 | 1800 | 13 | RL |
| AmoebaNet-B + cutout [4] | 2.55 | — | 2.8 | 3150 | 19 | evolution |
| PNAS [43] | 3.41 | — | 3.2 | 225 | 8 | SMBO |
| ENAS + cutout [3] | 2.89 | — | 4.6 | 0.5 | 6 | RL |
| DARTS (first-order) + cutout [6] | 3.00 | 17.76 | 3.3 | 1.5 | 7 | gradient-based |
| DARTS (second-order) + cutout [6] | 2.76 | 17.54 | 3.3 | 4.0 | 7 | gradient-based |
| SNAS (moderate) + cutout [14] | 2.85 | — | 2.8 | 1.5 | 7 | gradient-based |
| P-DARTS + cutout [7] | 2.50 | 16.55 | 3.4 | 0.3 | 7 | gradient-based |
| BayesNAS + cutout [44] | 2.81 | — | 3.4 | 0.18 | 7 | gradient-based |
| PC-DARTS + cutout [15] | 2.81 | — | 3.6 | 0.13 | 7 | gradient-based |
| GDAS + cutout [11] | 2.93 | — | 3.4 | 0.21 | 7 | gradient-based |
| Fair DARTS + cutout [8] | 2.54 | — | 2.8 | 0.4 | 7 | gradient-based |
| PR-DARTS + cutout | 2.32 | 16.45 | 3.4 | 0.17 | 7 | gradient-based |

wised regularization $-\mathcal{L}_{\text{path}}(\boldsymbol{\beta})$ on the stochastic gates to encourage more exploration to deep cells and then decide the depth of cells instead of greedily choosing shallow cell at the beginning of search.

Now we are ready to define our proposed PR-DARTS model as follows:

$$\min_{\boldsymbol{\beta}} F_{\text{val}}(\boldsymbol{W}^*(\boldsymbol{\beta}), \boldsymbol{\beta}) + \lambda_1 \mathcal{L}_{\text{skip}}(\boldsymbol{\beta}) + \lambda_2 \mathcal{L}_{\text{non-skip}}(\boldsymbol{\beta}) - \lambda_3 \mathcal{L}_{\text{path}}(\boldsymbol{\beta}), \text{ s.t. } \boldsymbol{W}^*(\boldsymbol{\beta}) = \text{argmin}_{\boldsymbol{W}} F_{\text{train}}(\boldsymbol{W}, \boldsymbol{\beta}),$$

where $\boldsymbol{W}$ denotes network parameters, $\boldsymbol{\beta}$ denotes the parameters for the stochastic gates. Similar to DARTS, we alternatively update parameters $\boldsymbol{W}$ and $\boldsymbol{\beta}$ via gradient descent. See optimization details in Algorithm 1 of Appendix A. After searching, following DARTS, we prune redundancy connections according to the activation probability in Theorem 3 to obtain more compact cells.

## 5 Experiments

Here we evaluate PR-DARTS on classification task and compare it with representative state-of-the-art NAS approaches, including RL based NAS, EA based NAS and differential NAS methods. Code is available at https://panzhous.github.io/.

**Datasets.** CIAFR10 [40] and CIFAR100 [40] contain 50K training and 10K test images which are of size $32 \times 32$ and distribute over 10 classes in CIFAR10 and 100 classes in CIFAR100. ImageNet [41] has 1.28M training and 50K test images which roughly equally distribute over 1K object categories.

**Implementations.** In the search phase, each cell contains two input nodes (outputs of two previous cells), four intermediate nodes and one output node (concatenation of all intermediate nodes). Then we stack $k$ cells for search. The $k/3$- and $2k/3$-th cells are reduction cells in which all operations have a stride of two, and the remaining cells are normal cells with operation stride of one. Reduction cells share the same architecture and normal cells also have the same architecture. The operation set $\mathcal{O}$ has eight choices: zero operation, skip connection, $3 \times 3$ and $5 \times 5$ separable convolutions, $3 \times 3$ and $5 \times 5$ dilated separable convolutions, $3 \times 3$ average pooling and $3 \times 3$ max pooling. For fairness, all above settings follow the convention [1, 2, 4, 6]. For each cell, we use the input node which is the output of the previous cell to construct the path-depth-wise regularization in (8) as illustrated by Fig. 1 (c).

### 5.1 Results on CIFAR

In the search phase, following [6] we stack 8 cells with channel number 16. We divide 50K training samples in CIFAR10 into two equal-sized training and validation datasets. In PR-DARTS, we set $\lambda_1 = 0.01$, $\lambda_2 = 0.005$, and $\lambda_3 = 0.005$ for regularization. Then we train the network 200 epochs with mini-batch size 128. For acceleration, per iteration, we follow [11] and randomly select only two operations on each edge to update. We respectively use SGD and ADAM [45] to optimize parameters $\boldsymbol{W}$ and $\boldsymbol{\beta}$ with detailed settings in Appendix A. We set temperature $\tau = 10$ and linearly reduce it to 0.1, $a = -0.1$ and $b = 1.1$. For pruning on each node, we compare the gate activation probabilities of all non-zero operations collected from all previous nodes and retain top two operations [6] .

For evaluation on CIFAR10 and CIFAR100, we set channel number 36 and then stack 18 normal cells and 2 reduction cells (the 7- and 14-th cells) to build a large network. We train the network 600

Table 2: Classification errors (%) on ImageNet (all methods use the cells searched on CIFAR10).

| Architecture | Test Error (%) Top-1 | Top-5 | Params (M) | ×+ (M) | Search Cost (GPU-days) | Search space #Ops/zero | Search method |
|---|---|---|---|---|---|---|---|
| MobileNet [48] | 29.4 | 10.5 | 4.2 | 569 | — | — | manual |
| ShuffleNet2×(v2) [49] | 25.1 | — | ∼5 | 591 | — | — | manual |
| NASNet-A [2] | 26.0 | 8.4 | 5.3 | 564 | 1800 | 13 | RL |
| AmoebaNet-C [4] | 24.3 | 7.6 | 6.4 | 570 | 3150 | 19 | evolution |
| PNAS [43] | 25.8 | 8.1 | 5.1 | 588 | 225 | 8 | SMBO |
| MnaNet-92 [5] | 25.2 | 8.0 | 4.4 | 388 | — | hierarchical | RL |
| DARTS (second-order) [6] | 26.7 | 8.7 | 4.7 | 574 | 4.0 | 7 | gradient-based |
| SNAS (mild) [14] | 27.3 | 9.2 | 4.3 | 522 | 1.5 | 7 | gradient-based |
| P-DARTS [7] | 24.4 | 7.4 | 4.9 | 557 | 0.3 | 7 | gradient-based |
| BayesNAS [44] | 26.5 | 8.9 | 3.9 | — | 0.18 | 7 | gradient-based |
| PC-DARTS [15] | 25.1 | 7.8 | 5.3 | 586 | 0.13 | 7 | gradient-based |
| GDAS [11] | 26.0 | 8.5 | 5.3 | 581 | 0.21 | 7 | gradient-based |
| Fair DARTS [8] | 24.9 | 7.5 | 4.8 | 541 | 0.4 | 7 | gradient-based |
| PR-DARTS | 24.1 | 7.3 | 4.98 | 543 | 0.17 | 7 | gradient-based |

epochs with a mini-batch size of 128 from scratch. See detailed settings of SGD in Appendix A. We also use drop-path with probability 0.2 and cutout [46] with length 16, for regularization.

Table 1 summarizes the classification results on CIFAR10 and CIFAR100. In merely 0.17 GPU-days on Tesla V100, PR-DARTS respectively achieves 2.31% and 16.45% classification errors on CIAR10 and CIFAR100, with both search time and accuracy significantly surpassing the DARTS baseline. By comparison, PR-DARTS also consistently outperforms other NAS approaches, including differential NAS (*e.g.* P-DARTS, PC-DARTS), RL based NAS (*e.g.* NASNet), as well as EA based NAS (*e.g.*Amobdanet). These results demonstrate the superiority and transferability of the selected cells by PR-DARTS. As shown in Fig. 1, this advantage comes from the group-structured binary gates and path-depth-wise regularization in PR-DARTS which can well alleviate unfair operation competition and cell-selection bias to shallow cells which are not well considered in the compared NAS methods. Fair DARTS imposes independent sigmoid distribution and zero-one loss for each operation, which actually does not encourage the important global operation competition and cooperation. PR-DARTS runs faster over DARTS, because (1) the sparsity regularization prunes unnecessary connections as illustrated in Fig. 4 in Appendix A, and thus reduces the costs; and (2) following [11] we randomly select only two operations instead of eight operations between two nodes to update per iteration, also helping reducing cost. Note, Proxyless NAS [13] reports an error rate of 2.08% on CIAFR10, but it performs architecture search on the tree-structured PyramidNet [47] which is much complex protocol than the DARTS search space in this work, and requires much longer time (4 GPU-days) for search.

For *ablation study*, Fig. 1 shows the individual benefits of the two complementary components, group-structured binary gates and path-depth-wise regularization, in PR-DARTS. See details in Fig. 1. Due to space limit, Appendix A investigates the *effects of regularization parameters* $\lambda_1 \sim \lambda_3$ to the performance of PR-DARTS. The results show the stable performance of PR-DARTS on CIAFR10 when tuning these parameters in a relatively large range, and thus testify the robustness of PR-DARTS.

## 5.2 Results on ImageNet

We further evaluate the transferability of the cells selected on CIFAR10 by testing them on more challenging ImageNet. Following DARTS, we rescale input size to $224 \times 224$. We stack three convolutional layers,12 normal cells and 2 reduction cells (channel number 48) to build a large network, and train it 250 epochs with mini-batch size 128. See detailed settings of SGD in Appendix A.

Table 2 reports the results on ImageNet. One can observe that PR-DARTS consistently outperforms the compared state-of-the-art approaches. In particular, it respectively improves DARTS by 2.4% and 1.4% on top-1 and top 5 accuracies. These results demonstrate the superior transferability of the cells selected by PR-DARTS behind which the potential reasons have been discussed in Sec. 5.1.

## 6 Conclusion

In this work, for the first time we theoretically explicitly show the benefits of more skip connections to fast network optimization in DARTS, explaining the dominated skip connections in the selected cells by DARTS. Then inspired by our theory, we propose PR-DARTS to improve DARTS by using group-structured binary gates and path-depth-wise regularization to alleviate unfair operation competition and cell-selection bias to shallow cells. Experimental results validated the advantages of PR-DARTS.

## Broader Impacts

This work advances network architecture search (NAS) in both theoretical performance analysis and practical algorithm design. As NAS can automatically design state-of-the-art architectures, this work alleviates substantial efforts from domain experts for effective architecture design, and could also help develop more intelligent algorithms. But NAS still needs an expert-designed search space which may have bias and prohibit NAS development. So automatically designing search space is desirable.

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
