[Supplementary Material]

# Theory-Inspired Path-Regularized Differential Network Architecture Search (Supplementary File)

**Pan Zhou**     **Caiming Xiong**     **Richard Socher**     **Steven C.H. Hoi**
Salesforce Research
{pzhou, cxiong, rsocher, shoi}@salesforce.com

This supplementary document contains the technical proofs of convergence results and some additional experimental results of the NeurIPS'20 submission entitled "Theory-Inspired Path-Regularized Differential Network Architecture Search". It is structured as follows. In Appendix A, we provides more experimental results and details, including the robustness investigation of PR-DARTS to regularization parameters, effects of group-structured sparse regularization to gate activate probability, and training algorithms and details of PR-DARTS. Appendix B summarizes the notations throughout this document and also provides the existing auxiliary theories and lemmas for subsequent analysis. Then Appendix C gives the proofs of the main results in Sec. 3, namely Theorem 1, by first introducing auxiliary theories and lemmas for subsequent analysis whose proofs are deferred to Appendix E. Next, in Appendix D we presents the results in Sec. 4, including Thoerems 2, 3 and 4. Finally, Appendix E provides the proofs for auxiliary theories and lemmas in Appendix C.

## A   More Experimental Results and Details

Due to space limitation, we defer more experimental results and details to this appendix. Here we first investigate robustness of PR-DARTS to regularization parameters. Then we present effects of group-structured sparse regularization to gate activate probability, and also show the reduction cell of PR-DARTS on CIFAR10. Next, we introduce the training algorithm of PR-DARTS, and finally present more setting details of optimizers for searching architectures and retraining from scratch.

### A.1   Robustness to Regularization Parameters

Fig. 3 reports the effects of regularization parameters $\lambda_1 \sim \lambda_3$ to the performance of PR-DARTS. Due to the high training cost, we fix two regularization parameters and then investigate the third one. From Fig. 3, one can observe that for each $\lambda$ ($\lambda_1$ or $\lambda_2$ or $\lambda_3$), when tuning it in a relatively large range, $e.g.$ $\lambda_1 \in [10^{-2}, 1]$, $\lambda_2 \in [10^{-4.5}, 10^{-2.5}]$ and $\lambda_3 \in [10^{-4}, 10^{-1.5}]$, PR-DARTS has relatively stable performance on CIFAR10. This testifies the robustness of PR-DARTS to regularization parameters.

Figure 3: Effects of regularization parameters $\lambda_1 \sim \lambda_3$ to the performance of PR-DARTS.

(a) reduction cell on CIRAR10      (b) gate activate probability of normal and reduction cells

Figure 4: Visualization of search results. (a) denotes the selected reduction cell on CIRAR10. The normal cell is displayed in Fig. 1 in the manuscript. (b) shows the gate activate probability of normal cell and reduction cell in PR-DARTS.

## A.2 Effects of Group-Structured Sparse Regularization to Gate Activate Probability

Here we first display the selected reduction cell on CIRAR10 in Fig. 4 (a). The normal cell selected on CIFAR10 is displayed in Fig. 1 in the manuscript.

Next, we also report the average gate activate probability in the normal and reduction cells in Fig. 4 (b). At the beginning of the search, we initialize the activation probability of each gate to be one. This is because (1) as shown in Theorem 3, the activation probability of the gate $\boldsymbol{g}_{s,t}^{(l)}$ is $\mathbb{P}(\boldsymbol{g}_{s,t}^{(l)} \neq 0) = \Theta(\boldsymbol{\beta}_{s,t}^{(l)} - \tau \ln \frac{-a}{b})$; (2) we set $a = -0.1, b = 1.1, \boldsymbol{\beta}_{s,t}^{(l)} = 0.5$ and initialize $\tau = 10$ which leads to $\mathbb{P}(\boldsymbol{g}_{s,t}^{(l)} \neq 0) = \Theta(\boldsymbol{\beta}_{s,t}^{(l)} - \tau \ln \frac{-a}{b}) \approx 1$. In this way, all gates will be well explored. With along more iterations, the group structured sparsity regularization encourages competition and cooperation among all operations to improve the performance, and also prunes redundancy and unnecessary connections in the cells as well. To measure the overall sparsity of the normal cell, we compute its overall average activation probability $\frac{1}{|\mathcal{G}|} \sum_{\boldsymbol{g}_{s,t}^{(l)} \in \mathcal{G}} \mathbb{P}(\boldsymbol{g}_{s,t}^{(l)} \neq 0)$, where the gate set $\mathcal{G}$ collects all the operation gate in the normal cell. Similarly, we can compute the average activation probability of gates in the reduction cell. As shown in Fig. 4 (b), for both normal and reduction cells, their average gate activate probability becomes smaller with along more iterations. This indicates the activation probability of the gates on redundancy and unnecessary connections becomes smaller, which means that sparsity regularizer gradually and automatically prunes redundancy and unnecessary connections which reduces the information loss of pruning at the end of search. Moreover, this sparsity regularizer defined on the whole cell can encourage global competition and cooperation of all operations in the cell, which differs from DARTS that only introduces local competition among the operations between two nodes. Actually, sparse cell also can reduce the computation cost and boost the search efficiency.

## A.3 Algorithm Framework of PR-DARTS

In this subsection, we introduce the training algorithm of PR-DARTS in details. Same as DARTS, we alternatively update the network parameter $\boldsymbol{W}$ and the architecture parameter $\boldsymbol{\beta}$ via gradient descent which is detailed in Algorithm 1. For notation in Algorithm 1, $F_{\mathcal{B}_{\text{train}}}(\boldsymbol{W}, \boldsymbol{\beta}) = \frac{1}{|\mathcal{B}_{\text{train}}|} \sum_{(\boldsymbol{x},\boldsymbol{y}) \in \mathcal{B}_{\text{train}}} f(\boldsymbol{W}, \boldsymbol{\beta}; (\boldsymbol{X}, \boldsymbol{y}))$ denotes the training loss on mini-batch $\mathcal{B}_{\text{train}}$. Similarly, the loss $F_{\mathcal{B}_{\text{val}}}(\boldsymbol{W}, \boldsymbol{\beta})$ denotes the validation loss on mini-batch $\mathcal{B}_{\text{val}}$. When we compute the gradient $\nabla_{\boldsymbol{\beta}} F_{\mathcal{B}_{\text{train}}}(\boldsymbol{W}, \boldsymbol{\beta})$, we ignore the second-order Hessian to accelerate the computation which is the same as first-order DARTS.

---

**Algorithm 1** Searching Algorithm for PR-DARTS

---

**Input:** training dataset $\mathcal{D}_{\text{train}}$ and validation dataset $\mathcal{D}_{\text{val}}$, mini-batch size $b$, learning rate $\eta$.
**while** not convergence **do**
    sample mini-batch $\mathcal{B}_{\text{train}}$ from $\mathcal{D}_{\text{train}}$ to update $\boldsymbol{W}$ by gradient descent $\boldsymbol{W} = \boldsymbol{W} - \eta \nabla_{\boldsymbol{W}} F_{\mathcal{B}_{\text{train}}}(\boldsymbol{W}, \boldsymbol{\beta})$.
    sample mini-batch $\mathcal{B}_{\text{val}}$ from $\mathcal{D}_{\text{val}}$ to update $\boldsymbol{\beta}$ by gradient descent $\boldsymbol{\beta} = \boldsymbol{\beta} - \eta \nabla_{\boldsymbol{\beta}} F_{\mathcal{B}_{\text{val}}}(\boldsymbol{W}, \boldsymbol{\beta})$.
**end while**
**Output:** $\boldsymbol{\beta}$

---

## A.4 Algorithm Parameter Settings

**CIFAR10 and CIAFR100.** In the search phase, following DARTS, we use momentum SGD to optimize network parameter $\boldsymbol{W}$, with an initial learning rate $0.025$ (annealed down to zero via cosine

decay [1]), a momentum of 0.9, and a weight decay of $3 \times 10^{-4}$. Architecture parameter $\beta$ is updated by ADAM [2] with a learning rate of $3 \times 10^{-4}$ and a weight decay of $10^{-3}$. For evaluation on CIFAR10 and CIFAR100, we use momentum SGD with an initial learning 0.025 (cosine decayed to zero), a momentum of 0.9, a weight decay of $3 \times 10^{-4}$, and gradient norm clipping parameter 5.0.

**ImageNet.** We evaluate the transfer ability of the cells selected on CIFAR10 by testing them on ImageNet. Following DARTS, we use momentum SGD with an initial learning 0.025 (cosine decayed to zero), a momentum of 0.9, a weight decay of $3 \times 10^{-4}$, and gradient norm clipping parameter 5.0.

## B  Notation and Preliminarily

### B.1  Notations

In this document, we use $\boldsymbol{X}_i^{(l)}(k)$ to denote the output $\boldsymbol{X}_i^{(l)}$ of the $i$-th sample in the $l$-th layer at the $k$-th iteration. For brevity, we usually ignore the notation $(k)$ and $i$ and use $\boldsymbol{X}^{(l)}$ to denote the output $\boldsymbol{X}^{(l)}$ of any sample $\boldsymbol{X}_i$ ($\forall i = 1, \cdots, n$) in the $l$-th layer at any iteration. We use $\boldsymbol{\Omega} = \{\boldsymbol{W}^{(0)}, \boldsymbol{W}_0^{(1)}, \boldsymbol{W}_0^{(2)}, \boldsymbol{W}_1^{(2)}, \cdots, \boldsymbol{W}_0^{(l)}, \cdots, \boldsymbol{W}_{l-1}^{(l)}, \cdots, \boldsymbol{W}_0^{h-1}, \cdots, \boldsymbol{W}_{h-2}^{(h-1)}, \boldsymbol{W}_0, \cdots, \boldsymbol{W}_{h-1}\}$ to denote the set of all $\frac{h(h+3)}{2}$ learnable matrix parameters, including the convolution parameters $\boldsymbol{W}_s^{(l)}$ and the linear mapping parameters $\boldsymbol{W}_s$. Let $\boldsymbol{\Omega}_i$ denote the $i$-th matrix parameters in $\boldsymbol{\Omega}$, $e.g.$ $\boldsymbol{\Omega}_1 = \boldsymbol{W}^{(0)}$. For notation simplicity, here we assume the input size is $m \times p$ to avoid using $\bar{m} \times \bar{p}$. The operation $\mathsf{vec}\,(\boldsymbol{X})$ vectorizes the matrix $\boldsymbol{X}$.

Then we define the loss

$$F(\boldsymbol{\Omega}) = \frac{1}{2n}\|\boldsymbol{y} - \boldsymbol{u}(k)\|_2^2 = \frac{1}{2n}\sum_{i=1}^n (y_i - u_i)^2 = \frac{1}{n}\sum_{i=1}^n \ell_i,$$

where $\boldsymbol{u}(k) = [u_1(k); u_2(k); \cdots, u_n(k)] \in \mathbb{R}^n$ denotes the prediction at the $k$-th iteration, $\boldsymbol{y} = [y_1; y_2; \cdots, y_n] \in \mathbb{R}^n$ is the labels for the $n$ samples $\{\boldsymbol{X}_i\}_{i=1}^n$, and $\ell_i = (y_i - u_i)^2$ denotes the individual loss of the $i$-th sample $\boldsymbol{X}_i$.

Then for brevity, $\ell(\boldsymbol{\Omega})$ and $\ell_i(\boldsymbol{\Omega})$ respectively denote the losses when feeding the input $(\boldsymbol{X}, \boldsymbol{y})$ and $(\boldsymbol{X}_i, y_i)$. Then we denote the gradient of $\ell(\boldsymbol{\Omega})$ with respect to all learnable parameters $\boldsymbol{\Omega}$ as

$$\nabla_{\boldsymbol{\Omega}}\ell(\boldsymbol{\Omega}) = \left[\mathsf{vec}\left(\frac{\partial\ell}{\partial\boldsymbol{W}^{(0)}}\right); \left\{\mathsf{vec}\left(\frac{\partial\ell}{\partial\boldsymbol{W}_s^{(l)}}\right)\right\}_{0\leq l\leq h-1, 0\leq s\leq l-1}; \left\{\mathsf{vec}\left(\frac{\partial\ell}{\partial\boldsymbol{W}_s}\right)\right\}_{0\leq s\leq h-1}\right],$$

where the $\mathsf{vec}\,(\boldsymbol{X})$ operation vectorizes the matrix $\boldsymbol{X}$ into vector. Here we also let $\nabla_{\boldsymbol{\Omega}_i}\ell(\boldsymbol{\Omega})$ denotes the gradient of $\ell(\boldsymbol{\Omega})$ with the $i$-th matrix parameter, $e.g.$ $\nabla_{\boldsymbol{\Omega}_1}\ell(\boldsymbol{\Omega}) = \mathsf{vec}\left(\frac{\partial\ell}{\partial\boldsymbol{W}^{(0)}}\right)$. Therefore, $\nabla_{\boldsymbol{\Omega}}F(\boldsymbol{\Omega}) = \frac{1}{n}\sum_{i=1}^n \nabla_{\boldsymbol{\Omega}}\partial\ell_i(\boldsymbol{\Omega})$ where $\ell_i(\boldsymbol{\Omega})$ is the loss given input $(\boldsymbol{X}_i, y_i)$. In this way, we can define the Gram matrix $\boldsymbol{G}(k) \in \mathbb{R}^{n \times n}$ at the $k$-th iteration in which its $(i, j)$-th entry is defined as

$$\boldsymbol{G}_{ij}(k) = \langle\nabla_{\boldsymbol{\Omega}}\ell_i(\boldsymbol{\Omega}(k)), \nabla_{\boldsymbol{\Omega}}\ell_j(\boldsymbol{\Omega}(k))\rangle,$$

where $\nabla_{\boldsymbol{\Omega}}\ell_i(\boldsymbol{\Omega}(k))$ denote the gradient of the loss $\ell_i$ on the $i$-th sample $(\boldsymbol{X}_i, y_i)$ with respect to all parameter $\boldsymbol{\Omega}$ at the $k$-th iteration. We often ignore the notation $k$ and use $\boldsymbol{G}$ to denote the Gram matrix that does not depend on iteration number $k$.

According to the definitions, we have

$$\boldsymbol{G}_{ij}(k) = \langle\nabla_{\boldsymbol{\Omega}}\ell_i(\boldsymbol{\Omega}(k)), \nabla_{\boldsymbol{\Omega}}\ell_j(\boldsymbol{\Omega}(k))\rangle = \sum_{t=1}^{\frac{h(h+3)}{2}} \langle\nabla_{\boldsymbol{\Omega}_t}\ell_i(\boldsymbol{\Omega}(k)), \nabla_{\boldsymbol{\Omega}_t}\ell_j(\boldsymbol{\Omega}(k))\rangle$$

$$= \left\langle\frac{\partial\ell_i}{\partial\boldsymbol{W}^{(0)}(k)}, \frac{\partial\ell_j}{\partial\boldsymbol{W}^{(0)}(k)}\right\rangle + \sum_{l=1}^{h-1}\sum_{s=0}^{l-1}\left\langle\frac{\partial\ell_i}{\partial\boldsymbol{W}_s^{(l)}(k)}, \frac{\partial\ell_j}{\partial\boldsymbol{W}_s^{(l)}(k)}\right\rangle + \sum_{s=0}^{h-1}\left\langle\frac{\partial\ell_i}{\partial\boldsymbol{W}_s(k)}, \frac{\partial\ell_j}{\partial\boldsymbol{W}_s(k)}\right\rangle$$

For brevity, we let

$$\bar{\boldsymbol{G}}_{ij}^0(k) = \left\langle\frac{\partial\ell_i}{\partial\boldsymbol{W}^{(0)}(k)}, \frac{\partial\ell_j}{\partial\boldsymbol{W}^{(0)}(k)}\right\rangle, \quad \boldsymbol{G}_{ij}^{ls}(k) = \left\langle\frac{\partial\ell_i}{\partial\boldsymbol{W}_s^{(l)}(k)}, \frac{\partial\ell_j}{\partial\boldsymbol{W}_s^{(l)}(k)}\right\rangle, \quad \boldsymbol{G}_{ij}^s(k) = \left\langle\frac{\partial\ell_i}{\partial\boldsymbol{W}_s(k)}, \frac{\partial\ell_j}{\partial\boldsymbol{W}_s(k)}\right\rangle.$$

Therefore, we have

$$\boldsymbol{G}_{ij}(k) = \bar{\boldsymbol{G}}_{ij}^0(k) + \sum_{l=1}^{h-1}\sum_{s=0}^{l-1}\boldsymbol{G}_{ij}^{ls}(k) + \sum_{s=0}^{h-1}\boldsymbol{G}_{ij}^s(k), \quad \boldsymbol{G}(k) = \bar{\boldsymbol{G}}^0(k) + \sum_{l=1}^{h-1}\sum_{s=0}^{l-1}\boldsymbol{G}^{ls}(k) + \sum_{s=0}^{h-1}\boldsymbol{G}^s(k).$$

Finally, since we need to compute the gradient. Here we define an operation for computing the gradient for convolution operation. For back-propagate, we define the inverse operation of $\Phi(\boldsymbol{X})$ as $\Psi\left(\frac{1}{\tau}\Phi(\boldsymbol{X})\right) = \boldsymbol{X} \in \mathbb{R}^{m\times p}$. For the $(i,j)$-th entry in $\Psi(\boldsymbol{X})$, it equals to the sum of all $\boldsymbol{X}_{i,j}$ in $\Phi(\boldsymbol{X})$.

### B.2 Auxiliary Lemmas

**Lemma 1.** *[3][Chebyshev's inequality] For any variable $x$, we have*

$$\mathbb{P}\left(|x - \mathbb{E}[x]| \geq a\right) \leq \frac{\mathsf{Var}(x)}{a^2},$$

*where $a$ is a positive constant, $\mathsf{Var}(x)$ denotes the variance of $x$.*

**Lemma 2.** *[4] Given a set of matrices $\{\boldsymbol{A}_i, \boldsymbol{B}_i\}$ with proper sizes, if $\|\boldsymbol{A}_i\|_2 \leq a_i$ and $\|\boldsymbol{B}_i\|_2 \leq a_i$ and $\|\boldsymbol{A}_i - \boldsymbol{B}_i\|_F \leq b_i a_i$, we have*

$$\left\|\prod_{i=1}^n \boldsymbol{A}_i - \prod_{i=1}^n \boldsymbol{B}_i\right\|_F \leq \left(\sum_{i=1}^n b_i\right)\prod_{i=1}^n a_i.$$

**Lemma 3.** *[5][Cauchy Interlace Theorem] Let $\boldsymbol{A}$ be a Hermitian matrix of order $n$ and let $\boldsymbol{B}$ be a principal submatrix of $\boldsymbol{A}$ of order $n-1$. If $\lambda_n \leq \lambda_{n-1} \leq \cdots \leq \lambda_1$ lists the eigenvalues of $\boldsymbol{A}$ and $\mu_n \leq \mu_{n-1} \leq \cdots \leq \mu_2$ the eigenvalues of $\boldsymbol{B}$, then $\lambda_n \leq \mu_n \leq \lambda_{n-1} \leq \mu_{n-1} \cdots \leq \lambda_2 \leq \mu_2 \leq \lambda_1$.*

**Lemma 4.** *[6][Chi-Square Variable Bound] Let $x$ be chi-square variable with $n$ degree of freedom. Then for any $t > 0$, it holds*

$$\mathbb{P}\left(x - n \geq 2\sqrt{nt} + 2t\right) \leq \exp(-t), \quad and \quad \mathbb{P}\left(x - n \leq -2\sqrt{nt}\right) \leq \exp(-t).$$

**Lemma 5.** *[4] Suppose $\sigma$ is analytic and not a polynomial function. Consider data $\{\boldsymbol{X}_{i=1}^n\}_{i=1}^n$ are not parallel, namely $\mathsf{vec}(\boldsymbol{X}_i) \notin span(\mathsf{vec}(\boldsymbol{X}_j))$ for all $i \neq j$, Then the smallest eigenvalue the matrix $\boldsymbol{G}$ which is defined as*

$$\boldsymbol{G}(\boldsymbol{X})_{ij} = \mathbb{E}_{\boldsymbol{W}\sim\mathcal{N}(0,\boldsymbol{I})}\ \sigma(\langle\boldsymbol{W}, \boldsymbol{X}_i\rangle)\sigma(\langle\boldsymbol{W}, \boldsymbol{X}_j\rangle)$$

*is larger than zero, namely $\lambda_{\min}(\boldsymbol{G}) > 0$.*

**Lemma 6.** *[4] Suppose $\sigma$ is analytic and not a polynomial function. Consider data $\{\boldsymbol{X}_{i=1}^n\}_{i=1}^n$ are not parallel, namely $\mathsf{vec}(\boldsymbol{X}_i) \notin span(\mathsf{vec}(\boldsymbol{X}_j))$ for all $i \neq j$, Then the smallest eigenvalue the matrix $\boldsymbol{G}$ which is defined as*

$$\boldsymbol{G}(\boldsymbol{X})_{ij} = \mathbb{E}_{\boldsymbol{W}\sim\mathcal{N}(0,\boldsymbol{I})}\ \sigma'(\langle\boldsymbol{W}, \boldsymbol{X}_i\rangle)\sigma'(\langle\boldsymbol{W}, \boldsymbol{X}_j\rangle)$$

*is larger than zero, namely $\lambda_{\min}(\boldsymbol{G}) > 0$.*

**Lemma 7.** *[4] Suppose the activation function $\sigma(\cdot)$ satisfies Assumption 1. Suppose there exists $c > 0$ such that*

$$\boldsymbol{A} = \begin{bmatrix} a_1^2 & \rho a_1 b_1 \\ \rho_1 a_1 b_1 & b_1^2 \end{bmatrix} \succ 0, \qquad \boldsymbol{B} = \begin{bmatrix} a_2^2 & \rho_2 a_2 b_2 \\ \rho a_2 b_2 & b_2^2 \end{bmatrix} \succ 0,$$

*where the parameter satisfies $1/c \leq x \leq c$ in which $x$ could be $a_1$, $a_2$, $b_1$, $b_2$. Let $g(\boldsymbol{A}) = \mathbb{E}_{(u,v)\sim\mathcal{N}(0,\boldsymbol{A})}\sigma(u)\sigma(v)$. Then we have*

$$|g(\boldsymbol{A}) - g(\boldsymbol{B})| \leq c\|\boldsymbol{A} - \boldsymbol{B}\|_F \leq 2C\|\boldsymbol{A} - \boldsymbol{B}\|_\infty,$$

*where $C$ is a constant that only depends on $c$ and the Lipschitz and smooth parameter of $\sigma(\cdot)$.*

## C Proofs of Results in Sec. 3

### C.1 Proof of Theorem 1

Suppose Assumptions 1, 2 and 3 hold. To prove our main results, namely the results in Theorem 1, we have two steps. In the first step, from Lemma 21, we have that if $m$ and $\eta$ satisfy

$$m \geq \frac{c_m' c^2 \rho k_c^2 c_{w0}^2 \mu^2}{\lambda^2 n}, \quad \eta \leq \frac{c_\eta' \lambda}{\sqrt{m}\mu^4 h^3 k_c^2 c^4},$$

where $c'_m$ and $c'_\eta$ are two constants, $c = \left(1 + \boldsymbol{\alpha}_2 + 2\boldsymbol{\alpha}_3\mu\sqrt{k_c}c_{w0}\right)^h$, $\boldsymbol{\alpha}_2 = \max_{s,l}\boldsymbol{\alpha}_{s,2}^{(l)}$ and $\boldsymbol{\alpha}_3 = \max_{s,l}\boldsymbol{\alpha}_{s,3}^{(l)}$. Then with probability at least $1 - \delta/2$ we have

$$\|\boldsymbol{y} - \boldsymbol{u}(k)\|_2^2 \le \left(1 - \frac{\eta\lambda_{\min}\left(\boldsymbol{G}(0)\right)}{4}\right)\|\boldsymbol{y} - \boldsymbol{u}(k-1)\|_2^2.$$

where $k$ denotes the iteration number, $\lambda_{\min}\left(\boldsymbol{G}(0)\right)$ denotes the smallest eigenvalue of the Gram matrix $\boldsymbol{G}(0)$ at the initialization. For this part, we prove it in Appendix C.3.

In the second step, we will prove that the smallest eigenvalue of can be lower bounded. Specifically, we prove this results in Lemma 24: if $m \ge \frac{c_4\mu^2p^2n^2\log(n/\delta)}{\lambda^2}$, it holds that with probability at least $1 - \delta/2$, the smallest eigenvalue the matrix $\boldsymbol{G}$ satisfies

$$\lambda_{\min}\left(\boldsymbol{G}(0)\right) \ge \frac{3c_\sigma}{4}\sum_{s=0}^{h-1}(\boldsymbol{\alpha}_{s,3}^{(h)})^2\left(\prod_{t=0}^{s-1}(\boldsymbol{\alpha}_{t,2}^{(s)})^2\right)\lambda_{\min}(\boldsymbol{K}).$$

where $\lambda = 3c_\sigma\sum_{s=0}^{h-1}(\boldsymbol{\alpha}_{s,3}^{(h)})^2\left(\prod_{t=0}^{s-1}(\boldsymbol{\alpha}_{t,2}^{(s)})^2\right)\lambda_{\min}(\boldsymbol{K})$, $c_\sigma$ is a constant that only depends on $\sigma$ and the input data, $\lambda_{\min}(\boldsymbol{K}) = \min_{i,j}\lambda_{\min}(\boldsymbol{K}_{ij})$ is larger than zero in which $\lambda_{\min}(\boldsymbol{K}_{ij})$ is the the smallest eigenvalue of $\boldsymbol{K}_{ij} = \begin{bmatrix} \boldsymbol{X}_i^\top\boldsymbol{X}_j, \boldsymbol{X}_i^\top\boldsymbol{X}_j \\ \boldsymbol{X}_j^\top\boldsymbol{X}_i, \boldsymbol{X}_j^\top\boldsymbol{X}_j \end{bmatrix}$. Appendix C.4 provides the proof for this result.

Finally, we combine these results in the above two steps and can obtain that if $m \ge \frac{c_m\mu^2}{\lambda^2}\left[\rho p^2n^2\log(n/\delta)+c^2k_c^2c_{w0}^2/n\right]$ and $\eta \le \frac{c_\eta\lambda}{\sqrt{m}\mu^4h^3k_c^2c^4}$, where $c_{w0}, c_m, c_\eta$ are constants, with probability at least $(1-\delta/2)^2 > 1-\delta$, we have

$$\|\boldsymbol{y} - \boldsymbol{u}(k)\|_2^2 \le (1-\eta\lambda/4)\|\boldsymbol{y} - \boldsymbol{u}(k-1)\|_2^2 \quad (\forall k \ge 1),$$

where $\lambda = \frac{3c_\sigma}{4}\lambda_{\min}(\boldsymbol{K})\sum_{s=0}^{h-2}(\boldsymbol{\alpha}_{s,3}^{(h-1)})^2\prod_{t=0}^{s-1}(\boldsymbol{\alpha}_{t,2}^{(s)})^2$, the positive constant $c_\sigma$ only depends on $\sigma$ and input data. On the other hand, we have

$$F_{\text{train}}(\boldsymbol{W}(k+1),\boldsymbol{\beta}) = \frac{1}{2n}\|\boldsymbol{y} - \boldsymbol{u}(k+1)\|_2^2,$$

then we can obtain the desired results in Theorem 1. Please refer to the proof details in Appendix C.3 and C.4 for the above two steps respectively.

Note that our proof framework is similar to [4]. But there are essential differences. The main difference is that here our network architecture is much complex (e.g. each layer connects all the previous layers) and each edge in our network also involves more operations, including zero operation, skip operation and convolution operation, which requires bounding many terms in this work differently and more elaborately.

For the following proofs, Appendix C.2 provides the auxiliary lemmas for the proofs for Step 1 and Step 2. Then Appendix C.3 and C.4 respectively present the proof details in Step 1 and Step 2.

## C.2 Auxiliary Lemmas

**Lemma 8.** *The gradient of the loss $\ell = \frac{1}{2}(u-y)^2$ with parameter and temporary output can be written as follows:*

$$\frac{\partial\ell}{\partial\boldsymbol{X}^{(l)}} = (u-y)\boldsymbol{W}_l + \sum_{s=l+1}^{h-1}\left(\boldsymbol{\alpha}_{l,2}^{(s)}\frac{\partial\ell}{\partial\boldsymbol{X}^{(s)}} + \boldsymbol{\alpha}_{l,3}^{(s)}\tau\Psi\left((\boldsymbol{W}_l^{(s)})^\top\left(\sigma'\left(\boldsymbol{W}_l^{(s)}\Phi(\boldsymbol{X}^{(l)})\right)\odot\frac{\partial\ell}{\partial\boldsymbol{X}^{(s)}}\right)\right)\right),$$

$$(0 \le l \le h-1, 0 \le s \le l-1),$$

$$\frac{\partial\ell}{\partial\boldsymbol{X}} = \tau\Psi\left((\boldsymbol{W}^{(0)})^\top\left(\sigma'\left(\boldsymbol{W}^{(0)}\Phi(\boldsymbol{X})\right)\odot\frac{\partial\ell}{\partial\boldsymbol{X}^{(0)}}\right)\right) \in \mathbb{R}^{m\times p},$$

$$\frac{\partial\ell}{\partial\boldsymbol{W}_s^{(l)}} = \boldsymbol{\alpha}_{s,3}^{(l)}\tau\Phi(\boldsymbol{X}^{(s)})\left(\sigma'\left(\boldsymbol{W}_s^{(l)}\Phi(\boldsymbol{X}^{(s)})\right)\odot\frac{\partial\ell}{\partial\boldsymbol{X}^{(l)}}\right)^\top \in \mathbb{R}^{m\times p} (0 \le l \le h-1, 0 \le s \le l-1),$$

$$\frac{\partial\ell}{\partial\boldsymbol{W}^{(0)}} = \tau\Phi(\boldsymbol{X})\left(\sigma'\left(\boldsymbol{W}^{(0)}\Phi(\boldsymbol{X})\right)\odot\frac{\partial\ell}{\partial\boldsymbol{X}^{(0)}}\right)^\top \in \mathbb{R}^{m\times p},$$

$$\frac{\partial\ell}{\partial\boldsymbol{W}_s} = (u-y)\boldsymbol{X}^{(s)} \in \mathbb{R}^{m\times p},$$

*where $\odot$ denotes the dot product, $\frac{\partial\ell}{\partial\boldsymbol{X}^{(l)}} \in \mathbb{R}^{m\times p}$.*

See its proof in Appendix E.1.

**Lemma 9.** *The gradient of the network output $u$ with respect to the output and convolution parameter can be written as follows:*

$$\frac{\partial u}{\partial \boldsymbol{X}^{(l)}} = \boldsymbol{W}_l + \sum_{s=l+1}^{h-1} \left( \boldsymbol{\alpha}_{l,2}^{(s)} \frac{\partial u}{\partial \boldsymbol{X}^{(s)}} + \boldsymbol{\alpha}_{l,3}^{(s)} \tau \Psi \left( (\boldsymbol{W}_l^{(s)})^\top \left( \sigma' \left( \boldsymbol{W}_l^{(s)} \Phi(\boldsymbol{X}^{(l)}) \right) \odot \frac{\partial u}{\partial \boldsymbol{X}^{(s)}} \right) \right) \right),$$

$$(0 \leq l \leq h-1, 0 \leq s \leq l-1),$$

$$\frac{\partial u}{\partial \boldsymbol{X}} = \tau \Psi \left( (\boldsymbol{W}^{(0)})^\top \left( \sigma' \left( \boldsymbol{W}^{(0)} \Phi(\boldsymbol{X}) \right) \odot \frac{\partial u}{\partial \boldsymbol{X}^{(0)}} \right) \right) \in \mathbb{R}^{m \times p},$$

$$\frac{\partial u}{\partial \boldsymbol{W}_s^{(l)}} = \boldsymbol{\alpha}_{s,3}^{(l)} \tau \Phi(\boldsymbol{X}^{(s)}) \left( \sigma' \left( \boldsymbol{W}_s^{(l)} \Phi(\boldsymbol{X}^{(s)}) \right) \odot \frac{\partial u}{\partial \boldsymbol{X}^{(l)}} \right)^\top \in \mathbb{R}^{m \times p} \ (0 \leq l \leq h-1, 0 \leq s \leq l-1),$$

$$\frac{\partial u}{\partial \boldsymbol{W}^{(0)}} = \tau \Phi(\boldsymbol{X}) \left( \sigma' \left( \boldsymbol{W}^{(0)} \Phi(\boldsymbol{X}) \right) \odot \frac{\partial u}{\partial \boldsymbol{X}^{(0)}} \right)^\top \in \mathbb{R}^{m \times p},$$

$$\frac{\partial u}{\partial \boldsymbol{W}_s} = \boldsymbol{X}^{(s)} \in \mathbb{R}^{m \times p}, \ (0 \leq s \leq h-1),$$

*where $\odot$ denotes the dot product and $\frac{\partial u}{\partial \boldsymbol{X}^{(l)}} \in \mathbb{R}^{m \times p}$.*

See its proof in Appendix E.2.

**Lemma 10.** *Suppose Assumptions 1, 2 and 3 hold. Given a constant $\delta \in (0,1)$, assume $m \geq \frac{16 c_1 n p^2}{c^2 \delta}$, where $c_1 = \sigma^4(0) + 4|\sigma^3(0)|\mu\sqrt{2/\pi} + 8|\sigma(0)|\mu^3\sqrt{2/\pi} + 32\mu^4$ and $c = \mathbb{E}_{\omega \sim \mathcal{N}(0, \frac{1}{\sqrt{p}})} \left[ \sigma^2(\omega) \right]$. Suppose $\boldsymbol{W}_s^{(l)}(0) \leq \sqrt{m} c_{w0} \ \forall 0 \leq l \leq h, 0 \leq s \leq l-1$. Then with probability at least $1 - \delta/4$, we have*

$$\frac{1}{c_{x0}} \leq \|\boldsymbol{X}^{(l)}(0)\|_F \leq c_{x0}.$$

*where $c_{x0} \geq 1$ is a constant.*

See its proof in Appendix E.3.

**Lemma 11.** *Suppose Assumptions 1, 2 and 3 hold. Assume $\|\boldsymbol{W}_s^l(0)\|_2 \leq \sqrt{m} c_{w0}$, $\|\boldsymbol{W}_s^l(k) - \boldsymbol{W}_s^l(0)\|_F \leq \sqrt{m} r$. Then for $\forall l$, we have*

$$\|\boldsymbol{X}^{(l)}(k) - \boldsymbol{X}^{(l)}(0)\|_F \leq \left( 1 + \boldsymbol{\alpha}_2 + \boldsymbol{\alpha}_3 \mu \sqrt{k_c} (r + c_{w0}) \right)^l \mu \sqrt{k_c} r,$$

$$\left\| \boldsymbol{W}_s^{(l)}(k) \Phi(\boldsymbol{X}^{(s)}(k)) - \boldsymbol{W}_s^{(l)}(0) \Phi(\boldsymbol{X}^{(s)}(0)) \right\|_F \leq \frac{1}{\boldsymbol{\alpha}_3} \left( 1 + \boldsymbol{\alpha}_2 + \boldsymbol{\alpha}_3 \mu \sqrt{k_c} (r + c_{w0}) \right)^l \sqrt{k_c} m r,$$

*where $\boldsymbol{\alpha}_2 = \max_{s,l} \boldsymbol{\alpha}_{s,2}^{(l)}$ and $\boldsymbol{\alpha}_3 = \max_{s,l} \boldsymbol{\alpha}_{s,3}^{(l)}$, and $c_{x0} \geq 1$ is given in Lemma 10.*

See its proof in Appendix E.4

**Lemma 12.** *Suppose Assumptions 1, 2 and 3 hold. Assume $\frac{1}{\sqrt{n}} \|\boldsymbol{u}(t) - \boldsymbol{y}\|_F = c_y$ and $\|\boldsymbol{W}_h(t)\|_F \leq c_u$, $\|\boldsymbol{W}_l^{(s)}(t) - \boldsymbol{W}_l^{(s)}(0)\|_F \leq \sqrt{m} r$, and $\|\boldsymbol{W}_l^{(s)}(0)\|_F \leq \sqrt{m} c_{w0}$. Then for $\forall l$, we have*

$$\frac{1}{n} \sum_{i=1}^{n} \left\| \frac{\partial \ell}{\partial \boldsymbol{X}_i^{(l)}(t)} \right\|_F \leq \left( 1 + \boldsymbol{\alpha}_2 + \boldsymbol{\alpha}_3 \mu \sqrt{k_c} (r + c_{w0}) \right)^l c_y c_u,$$

*where $\boldsymbol{\alpha}_2 = \max_{s,l} \boldsymbol{\alpha}_{s,2}^{(l)}$ and $\boldsymbol{\alpha}_3 = \max_{s,l} \boldsymbol{\alpha}_{s,3}^{(l)}$.*

See its proof in Appendix E.5.

**Lemma 13.** *Suppose Assumptions 1, 2 and 3 hold. Assume $\|\boldsymbol{y} - \boldsymbol{u}(t)\|_2^2 \leq (1 - \frac{\eta\lambda}{2})^t \|\boldsymbol{y} - \boldsymbol{u}(0)\|_2^2$ holds for $t = 1, \cdots, k$. Then by setting*

$$\widetilde{r} = \frac{8 c_{x0} \|\boldsymbol{y} - \boldsymbol{u}(0)\|_2}{\lambda \sqrt{mn}} \max \left( 1, 2 \left( 1 + \boldsymbol{\alpha}_2 + 2\boldsymbol{\alpha}_3 \mu \sqrt{k_c} c_{w0} \right)^l \boldsymbol{\alpha}_{s,3}^{(l)} \mu \sqrt{k_c} c_{w0} \right) \leq c_{w0},$$

*we have that for any $s = 1, \cdots, k+1$,*

$$\|\boldsymbol{W}^{(0)}(t) - \boldsymbol{W}^{(0)}(0)\|_F \leq \sqrt{m}\widetilde{r}, \quad \|\boldsymbol{W}_s^{(l)}(t) - \boldsymbol{W}_s^{(l)}(0)\|_F \leq \sqrt{m}\widetilde{r}, \quad \|\boldsymbol{W}_s(t) - \boldsymbol{W}_s(0)\|_F \leq \sqrt{m}\widetilde{r},$$

$$\|\boldsymbol{W}^{(0)}(t+1) - \boldsymbol{W}^{(0)}(t)\|_F = \eta \left\|\frac{\partial F(\Omega)}{\partial \boldsymbol{W}^{(0)}(t)}\right\|_F \leq \frac{4c\eta\mu c_{x0}c_{w0}\sqrt{k_c}}{\sqrt{n}}\|\boldsymbol{u}(t) - \boldsymbol{y}\|_2,$$

$$\|\boldsymbol{W}_s^{(l)}(t+1) - \boldsymbol{W}_s^{(l)}(t)\|_F = \eta \left\|\frac{\partial F(\Omega)}{\partial \boldsymbol{W}_s^{(l)}(t)}\right\|_F \leq \frac{4c\boldsymbol{\alpha}_{s,3}^{(l)}\mu c_{x0}c_{w0}\sqrt{k_c}}{\sqrt{n}}\|\boldsymbol{u}(t) - \boldsymbol{y}\|_2,$$

$$\|\boldsymbol{W}_s(t+1) - \boldsymbol{W}_s(t)\|_F = \eta \left\|\frac{\partial F(\Omega)}{\partial \boldsymbol{W}_s(t)}\right\|_F \leq \frac{2\eta c_{x0}}{\sqrt{n}}\|\boldsymbol{u}(t) - \boldsymbol{y}\|_2,$$

*where $c = \left(1 + \boldsymbol{\alpha}_2 + 2\boldsymbol{\alpha}_3\mu\sqrt{k_c}c_{w0}\right)^l$ with $\boldsymbol{\alpha}_2 = \max_{s,l}\boldsymbol{\alpha}_{s,2}^{(l)}$ and $\boldsymbol{\alpha}_3 = \max_{s,l}\boldsymbol{\alpha}_{s,3}^{(l)}$.*

See its proof in Appendix E.6.

**Lemma 14.** *Suppose Assumptions 1, 2 and 3 hold. Then we have*

$$\left\|\boldsymbol{X}^{(l)}(k+1) - \boldsymbol{X}^{(l)}(k)\right\|_F$$

$$\leq \left(1 + \boldsymbol{\alpha}_2 + 2\sqrt{k_c}c_{w0}\boldsymbol{\alpha}_3\mu\right)^l \left(1 + \frac{2(\boldsymbol{\alpha}_3)^2 c_{x0}}{(\boldsymbol{\alpha}_2 + 2\sqrt{k_c}c_{w0}\boldsymbol{\alpha}_3\mu)\sqrt{n}}\right)\frac{4c\tau\eta\mu^2 c_{x0}c_{w0}k_c}{\sqrt{n}}\|\boldsymbol{u}(k) - \boldsymbol{y}\|_F,$$

*where $\boldsymbol{\alpha}_2 = \max_{s,l}\boldsymbol{\alpha}_{s,2}^{(l)}$ and $\boldsymbol{\alpha}_3 = \max_{s,l}\boldsymbol{\alpha}_{s,3}^{(l)}$.*

See its proof in Appendix E.7.

**Lemma 15.** *Suppose Assumptions 1, 2 and 3 hold. Then we have*

$$\left\|\boldsymbol{W}^{(0)}(k)\right\|_F \leq 2\sqrt{m}c_{w0}, \quad \left\|\boldsymbol{W}_s^{(l)}(k)\right\|_F \leq 2\sqrt{m}c_{w0}, \quad \|\boldsymbol{W}_s(k)\|_F \leq 2\sqrt{m}c_{w0}.$$

*If $\widetilde{r}$ in Lemma 13 satisfies $\widetilde{r} \leq \frac{c_{x0}}{\left(1 + \boldsymbol{\alpha}_2 + 2\boldsymbol{\alpha}_3\mu\sqrt{k_c}c_{w0}\right)^l \mu\sqrt{k_c}}$ which can be achieved by using large $m$, then we have*

$$\left\|\boldsymbol{X}_i^{(l)}(k)\right\|_F \leq 2c_{x0},$$

*where $\boldsymbol{\alpha}_2 = \max_{s,l}\boldsymbol{\alpha}_{s,2}^{(l)}$ and $\boldsymbol{\alpha}_3 = \max_{s,l}\boldsymbol{\alpha}_{s,3}^{(l)}$.*

See its proof in Appendix E.8.

**Lemma 16.** *Suppose Assumptions 1, 2 and 3 hold. Then we have*

$$\|\boldsymbol{X}_i^{(0)}(k) - \boldsymbol{X}_i^{(0)}(0)\|_F \leq \mu\sqrt{k_c}\widetilde{r}, \quad \|\boldsymbol{X}_i^{(l)}(k) - \boldsymbol{X}_i^{(l)}(0)\|_F \leq c(1 + 2\boldsymbol{\alpha}_3 c_{x0})\mu\sqrt{k_c}\widetilde{r},$$

*where $c = \left(1 + \boldsymbol{\alpha}_2 + 2\boldsymbol{\alpha}_3\mu\sqrt{k_c}c_{w0}\right)^l$ with $\boldsymbol{\alpha}_2 = \max_{s,l}\boldsymbol{\alpha}_{s,2}^{(l)}$ and $\boldsymbol{\alpha}_3 = \max_{s,l}\boldsymbol{\alpha}_{s,3}^{(l)}$. Here $\widetilde{r}$ is given in Lemma 13.*

See its proof in Appendix E.9.

**Lemma 17.** *Suppose Assumptions 1, 2 and 3 hold.*

$$|u_i(k) - u_i(0)| \leq 2\sqrt{m}h\left(c_{x0} + c_{w0}c(1 + 2\boldsymbol{\alpha}_3 c_{x0})\mu\sqrt{k_c}\right)\widetilde{r},$$

*where $c = \left(1 + \boldsymbol{\alpha}_2 + 2\boldsymbol{\alpha}_3\mu\sqrt{k_c}c_{w0}\right)^l$ with $\boldsymbol{\alpha}_2 = \max_{s,l}\boldsymbol{\alpha}_{s,2}^{(l)}$ and $\boldsymbol{\alpha}_3 = \max_{s,l}\boldsymbol{\alpha}_{s,3}^{(l)}$. Here $\widetilde{r}$ is given in Lemma 13. Besides, we have*

$$\left\|\frac{\partial \ell}{\partial \boldsymbol{X}_i^{(l)}(k)} - \frac{\partial \ell}{\partial \boldsymbol{X}_i^{(l)}(0)}\right\|_F \leq c_1 c\boldsymbol{\alpha}_3 c_{w0}^2 c_{x0}\rho k_c m\widetilde{r},$$

*where $c_1$ is a constant.*

See its proof in Appendix E.10.

**Lemma 18.** *Suppose Assumption 2 holds. Then with probability at least $1 - \delta/4$, it holds*

$$\begin{cases} \|\boldsymbol{W}^0\|_F \leq \sqrt{m}c_{w0}, \\ \|\boldsymbol{W}_s^{(l)}(0)\|_F \leq \sqrt{m}c_{w0} \ (\forall 0 \leq l \leq h-1, 0 \leq s \leq l-1), \\ \|\boldsymbol{W}_s(0)\|_F \leq \sqrt{m}c_{w0} \ (\forall 0 \leq s \leq h-1). \end{cases}$$

See its proof in Appendix E.11.

## C.3 Step 1 Linear Convergence of $\|\boldsymbol{y} - \boldsymbol{u}(k)\|_2^2$

Here we first present our results and then provides their proofs.

**Lemma 19.** *Suppose Assumptions 1, 2 and 3 hold. If $m$ and $\eta$ satisfy*

$$
\begin{cases}
m \geq \frac{c_1 \rho k_c^2 c_{w0}^2 \|\boldsymbol{y}-\boldsymbol{u}(0)\|_2^2}{\lambda^2 n} \left(1 + \boldsymbol{\alpha}_2 + 2\boldsymbol{\alpha}_3 \mu \sqrt{k_c} c_{w0}\right)^{2h}, \\
\eta \leq \frac{c_2 \lambda}{\sqrt{m} \mu^4 c_{w0}^4 c_{x0}^2 h^3 k_c^2 \left(1 + \boldsymbol{\alpha}_2 + 2\sqrt{k_c} c_{w0} \boldsymbol{\alpha}_3 \mu\right)^{4h}},
\end{cases}
$$

*where $c_1$ and $c_2$ are two constants and $\lambda$ is smallest eigenvalue of the Gram matrix $\boldsymbol{G}(t)$ ($t = 1, \cdots, k-1$), then with probability at least $1 - \delta/2$ we have*

$$
\|\boldsymbol{y} - \boldsymbol{u}(k)\|_2^2 \leq \left(1 - \frac{\eta\lambda}{2}\right) \|\boldsymbol{y} - \boldsymbol{u}(k-1)\|_2^2 \leq \left(1 - \frac{\eta\lambda}{2}\right)^k \|\boldsymbol{y} - \boldsymbol{u}(0)\|_2^2.
$$

See its proof in Appendix C.3.1.

**Lemma 20.** *Suppose Assumptions 1, 2 and 3 hold. If $m$ satisfy*

$$
m \geq \frac{c_3 \boldsymbol{\alpha}_3^2 \mu^2 k_c c_{x0}^2 c^2}{\lambda^2 n},
$$

*where $c_3$ is a constant, $c = \left(1 + \boldsymbol{\alpha}_2 + 2\boldsymbol{\alpha}_3 \mu \sqrt{k_c} c_{w0}\right)^h$, $\boldsymbol{\alpha}_2 = \max_{s,l} \boldsymbol{\alpha}_{s,2}^{(l)}$ and $\boldsymbol{\alpha}_3 = \max_{s,l} \boldsymbol{\alpha}_{s,3}^{(l)}$, then we have*

$$
\|\boldsymbol{G}(k) - \boldsymbol{G}(0)\|_2 \leq \frac{\eta \lambda_{\min}(\boldsymbol{G}(0))}{2},
$$

*where $\lambda_{\min}(\boldsymbol{G}(0))$ is the smallest eigenvalue of $\boldsymbol{G}(0)$.*

See its proof in Appendix C.3.2.

**Lemma 21.** *Suppose Assumptions 1, 2 and 3 hold. If $m$ and $\eta$ satisfy*

$$
\begin{cases}
m \geq \frac{c_m' c^2 \rho k_c^2 c_{w0}^2 \mu^2}{\lambda^2 n}, \\
\eta \leq \frac{c_\eta' \lambda}{\sqrt{m} \mu^4 h^3 k_c^2 c^4},
\end{cases}
$$

*where $c_m$ and $c_\eta$ are two constants, $c = \left(1 + \boldsymbol{\alpha}_2 + 2\boldsymbol{\alpha}_3 \mu \sqrt{k_c} c_{w0}\right)^h$, $\boldsymbol{\alpha}_2 = \max_{s,l} \boldsymbol{\alpha}_{s,2}^{(l)}$ and $\boldsymbol{\alpha}_3 = \max_{s,l} \boldsymbol{\alpha}_{s,3}^{(l)}$. Then with probability at least $1 - \delta$ we have*

$$
\|\boldsymbol{y} - \boldsymbol{u}(k)\|_2^2 \leq \left(1 - \frac{\eta \lambda_{\min}(\boldsymbol{G}(0))}{4}\right) \|\boldsymbol{y} - \boldsymbol{u}(k-1)\|_2^2 \leq \left(1 - \frac{\eta \lambda_{\min}(\boldsymbol{G}(0))}{4}\right)^k \|\boldsymbol{y} - \boldsymbol{u}(0)\|_2^2.
$$

See its proof in Appendix C.3.3.

### C.3.1 Proof of Lemma 19

*Proof.* Here we use mathematical induction to prove the result. For $k = 0$, the results in Theorem 19 holds. Then we assume for $j = 1, \cdots, k$, it holds

$$
\|\boldsymbol{y} - \boldsymbol{u}(j)\|_2^2 \leq \left(1 - \frac{\eta\lambda}{2}\right) \|\boldsymbol{y} - \boldsymbol{u}(j-1)\|_2^2 \leq \left(1 - \frac{\eta\lambda}{2}\right)^j \|\boldsymbol{y} - \boldsymbol{u}(0)\|_2^2 \quad (j = 1, \cdots, k).
$$

Then we need to prove $j = k + 1$ still holds. Our proof has four steps. In the first step, we establish the relation between $\|\boldsymbol{y} - \boldsymbol{u}(j)\|_2^2 \leq \|\boldsymbol{y} - \boldsymbol{u}(j)\|_2^2 + H_1 + H_2$. Then in the second, third and fourth steps, we bound the terms $H_1$, $H_2$, $H_3$ respectively. Finally, we combine results to obtain the desired result.

**Step 1. Establishing relation between $\|\boldsymbol{y} - \boldsymbol{u}(j)\|_2^2 \leq \|\boldsymbol{y} - \boldsymbol{u}(j)\|_2^2 + H_1 + H_2 + H_3$.**

According to the definition, we can obtain

$$
\begin{aligned}
\|\boldsymbol{y} - \boldsymbol{u}(k+1)\|_2^2 &= \|\boldsymbol{y} - \boldsymbol{u}(k) + \boldsymbol{u}(k) - \boldsymbol{u}(k+1)\|_2^2 \\
&= \|\boldsymbol{y} - \boldsymbol{u}(k)\|_2^2 + 2\langle \boldsymbol{y} - \boldsymbol{u}(k), \boldsymbol{u}(k) - \boldsymbol{u}(k+1)\rangle + \|\boldsymbol{u}(k) - \boldsymbol{u}(k+1)\|_2^2.
\end{aligned}
$$

Then for brevity, $\ell(\boldsymbol{\Omega})$ and $\ell_i(\boldsymbol{\Omega})$ respectively denote the losses when feeding the input $(\boldsymbol{X}, \boldsymbol{y})$ and $(\boldsymbol{X}_i, y_i)$. Then as introduced in Sec. B, we denote the gradient of $\ell(\boldsymbol{\Omega})$ with respect to all learnable parameters $\boldsymbol{\Omega}$ as

$$\nabla_{\boldsymbol{\Omega}} \ell(\boldsymbol{\Omega}) = \left[ \mathsf{vec}\left( \frac{\partial \ell}{\partial \boldsymbol{W}^{(0)}} \right); \left\{ \mathsf{vec}\left( \frac{\partial \ell}{\partial \boldsymbol{W}_s^{(l)}} \right) \right\}_{0 \le l \le h-1, 0 \le s \le l-1} ; \left\{ \mathsf{vec}\left( \frac{\partial \ell}{\partial \boldsymbol{W}_s} \right) \right\}_{0 \le s \le h-1} \right].$$

Based on the above definitions, when we use gradient descent algorithm to update the variables with learning rate $\eta$, we have

$$
\begin{aligned}
u_i(k+1) - u_i(k) =& u_i\left(\boldsymbol{\Omega}(k) - \eta \nabla_{\boldsymbol{\Omega}} F(\boldsymbol{\Omega}(k))\right) - u_i(\boldsymbol{\Omega}(k)) \\
=& - \int_{t=0}^{\eta} \langle \nabla_{\boldsymbol{\Omega}} F(\boldsymbol{\Omega}(k)), \nabla_{\boldsymbol{\Omega}} u_i\left(\boldsymbol{\Omega}(k) - s \nabla_{\boldsymbol{\Omega}} F(\boldsymbol{\Omega}(k))\right) \rangle \, dt = \boldsymbol{\Delta}_1^i(k) + \boldsymbol{\Delta}_2^i(k),
\end{aligned}
$$

where

$$
\begin{aligned}
\boldsymbol{\Delta}_1^i(k) &= - \int_{t=0}^{\eta} \langle \nabla_{\boldsymbol{\Omega}} F(\boldsymbol{\Omega}(k)), \nabla_{\boldsymbol{\Omega}} u_i\left(\boldsymbol{\Omega}(k)\right) \rangle \, dt \\
\boldsymbol{\Delta}_2^i(k) &= \int_{t=0}^{\eta} \langle \nabla_{\boldsymbol{\Omega}} F(\boldsymbol{\Omega}(k)), \nabla_{\boldsymbol{\Omega}} u_i\left(\boldsymbol{\Omega}(k)\right) - \nabla_{\boldsymbol{\Omega}} u_i\left(\boldsymbol{\Omega}(k) - t \nabla_{\boldsymbol{\Omega}} F(\boldsymbol{\Omega}(k))\right) \rangle \, dt.
\end{aligned}
$$

Then we define two important notations:

$$\boldsymbol{\Delta}_1(k) = [\boldsymbol{\Delta}_1^1(k); \boldsymbol{\Delta}_1^2(k); \cdots ; \boldsymbol{\Delta}_1^n(k)] \in \mathbb{R}^n, \qquad \boldsymbol{\Delta}_2(k) = [\boldsymbol{\Delta}_2^1(k); \boldsymbol{\Delta}_2^2(k); \cdots ; \boldsymbol{\Delta}_2^n(k)] \in \mathbb{R}^n.$$

In this way, we have $\boldsymbol{u}(k+1) - \boldsymbol{u}(k) = \boldsymbol{\Delta}_1(k) + \boldsymbol{\Delta}_2(k)$. Now we consider

$$
\begin{aligned}
\boldsymbol{\Delta}_1^i(k) &= - \int_{s=0}^{\eta} \langle \nabla_{\boldsymbol{\Omega}} F(\boldsymbol{\Omega}(k)), \nabla_{\boldsymbol{\Omega}} u_i\left(\boldsymbol{\Omega}(k)\right) \rangle \\
&= - \eta \langle \nabla_{\boldsymbol{\Omega}} F(\boldsymbol{\Omega}(k)), \nabla_{\boldsymbol{\Omega}} u_i\left(\boldsymbol{\Omega}(k)\right) \rangle \\
&= - \frac{\eta}{n} \sum_{j=1}^{n} (y_j - u_j) \langle \nabla_{\boldsymbol{\Omega}} u_j\left(\boldsymbol{\Omega}(k)\right), \nabla_{\boldsymbol{\Omega}} u_i\left(\boldsymbol{\Omega}(k)\right) \rangle \\
&= - \frac{\eta}{n} \sum_{j=1}^{n} (y_j - u_j) \sum_{t=1}^{(h+1)(\frac{h}{2}+1)} \langle \nabla_{\boldsymbol{\Omega}_t} u_j\left(\boldsymbol{\Omega}(k)\right), \nabla_{\boldsymbol{\Omega}_t} u_i\left(\boldsymbol{\Omega}(k)\right) \rangle.
\end{aligned}
$$

Let $\boldsymbol{G}_{ij}^t(k) = \langle \nabla_{\boldsymbol{\Omega}_t} u_j\left(\boldsymbol{\Omega}(k)\right), \nabla_{\boldsymbol{\Omega}_t} u_i\left(\boldsymbol{\Omega}(k)\right) \rangle$. In this way, we have $\boldsymbol{G}(k) = \sum_{t=1}^{(h+1)(\frac{h}{2}+1)} \boldsymbol{G}^t$. Then $\boldsymbol{\Delta}_1(k)$ can be formulated as follows:

$$\boldsymbol{\Delta}_1(k) = -\eta \boldsymbol{G}(k)(\boldsymbol{u}(k) - \boldsymbol{y}).$$

In this way, we can compute

$$
\begin{aligned}
2\langle \boldsymbol{y} - \boldsymbol{u}(k), \boldsymbol{u}(k) - \boldsymbol{u}(k+1) \rangle =& - 2\langle \boldsymbol{y} - \boldsymbol{u}(k), \boldsymbol{\Delta}_1(k) + \boldsymbol{\Delta}_2(k) \rangle \\
=& - 2\eta (\boldsymbol{u}(k) - \boldsymbol{y})^{\top} \boldsymbol{G}(k)(\boldsymbol{u}(k) - \boldsymbol{y}) - 2\langle \boldsymbol{y} - \boldsymbol{u}(k), \boldsymbol{\Delta}_2(k) \rangle
\end{aligned}
$$

Therefore, we can decompose $\|\boldsymbol{y} - \boldsymbol{u}(k+1)\|_2^2$ into

$$
\begin{aligned}
&\|\boldsymbol{y} - \boldsymbol{u}(k+1)\|_2^2 \\
=&\|\boldsymbol{y} - \boldsymbol{u}(k)\|_2^2 + 2\langle \boldsymbol{y} - \boldsymbol{u}(k), \boldsymbol{u}(k) - \boldsymbol{u}(k+1) \rangle + \|\boldsymbol{u}(k) - \boldsymbol{u}(k+1)\|_2^2 \\
=&\|\boldsymbol{y} - \boldsymbol{u}(k)\|_2^2 - 2\eta (\boldsymbol{u}(k) - \boldsymbol{y})^{\top} \boldsymbol{G}(k)(\boldsymbol{u}(k) - \boldsymbol{y}) - 2\langle \boldsymbol{y} - \boldsymbol{u}(k), \boldsymbol{\Delta}_2(k) \rangle + \|\boldsymbol{u}(k) - \boldsymbol{u}(k+1)\|_2^2 \\
\le&\|\boldsymbol{y} - \boldsymbol{u}(k)\|_2^2 - 2\eta (\boldsymbol{u}(k) - \boldsymbol{y})^{\top} \boldsymbol{G}(k)(\boldsymbol{u}(k) - \boldsymbol{y}) + 2\|\boldsymbol{y} - \boldsymbol{u}(k)\|_2 \|\boldsymbol{\Delta}_2(k)\|_2 + \|\boldsymbol{u}(k) - \boldsymbol{u}(k+1)\|_2^2.
\end{aligned}
\tag{9}
$$

Let $H_1 = -2\eta (\boldsymbol{u}(k) - \boldsymbol{y})^{\top} \boldsymbol{G}(k)(\boldsymbol{u}(k) - \boldsymbol{y})$, $H_2 = 2\|\boldsymbol{y} - \boldsymbol{u}(k)\|_2 \|\boldsymbol{\Delta}_2(k)\|_2$ and $H_3 = \|\boldsymbol{u}(k) - \boldsymbol{u}(k+1)\|_2^2$. The remaining task is to upper bound $H_1 \sim H_3$.

**Step 2. Bound of $H_1$.**

To bound $H_1$, we can easily to bound it as follows:
$$H_1 = -2\eta(\boldsymbol{u}(k) - \boldsymbol{y})^\top \boldsymbol{G}(k)(\boldsymbol{u}(k) - \boldsymbol{y}) \leq -2\eta\lambda\|\boldsymbol{u}(k) - \boldsymbol{y}\|_2^2,$$
where $\lambda = \min_k \lambda_{\min}(\boldsymbol{G}(k))$.

**Step 3. Bound of $H_2$.**

In this step, we aim to bound $H_2 = 2\|\boldsymbol{y} - \boldsymbol{u}(k)\|_2 \|\boldsymbol{\Delta}_2(k)\|_2$ by bounding $\|\boldsymbol{\Delta}_2^i(k)\|_2$. According to the definition, we have
$$\boldsymbol{\Delta}_2^i(k) = \int_{t=0}^{\eta} \langle \nabla_{\boldsymbol{\Omega}} F(\boldsymbol{\Omega}(k)), \nabla_{\boldsymbol{\Omega}} u_i(\boldsymbol{\Omega}(k)) - \nabla_{\boldsymbol{\Omega}} u_i(\boldsymbol{\Omega}(k) - s\nabla_{\boldsymbol{\Omega}} F(\boldsymbol{\Omega}(k))) \rangle \, dt$$
$$\leq \eta \max_{t \in [0,\eta]} \|\nabla_{\boldsymbol{\Omega}} F(\boldsymbol{\Omega}(k))\|_F \|\nabla_{\boldsymbol{\Omega}} u_i(\boldsymbol{\Omega}(k)) - \nabla_{\boldsymbol{\Omega}} u_i(\boldsymbol{\Omega}(k) - t\nabla_{\boldsymbol{\Omega}} F(\boldsymbol{\Omega}(k)))\|_F.$$

In this way, we need to bound $\max_{t \in [0,\eta]} \|\nabla_{\boldsymbol{\Omega}} u_i(\boldsymbol{\Omega}(k)) - \nabla_{\boldsymbol{\Omega}} u_i(\boldsymbol{\Omega}(k) - t\nabla_{\boldsymbol{\Omega}} F(\boldsymbol{\Omega}(k)))\|_F$ and $\|\nabla_{\boldsymbol{\Omega}} F(\boldsymbol{\Omega}(k))\|_F$.

**Step 3.1 Bound of $\|\nabla_{\boldsymbol{\Omega}} F(\boldsymbol{\Omega}(k))\|_F$ in $H_2$.** According to the definition, we have
$$\|\nabla_{\boldsymbol{\Omega}} F(\boldsymbol{\Omega}(k))\|_F \leq \sum_{t=1}^{(h+1)(h/2+1)} \|\nabla_{\boldsymbol{\Omega}_t} F(\boldsymbol{\Omega}(k))\|_F$$
$$= \left\|\frac{\partial F(\Omega)}{\partial \boldsymbol{W}^{(0)}(k)}\right\|_F + \sum_{l=0}^{h-1}\sum_{s=0}^{l-1} \left\|\frac{\partial F(\Omega)}{\partial \boldsymbol{W}_s^{(l)}(k)}\right\|_F + \sum_{s=0}^{h-1} \left\|\frac{\partial F(\Omega)}{\partial \boldsymbol{W}_s(k)}\right\|_F$$
$$\overset{①}{\leq} \left(h + 2c\mu c_{w0}\sqrt{k_c}\left(1 + \sum_{l=0}^{h-1}\sum_{s=0}^{l-1}\boldsymbol{\alpha}_{s,3}^{(l)}\right)\right)\frac{2c_{x0}}{\sqrt{n}}\|\boldsymbol{u}(t) - \boldsymbol{y}\|_2,$$

where ① holds by using Lemma 13 with $c = \left(1 + \boldsymbol{\alpha}_2 + 2\boldsymbol{\alpha}_3\mu\sqrt{k_c}c_{w0}\right)^l$, $\boldsymbol{\alpha}_2 = \max_{s,l}\boldsymbol{\alpha}_{s,2}^{(l)}$ and $\boldsymbol{\alpha}_3 = \max_{s,l}\boldsymbol{\alpha}_{s,3}^{(l)}$ since Lemma 13 proves
$$\left\|\frac{\partial F(\Omega)}{\partial \boldsymbol{W}^{(0)}(t)}\right\|_F \leq \frac{4c\mu c_{x0}c_{w0}\sqrt{k_c}}{\sqrt{n}}\|\boldsymbol{u}(t) - \boldsymbol{y}\|_2, \quad \left\|\frac{\partial F(\Omega)}{\partial \boldsymbol{W}_s^{(l)}(t)}\right\|_F \leq \frac{4c\boldsymbol{\alpha}_{s,3}^{(l)}\mu c_{x0}c_{w0}\sqrt{k_c}}{\sqrt{n}}\|\boldsymbol{u}(t) - \boldsymbol{y}\|_2,$$
$$\left\|\frac{\partial F(\Omega)}{\partial \boldsymbol{W}_s(t)}\right\|_F \leq \frac{2c_{x0}}{\sqrt{n}}\|\boldsymbol{u}(t) - \boldsymbol{y}\|_2,$$

**Step 3.2 Bound of $\|\nabla_{\boldsymbol{\Omega}} u_i(\boldsymbol{\Omega}(k)) - \nabla_{\boldsymbol{\Omega}} u_i(\boldsymbol{\Omega}(k) - t\nabla_{\boldsymbol{\Omega}} F(\boldsymbol{\Omega}(k)))\|_F$ in $H_2$.**

For brevity, let $\boldsymbol{\Omega}(k,t) = \boldsymbol{\Omega}(k) - t\nabla_{\boldsymbol{\Omega}} F(\boldsymbol{\Omega}(k))$. In this way, we can bound
$$\|\nabla_{\boldsymbol{\Omega}} u_i(\boldsymbol{\Omega}(k)) - \nabla_{\boldsymbol{\Omega}} u_i(\boldsymbol{\Omega}(k,t)))\|_F \leq \sum_{o=1}^{(h+1)(h/2+1)} \|\nabla_{\boldsymbol{\Omega}_o} u_i(\boldsymbol{\Omega}(k)) - \nabla_{\boldsymbol{\Omega}_o} u_i(\boldsymbol{\Omega}(k,s))\|_F$$
$$= \left\|\frac{\partial u_i}{\partial \boldsymbol{W}^{(0)}(k)} - \frac{\partial u_i}{\partial \boldsymbol{W}^{(0)}(k,t)}\right\|_F + \sum_{l=0}^{h-1}\sum_{s=0}^{l-1}\left\|\frac{\partial u_i}{\partial \boldsymbol{W}_s^{(l)}(k)} - \frac{\partial u_i}{\partial \boldsymbol{W}_s^{(l)}(k,t)}\right\|_F + \sum_{s=0}^{h-1}\left\|\frac{\partial u_i}{\partial \boldsymbol{W}_s(k)} - \frac{\partial u_i}{\partial \boldsymbol{W}_s(k,t)}\right\|_F.$$

In the following, we will bound each term. We first look at $\left\|\frac{\partial u_i}{\partial \boldsymbol{W}_s(k)} - \frac{\partial u_i}{\partial \boldsymbol{W}_s(k,t)}\right\|_F$. By using Lemma 8, we have $\frac{\partial u_i}{\partial \boldsymbol{W}_s(k)} = \boldsymbol{X}_i^{(l)}(k)$. Therefore, we can obtain
$$\left\|\frac{\partial u_i}{\partial \boldsymbol{W}_s(k)} - \frac{\partial u_i}{\partial \boldsymbol{W}_s(k,t)}\right\|_F = \left\|\boldsymbol{X}_i^{(l)}(k) - \boldsymbol{X}_i^{(l)}(k,t)\right\|_F = t\left\|\frac{\partial F(\boldsymbol{\Omega})}{\partial \boldsymbol{X}_i^{(l)}(k)}\right\|_F$$
$$\leq t\frac{1}{n}\sum_{i=1}^{n}\left\|\frac{\partial \ell_i}{\partial \boldsymbol{X}_i^{(l)}(k)}\right\|_F \overset{①}{\leq} \eta\left(1 + \boldsymbol{\alpha}_2 + 2\boldsymbol{\alpha}_3\mu\sqrt{k_c}c_{w0}\right)^l c_y c_u, \tag{10}$$

where ① holds since in Lemma 13, we have show
$$\max\left(\|\boldsymbol{W}^{(0)}(t) - \boldsymbol{W}^{(0)}(0)\|_F, \|\boldsymbol{W}_s^{(l)}(t) - \boldsymbol{W}_s^{(l)}(0)\|_F, \|\boldsymbol{W}_s(t) - \boldsymbol{W}_s(0)\|_F\right) \leq \sqrt{m}\tilde{r} \leq \sqrt{m}c_{w0}, \tag{11}$$

which allows us to use Lemma 12 which shows
$$\frac{1}{n}\sum_{i=1}^{n}\left\|\frac{\partial \ell_i}{\partial \boldsymbol{X}_i^{(l)}(k)}\right\|_F \leq \left(1 + \boldsymbol{\alpha}_2 + \boldsymbol{\alpha}_3\mu\sqrt{k_c}(\tilde{r} + c_{w0})\right)^l c_y c_u \leq \left(1 + \boldsymbol{\alpha}_2 + 2\boldsymbol{\alpha}_3\mu\sqrt{k_c}c_{w0}\right)^l c_y c_u, \tag{12}$$

where parameters $\frac{1}{\sqrt{n}}\|\boldsymbol{u}(t)-\boldsymbol{y}\|_2 = c_y$ and $\|\boldsymbol{W}_h(t)\|_F \le c_u$, $\boldsymbol{\alpha}_2 = \max_{s,l}\boldsymbol{\alpha}_{s,2}^{(l)}$ and $\boldsymbol{\alpha}_3 = \max_{s,l}\boldsymbol{\alpha}_{s,3}^{(l)}$. Moreover, from Lemma 13, we have $\|\boldsymbol{W}_h(t)\|_F \le \|\boldsymbol{W}_h(t)-\boldsymbol{W}_h(0)\|_F + \|\boldsymbol{W}_h(0)\|_F \le 2\sqrt{m}c_{w0}$. In this way, we have

$$\sum_{s=1}^{h}\left\|\frac{\partial u_i}{\partial \boldsymbol{W}_s(k)} - \frac{\partial u_i}{\partial \boldsymbol{W}_s(k,t)}\right\|_F \le \eta h\left(1+\boldsymbol{\alpha}_2+2\boldsymbol{\alpha}_3\mu\sqrt{k_c}c_{w0}\right)^l\sqrt{m}c_{w0}\frac{1}{\sqrt{n}}\|\boldsymbol{u}(t)-\boldsymbol{y}\|_2$$

$$\le\eta h\left(1+\boldsymbol{\alpha}_2+2\boldsymbol{\alpha}_3\mu\sqrt{k_c}c_{w0}\right)^l\sqrt{m}c_{w0}\frac{1}{\sqrt{n}}\left(1-\frac{\eta\lambda}{2}\right)^{t/2}\|\boldsymbol{u}(0)-\boldsymbol{y}\|_2 = \eta c_1,$$

where $c_1 = h\left(1+\boldsymbol{\alpha}_2+2\boldsymbol{\alpha}_3\mu\sqrt{k_c}c_{w0}\right)^l\sqrt{m}c_{w0}\frac{1}{\sqrt{n}}\left(1-\frac{\eta\lambda}{2}\right)^{t/2}\|\boldsymbol{u}(0)-\boldsymbol{y}\|_F$ is a constant.

Then we consider $\left\|\frac{\partial u_i}{\partial \boldsymbol{W}_s^{(l)}(k)} - \frac{\partial u_i}{\partial \boldsymbol{W}_s^{(l)}(k,t)}\right\|_F$ as follows:

$$\left\|\frac{\partial u_i}{\partial \boldsymbol{W}_s^{(l)}(k)} - \frac{\partial u_i}{\partial \boldsymbol{W}_s^{(l)}(k,t)}\right\|_F = \boldsymbol{\alpha}_{s,3}^{(l)}\tau\left[\left\|\Phi(\boldsymbol{X}_i^{(s)}(k))\left(\sigma'\left(\boldsymbol{W}_s^{(l)}(k)\Phi(\boldsymbol{X}_i^{(s)}(k))\right)\odot\frac{\partial u_i}{\partial \boldsymbol{X}_i^{(l)}(k)}\right)^{\top}\right.\right.$$

$$\left.\left.-\Phi(\boldsymbol{X}_i^{(s)}(k,t))\left(\sigma'\left(\boldsymbol{W}_s^{(l)}(k,t)\Phi(\boldsymbol{X}_i^{(s)}(k,t))\right)\odot\frac{\partial u_i}{\partial \boldsymbol{X}_i^{(l)}(k,t)}\right)^{\top}\right\|_F\right]$$

$$\overset{\textcircled{1}}{\le}\boldsymbol{\alpha}_{s,3}^{(l)}\tau\frac{a_1 a_2(b_1+b_2)}{\max(a_1,a_2)},$$

where $\textcircled{1}$ uses Lemma 2. For parameters $a_1, a_2, b_1, b_2$ satisfies

$$a_1 = \max\left(\left\|\Phi(\boldsymbol{X}_i^{(s)}(k))\right\|_2, \left\|\Phi(\boldsymbol{X}_i^{(s)}(k,t))\right\|_2\right) \le \sqrt{k_c}\max\left(\left\|\boldsymbol{X}_i^{(s)}(k)\right\|_2, \left\|\boldsymbol{X}_i^{(s)}(k,t)\right\|_2\right),$$

$$a_2 = \max\left(\left\|\sigma'\left(\boldsymbol{W}_s^{(l)}(k)\Phi(\boldsymbol{X}_i^{(s)}(k))\right)\odot\frac{\partial u_i}{\partial \boldsymbol{X}_i^{(l)}(k)}\right\|_2, \left\|\sigma'\left(\boldsymbol{W}_s^{(l)}(k,t)\Phi(\boldsymbol{X}_i^{(s)}(k,t))\right)\odot\frac{\partial u_i}{\partial \boldsymbol{X}_i^{(l)}(k,t)}\right\|_2\right),$$

$$b_1 = \left\|\Phi(\boldsymbol{X}_i^{(s)}(k)) - \Phi(\boldsymbol{X}_i^{(s)}(k,t))\right\|_2 \le \sqrt{k_c}\left\|\boldsymbol{X}_i^{(s)}(k) - \boldsymbol{X}_i^{(s)}(k,t)\right\|_2,$$

$$b_2 = \left\|\sigma'\left(\boldsymbol{W}_s^{(l)}(k)\Phi(\boldsymbol{X}_i^{(s)}(k))\right)\odot\frac{\partial u_i}{\partial \boldsymbol{X}_i^{(l)}(k)} - \sigma'\left(\boldsymbol{W}_s^{(l)}(k,t)\Phi(\boldsymbol{X}_i^{(s)}(k,t))\right)\odot\frac{\partial u_i}{\partial \boldsymbol{X}_i^{(l)}(k,t)}\right\|_2.$$

In Lemma 10, we show that when Eqn. (10) holds which is proven in Lemma 13, then $\|\boldsymbol{X}_i^{(l)}(0)\|_F \le c_{x0}$. Under Eqn. (10), Lemma 11 shows

$$\|\boldsymbol{X}_i^{(l)}(k) - \boldsymbol{X}_i^{(l)}(0)\|_F \le \left(1+\boldsymbol{\alpha}_2+2\boldsymbol{\alpha}_3\mu\sqrt{k_c}c_{w0}\right)^l\mu\sqrt{k_c}\widetilde{r}\overset{\textcircled{1}}{\le}c_{x0}, \tag{13}$$

where $\textcircled{1}$ holds since in Lemma 13, we set $m = \mathcal{O}\left(\frac{\rho k_c^2 c_{w0}^2\|\boldsymbol{y}-\boldsymbol{u}(0)\|_2^2}{\lambda^2 n}\left(1+\boldsymbol{\alpha}_2+2\boldsymbol{\alpha}_3\mu\sqrt{k_c}c_{w0}\right)^{2h}\right)$ such that

$$\widetilde{r} = \frac{8c_{x0}\|\boldsymbol{y}-\boldsymbol{u}(0)\|_2}{\lambda\sqrt{mn}}\max\left(1, 2\left(1+\boldsymbol{\alpha}_2+2\boldsymbol{\alpha}_3\mu\sqrt{k_c}c_{w0}\right)^l\boldsymbol{\alpha}_{s,3}^{(l)}\mu\sqrt{k_c}c_{w0}\right)$$

$$\le\frac{c_{x0}}{\left(1+\boldsymbol{\alpha}_2+2\boldsymbol{\alpha}_3\mu\sqrt{k_c}c_{w0}\right)^l\mu\sqrt{k_c}}.$$

By using Lemma 11 and Lemma 10, we have

$$\|\boldsymbol{X}^{(s)}(t)\| \le \|\boldsymbol{X}_i^{(l)}(k) - \boldsymbol{X}_i^{(l)}(0)\|_F + \|\boldsymbol{X}_i^{(l)}(0)\|_F \le 2c_{x0}. \tag{14}$$

Then by using Eqn. (12) we upper bound $\left\|\boldsymbol{X}_i^{(s)}(k,t)\right\|_2$ as follows:

$$\left\|\boldsymbol{X}_i^{(s)}(k,t)\right\|_2 \le \left\|\boldsymbol{X}_i^{(s)}(k) - t\frac{\partial F(\boldsymbol{\Omega})}{\partial \boldsymbol{X}_i^{(s)}(k)}\right\|_2 \le \left\|\boldsymbol{X}_i^{(s)}(k)\right\|_2 + t\frac{1}{n}\sum_{i=1}^{n}\left\|\frac{\partial \ell_i}{\partial \boldsymbol{X}_i^{(s)}(k)}\right\|_2$$

$$\le 2c_{x0} + \eta\left(1+\boldsymbol{\alpha}_2+2\boldsymbol{\alpha}_3\mu\sqrt{k_c}c_{w0}\right)^l\sqrt{m}c_{w0}\frac{1}{\sqrt{n}}\|\boldsymbol{u}(t)-\boldsymbol{y}\|_F \le c_2,$$

where $c_2 = 2c_{x0} + \eta \left(1 + \boldsymbol{\alpha}_2 + 2\boldsymbol{\alpha}_3 \mu \sqrt{k_c} c_{w0}\right)^l \sqrt{m} c_{w0} \frac{1}{\sqrt{n}} \left(1 - \frac{\eta\lambda}{2}\right)^{t/2} \|\boldsymbol{u}(0) - \boldsymbol{y}\|_F$ is a constant. In this way, we can upper bound

$$a_1 \leq \sqrt{k_c} \max\left(2c_{w0}, c_2\right), \qquad b_1 \overset{\text{①}}{\leq} \frac{\sqrt{k_c} c_1 \eta}{h},$$

where ① uses the results in Eqn. (10). Now we try to bound $a_2$ and $b_2$ as follows:

$$a_2 = \max\left(\left\|\sigma'\left(\boldsymbol{W}_s^{(l)}(k)\Phi(\boldsymbol{X}_i^{(s)}(k))\right) \odot \frac{\partial u_i}{\partial \boldsymbol{X}_i^{(l)}(k)}\right\|_2, \left\|\sigma'\left(\boldsymbol{W}_s^{(l)}(k,t)\Phi(\boldsymbol{X}_i^{(s)}(k,t))\right) \odot \frac{\partial u_i}{\partial \boldsymbol{X}_i^{(l)}(k,t)}\right\|_2\right)$$

$$\leq \mu \max\left(\left\|\frac{\partial u_i}{\partial \boldsymbol{X}_i^{(l)}(k)}\right\|_2, \left\|\frac{\partial u_i}{\partial \boldsymbol{X}_i^{(l)}(k,t)}\right\|_2\right) \overset{\text{①}}{\leq} \mu(1+L)c_1^2\eta^2,$$

where ① uses $\left\|\frac{\partial u_i}{\partial \boldsymbol{X}_i^{(l)}(k,t)}\right\|_2 \leq \left\|\frac{\partial u_i}{\partial \boldsymbol{X}_i^{(l)}(k,t)}\right\|_F \leq \left\|\frac{\partial u_i}{\partial \boldsymbol{X}_i^{(l)}(k)}\right\|_F + L\|\boldsymbol{X}_i^{(l)}(k,t) - \boldsymbol{X}_i^{(l)}(k)\|_F^2 \overset{\text{②}}{\leq} (1+L)c_1^2\eta^2$
where $L$ is the Lipschitz constant of $\frac{\partial u_i}{\partial \boldsymbol{X}^{(l)}}$. In ② we use the results in Eqn. (14). Since $\sigma$ is $\rho$-smooth and $u$ is $h$-layered, by computing, we know $L$ is at the order of $\mathcal{O}\left(\beta^h\right)$ and is a constant. For $b_2$ we can bound it as follows:

$$b_2 \leq \mu \left\|\frac{\partial u_i}{\partial \boldsymbol{X}_i^{(l)}(k)} - \frac{\partial u_i}{\partial \boldsymbol{X}_i^{(l)}(k,t)}\right\|_2 \leq 2\mu(1+L)c_1^2\eta^2.$$

Therefore, we can bound

$$\sum_{l=1}^{h}\sum_{s=0}^{l-1}\left\|\frac{\partial u_i}{\partial \boldsymbol{W}_s^{(l)}(k)} - \frac{\partial u_i}{\partial \boldsymbol{W}_s^{(l)}(k,t)}\right\|_F \leq \tau\frac{a_1 a_2(b_1+b_2)}{\max(a_1,a_2)}\sum_{l=1}^{h}\sum_{s=0}^{l-1}\boldsymbol{\alpha}_{s,3}^{(l)} = c_3\eta,$$

where $\boldsymbol{\alpha}_3 = \max \boldsymbol{\alpha}_{s,3}^{(l)}$ and $c_3 = \frac{\tau\sqrt{k_c}\max(2c_{w0},c_2)\mu(1+L)c_1^2\eta^2}{\max(\sqrt{k_c}\max(2c_{w0},c_2),\mu(1+L)c_1^2\eta^2,)}\left(\frac{\sqrt{k_c}c_1}{h} + 2\mu(1+L)c_1^2\eta\right)$ is a constant. By using the same method, we can bound

$$\left\|\frac{\partial u_i}{\partial \boldsymbol{W}^{(0)}(k)} - \frac{\partial u_i}{\partial \boldsymbol{W}^{(0)}(k,t)}\right\|_F$$

$$= \tau\left\|\Phi(\boldsymbol{X}_i)\left(\sigma'\left(\boldsymbol{W}^{(0)}(k)\Phi(\boldsymbol{X}_i)\right) \odot \frac{\partial u_i}{\partial \boldsymbol{X}_i^{(0)}(k)}\right)^\top - \Phi(\boldsymbol{X}_i)\left(\sigma'\left(\boldsymbol{W}^{(0)}(k,t)\Phi(\boldsymbol{X}_i)\right) \odot \frac{\partial u_i}{\partial \boldsymbol{X}_i^{(0)}(k,t)}\right)^\top\right\|_F$$

$$\overset{\text{①}}{\leq} \tau\sqrt{k_c}\left\|\frac{\partial u_i}{\partial \boldsymbol{X}_i^{(0)}(k)} - \frac{\partial u_i}{\partial \boldsymbol{X}_i^{(0)}(k,t)}\right\|_F \leq 2\mu(1+L)c_1^2\eta^2 = c_4\eta,$$

where ① uses $\|\Phi(\boldsymbol{X}_i)\|_F \leq \sqrt{k_c}\|\boldsymbol{X}_i\|_F \leq \sqrt{k_c}$ and $\sigma$ is $\mu$-Lipschitz, and $c_4 = 2\mu(1+L)c_1^2\eta$. By combing the above results, we can further conclude

$$\|\nabla_{\boldsymbol{\Omega}} u_i\left(\boldsymbol{\Omega}(k)\right) - \nabla_{\boldsymbol{\Omega}} u_i\left(\boldsymbol{\Omega}(k,t)\right)\|_F \leq (c_1 + c_3 + c_4)\eta = c_5\eta,$$

which further gives

$$\boldsymbol{\Delta}_2^i(k) \leq \eta \max_{t\in[0,\eta]}\|\nabla_{\boldsymbol{\Omega}} F(\boldsymbol{\Omega}(k))\|_F \|\nabla_{\boldsymbol{\Omega}} u_i\left(\boldsymbol{\Omega}(k)\right) - \nabla_{\boldsymbol{\Omega}} u_i\left(\boldsymbol{\Omega}(k) - t\nabla_{\boldsymbol{\Omega}} F(\boldsymbol{\Omega}(k))\right)\|_F.$$

$$\leq \eta^2 c_5 \left(h + 2c\mu c_{w0}\sqrt{k_c}\left(1 + \sum_{l=1}^{h}\sum_{s=0}^{l-1}\boldsymbol{\alpha}_{s,3}^{(l)}\right)\right)\frac{2c_{x0}}{\sqrt{n}}\|\boldsymbol{u}(t) - \boldsymbol{y}\|_F = \hat{c}\eta^2 \|\boldsymbol{u}(t) - \boldsymbol{y}\|_F,$$

where $\hat{c} = c_5 \left(h + 2c\mu c_{w0}\sqrt{k_c}\left(1 + \sum_{l=1}^{h}\sum_{s=0}^{l-1}\boldsymbol{\alpha}_{s,3}^{(l)}\right)\right)\frac{2c_{x0}}{\sqrt{n}}$. Therefore we have

**Step 3.3 Upper bound** $H_2 = 2\|\boldsymbol{y} - \boldsymbol{u}(k)\|_2\|\boldsymbol{\Delta}_2(k)\|_2$. By combining the above results, we can bound

$$H_2 = 2\|\boldsymbol{y} - \boldsymbol{u}(k)\|_2\|\boldsymbol{\Delta}_2(k)\|_2 \leq \hat{c}\eta^2 \|\boldsymbol{u}(t) - \boldsymbol{y}\|_2^2,$$

where $\hat{c} = \mathcal{O}\left(\frac{\mu c_{x0} c_{w0}^2 \sqrt{k_c m} h^3 (1+\boldsymbol{\alpha}_2+2\boldsymbol{\alpha}_3\mu\sqrt{k_c}c_{w0})^h}{n}\right)$.

**Step 4. Upper bound $H_3 = \|\boldsymbol{u}(k) - \boldsymbol{u}(k+1)\|_2^2$.**

$$\|\boldsymbol{u}(k) - \boldsymbol{u}(k+1)\|_2^2 = \sum_{i=1}^{n} \left( \sum_{s=0}^{h-1} \left( \langle \boldsymbol{W}_s(k), \boldsymbol{X}_i^{(l)}(k) \rangle - \langle \boldsymbol{W}_s(k+1), \boldsymbol{X}_i^{(l)}(k+1) \rangle \right) \right)^2$$

$$\leq \sqrt{h} \sum_{i=1}^{n} \sum_{s=0}^{h-1} \left( \langle \boldsymbol{W}_s(k), \boldsymbol{X}_i^{(l)}(k) \rangle - \langle \boldsymbol{W}_s(k+1), \boldsymbol{X}_i^{(l)}(k+1) \rangle \right)^2.$$

Now we consider each term:

$$\left( \langle \boldsymbol{W}_s(k), \boldsymbol{X}_i^{(l)}(k) \rangle - \langle \boldsymbol{W}_s(k+1), \boldsymbol{X}_i^{(l)}(k+1) \rangle \right)^2$$

$$= \left( \langle \boldsymbol{W}_s(k) - \boldsymbol{W}_s(k+1), \boldsymbol{X}_i^{(l)}(k+1) \rangle + \langle \boldsymbol{W}_s(k), \boldsymbol{X}_i^{(l)}(k) - \boldsymbol{X}_i^{(l)}(k+1) \rangle \right)^2$$

$$\leq 2\|\boldsymbol{W}_s(k) - \boldsymbol{W}_s(k+1)\|_F^2 \|\boldsymbol{X}_i^{(l)}(k+1)\|_F^2 + 2\|\boldsymbol{W}_s(k)\|_F^2 \|\boldsymbol{X}_i^{(l)}(k) - \boldsymbol{X}_i^{(l)}(k+1)\|_F^2$$

$$\overset{①}{\leq} 8c_{x0}^2 \|\boldsymbol{W}_s(k) - \boldsymbol{W}_s(k+1)\|_F^2 + 8mc_{w0}^2 \|\boldsymbol{X}_i^{(l)}(k) - \boldsymbol{X}_i^{(l)}(k+1)\|_F^2$$

$$\overset{②}{\leq} \frac{32\eta^2 c_{x0}^2}{n} \left[ c_{x0}^2 + 4c^2\mu^4 c_{w0}^4 k_c^2 \left( 1 + \boldsymbol{\alpha}_2 + 2\sqrt{k_c}c_{w0}\boldsymbol{\alpha}_3\mu \right)^{2l} \left( 1 + \frac{2(\boldsymbol{\alpha}_3)^2 c_{x0}}{(\boldsymbol{\alpha}_2 + 2\sqrt{k_c}c_{w0}\boldsymbol{\alpha}_3\mu)\sqrt{n}} \right)^2 \right]$$

$$\cdot \|\boldsymbol{u}(k) - \boldsymbol{y}\|_2^2,$$

where ① uses $\|\boldsymbol{X}_i^{(l)}(k+1)\|_F^2 \leq 4c_{x0}^2$ in Eqn. (14), and the results in Eqn. (11) that $\|\boldsymbol{W}_s(k)\|_F \leq \|\boldsymbol{W}_s(k) - \boldsymbol{W}_s(0)\|_F + \|\boldsymbol{W}_s(0)\|_F \leq 2\sqrt{m}c_{w0}$; ② holds since (1) in Lemma 13 we have $\|\boldsymbol{W}_s(t+1) - \boldsymbol{W}_s(t)\|_F = \eta \left\| \frac{\partial F(\Omega)}{\partial \boldsymbol{W}_s(t)} \right\|_F \leq \frac{2\eta c_{x0}}{\sqrt{n}} \|\boldsymbol{u}(t) - \boldsymbol{y}\|_2$ where $c = \left( 1 + \boldsymbol{\alpha}_2 + 2\boldsymbol{\alpha}_3\mu\sqrt{k_c}c_{w0} \right)^l$ with $\boldsymbol{\alpha}_2 = \max_{s,l} \boldsymbol{\alpha}_{s,2}^{(l)}$ and $\boldsymbol{\alpha}_3 = \max_{s,l} \boldsymbol{\alpha}_{s,3}^{(l)}$, and (2) in Lemma 14 we have

$$\left\| \boldsymbol{X}^{(l)}(k+1) - \boldsymbol{X}^{(l)}(k) \right\|_F$$

$$\leq \left( 1 + \boldsymbol{\alpha}_2 + 2\sqrt{k_c}c_{w0}\boldsymbol{\alpha}_3\mu \right)^l \left( 1 + \frac{2(\boldsymbol{\alpha}_3)^2 c_{x0}}{(\boldsymbol{\alpha}_2 + 2\sqrt{k_c}c_{w0}\boldsymbol{\alpha}_3\mu)\sqrt{n}} \right) \frac{4c\tau\eta\mu^2 c_{x0}c_{w0}k_c}{\sqrt{n}} \|\boldsymbol{u}(k) - \boldsymbol{y}\|_2.$$

In this way, we can conclude

$$\|\boldsymbol{u}(k) - \boldsymbol{u}(k+1)\|_2^2 \leq \eta^2 \tilde{c} \|\boldsymbol{u}(k) - \boldsymbol{y}\|_2^2,$$

where $\tilde{c} = 32c_{x0}^2 h^{1.5} \left[ c_{x0}^2 + 4c^2\mu^4 c_{w0}^4 k_c^2 \left( 1 + \boldsymbol{\alpha}_2 + 2\sqrt{k_c}c_{w0}\boldsymbol{\alpha}_3\mu \right)^{2l} \left( 1 + \frac{2(\boldsymbol{\alpha}_3)^2 c_{x0}}{(\boldsymbol{\alpha}_2 + 2\sqrt{k_c}c_{w0}\boldsymbol{\alpha}_3\mu)\sqrt{n}} \right)^2 \right] = \mathcal{O}\left( \mu^4 c_{w0}^4 c_{x0}^2 h^{1.5} k_c^2 \left( 1 + \boldsymbol{\alpha}_2 + 2\sqrt{k_c}c_{w0}\boldsymbol{\alpha}_3\mu \right)^{4l} \right)$.

**Step 5. Upper bound $\|\boldsymbol{y} - \boldsymbol{u}(k+1)\|_2^2$.**

In this way, by using Eqn. (9) we can finally obtain

$$\|\boldsymbol{y} - \boldsymbol{u}(k+1)\|_2^2 \leq \|\boldsymbol{y} - \boldsymbol{u}(k)\|_2^2 + H_1 + H_2 + H_3$$

$$\overset{①}{\leq} \|\boldsymbol{y} - \boldsymbol{u}(k)\|_2^2 - 2\eta\lambda\|\boldsymbol{u}(k) - \boldsymbol{y}\|_2^2 + 2\hat{c}\eta^2 \|\boldsymbol{u}(t) - \boldsymbol{y}\|_2^2 + \eta^2\tilde{c} \|\boldsymbol{u}(k) - \boldsymbol{y}\|_2^2$$

$$= \left( 1 - \eta\lambda + (2\hat{c} + \tilde{c})\eta^2 \right) \|\boldsymbol{y} - \boldsymbol{u}(k)\|_2^2$$

$$\overset{②}{\leq} \left( 1 - \frac{\eta\lambda}{2} \right) \|\boldsymbol{y} - \boldsymbol{u}(k)\|_2^2$$

where ① holds by using $H_1 \leq -2\eta\lambda\|\boldsymbol{u}(k) - \boldsymbol{y}\|_2^2$, $H_2 \leq 2\hat{c}\eta^2 \|\boldsymbol{u}(t) - \boldsymbol{y}\|_2^2$ and $H_3 \leq \eta^2\tilde{c} \|\boldsymbol{u}(k) - \boldsymbol{y}\|_2^2$; ② holds by setting $\eta \leq \frac{\lambda}{2(2\hat{c}+\tilde{c})} = \mathcal{O}\left( \frac{\lambda}{\sqrt{m}\mu^4 c_{w0}^4 c_{x0}^2 h^3 k_c^2 \left( 1 + \boldsymbol{\alpha}_2 + 2\sqrt{k_c}c_{w0}\boldsymbol{\alpha}_3\mu \right)^{4l}} \right)$. The proof is completed.

$\square$

### C.3.2 Proof of Lemma 20

*Proof.* According to the definitions in Sec. B, we can write

$$\|\boldsymbol{G}(k) - \boldsymbol{G}(0)\|_2 \leq \left\| \bar{\boldsymbol{G}}^0(k) - \bar{\boldsymbol{G}}^0(0) \right\|_2 + \sum_{l=0}^{h-1} \sum_{s=0}^{l-1} \left\| \boldsymbol{G}^{ls}(k) - \boldsymbol{G}^{ls}(0) \right\|_2 + \sum_{s=0}^{h-1} \|\boldsymbol{G}^s(k) - \boldsymbol{G}^s(0)\|_2.$$

In this way, we only need to upper bound $\left\|\bar{\boldsymbol{G}}^0(k) - \bar{\boldsymbol{G}}^0(0)\right\|_2$, $\left\|\boldsymbol{G}^{ls}(k) - \boldsymbol{G}^{ls}(0)\right\|_2$ and $\left\|\boldsymbol{G}^s(k) - \boldsymbol{G}^s(0)\right\|_2$.

**Step 1. Bound of $\left\|\boldsymbol{G}^s(k) - \boldsymbol{G}^s(0)\right\|_2$ ($s = 0, \cdots, h-1$).**

For analysis, we first recall existing results. Lemma 13 shows

$$\max\left(\|\boldsymbol{W}^{(0)}(t) - \boldsymbol{W}^{(0)}(0)\|_F, \|\boldsymbol{W}_s^{(l)}(t) - \boldsymbol{W}_s^{(l)}(0)\|_F, \|\boldsymbol{W}_s(t) - \boldsymbol{W}_s(0)\|_F\right) \leq \sqrt{m}\widetilde{r} \leq \sqrt{m}c_{w0}, \quad (15)$$

where $c = \left(1 + \boldsymbol{\alpha}_2 + 2\boldsymbol{\alpha}_3\mu\sqrt{k_c}c_{w0}\right)^l$ with $\boldsymbol{\alpha}_2 = \max_{s,l} \boldsymbol{\alpha}_{s,2}^{(l)}$ and $\boldsymbol{\alpha}_3 = \max_{s,l} \boldsymbol{\alpha}_{s,3}^{(l)}$. Based on this result, Lemma 15 shows

$$\left\|\boldsymbol{W}^{(0)}(k)\right\|_F \leq 2\sqrt{m}c_{w0}, \ \left\|\boldsymbol{W}_s^{(l)}(k)\right\|_F \leq 2\sqrt{m}c_{w0}, \ \|\boldsymbol{W}_s(k)\|_F \leq 2\sqrt{m}c_{w0}, \ \left\|\boldsymbol{X}_i^{(l)}(k)\right\|_F \leq 2c_{x0}. \tag{16}$$

Moreover, Lemma 16 shows

$$\|\boldsymbol{X}_i^{(0)}(k) - \boldsymbol{X}_i^{(0)}(0)\|_F \leq \mu\sqrt{k_c}\widetilde{r}, \quad \|\boldsymbol{X}_i^{(l)}(k) - \boldsymbol{X}_i^{(l)}(0)\|_F \leq c(1 + 2\boldsymbol{\alpha}_3 c_{x0})\mu\sqrt{k_c}\widetilde{r}.$$

To bound $H_s$, we only need to bound each entry in $(\boldsymbol{G}^s(k) - \boldsymbol{G}^s(0))$:

$$\begin{aligned}
|\boldsymbol{G}^s(k) - \boldsymbol{G}^s(0)| &= \left|\left\langle \frac{\partial \ell_i}{\partial \boldsymbol{W}_s(k)}, \frac{\partial \ell_j}{\partial \boldsymbol{W}_s(k)}\right\rangle - \left\langle \frac{\partial \ell_i}{\partial \boldsymbol{W}_s(0)}, \frac{\partial \ell_j}{\partial \boldsymbol{W}_s(0)}\right\rangle\right| \\
&= \left|\left\langle \boldsymbol{X}_i^{(s)}(k), \boldsymbol{X}_j^{(s)}(k)\right\rangle - \left\langle \boldsymbol{X}_i^{(s)}(0), \boldsymbol{X}_j^{(s)}(0)\right\rangle\right| \\
&\leq \left|\left\langle \boldsymbol{X}_i^{(s)}(k) - \boldsymbol{X}_i^{(s)}(0), \boldsymbol{X}_j^{(s)}(k)\right\rangle\right| + \left|\left\langle \boldsymbol{X}_i^{(s)}(0), \boldsymbol{X}_j^{(s)}(k) - \boldsymbol{X}_j^{(s)}(0)\right\rangle\right| \\
&\leq \left\|\boldsymbol{X}_i^{(s)}(k) - \boldsymbol{X}_i^{(s)}(0)\right\|_F \left\|\boldsymbol{X}_j^{(s)}(k)\right\|_F + \left\|\boldsymbol{X}_i^{(s)}(0)\right\|_F \left\|\boldsymbol{X}_j^{(s)}(k) - \boldsymbol{X}_j^{(s)}(0)\right\|_F \\
&\overset{\textcircled{1}}{\leq} 4c_{x0}c(1 + 2\boldsymbol{\alpha}_3 c_{x0})\mu\sqrt{k_c}\widetilde{r},
\end{aligned}$$

So we can further bound

$$\|\boldsymbol{G}^s(k) - \boldsymbol{G}^s(0)\|_2 \leq \sqrt{n} \|\boldsymbol{G}^s(k) - \boldsymbol{G}^s(0)\|_\infty \leq 4c_{x0}c(1 + 2\boldsymbol{\alpha}_3 c_{x0})\mu\sqrt{k_c}\widetilde{r}, \ (1 \leq s \leq h).$$

**Step 2. Bound of $\left\|\boldsymbol{G}^{ls}(k) - \boldsymbol{G}^{ls}(0)\right\|_2$ ($0 \leq l \leq h-1, 0 \leq s \leq l-1$).**

**We first consider $l = h - 1$, namely bound of $\left\|\boldsymbol{G}^{hs}(k) - \boldsymbol{G}^{hs}(0)\right\|_2$ ($0 \leq s \leq h-2$).** For notation simplicity, we use $h$ to denote $h-1$. In this way, according to Lemma 8, we have

$$\frac{\partial u}{\partial \boldsymbol{W}_s^{(h)}} = \boldsymbol{\alpha}_{s,3}^{(h)}\tau\Phi(\boldsymbol{X}^{(s)})\left(\sigma'\left(\boldsymbol{W}_s^{(h)}\Phi(\boldsymbol{X}^{(s)})\right) \odot \boldsymbol{W}_h\right)^\top \ (1 \leq s \leq h-1).$$

Let $\boldsymbol{H}_i = \Phi(\boldsymbol{X}_i^{(s)})$, $\boldsymbol{H}_{i,:t} = [\boldsymbol{H}_i]_{:,t}$, $\boldsymbol{H}_{i,tr} = [\boldsymbol{H}_i]_{t,r}$, and $\boldsymbol{Z}_{i,tr} = (\boldsymbol{W}_{s,:r}^{(h)})^\top\boldsymbol{H}_{i,:t}$. In this way, for $1 \leq s \leq h-1$ we can write $\boldsymbol{G}_{ij}^{hs}$ as

$$\begin{aligned}
\boldsymbol{G}_{ij}^{hs} &= (\boldsymbol{\alpha}_{s,3}^{(h)}\tau)^2 \sum_{r=1}^m \left[\sum_{t=1}^p \boldsymbol{W}_{h,tr}\boldsymbol{H}_{i,:t}(\sigma'\left((\boldsymbol{W}_{s,:r}^{(h)})^\top\boldsymbol{H}_{i,:t}\right)\right]^\top \left[\sum_{q=1}^p \boldsymbol{W}_{h,qr}\boldsymbol{H}_{j,:q}(\sigma'\left((\boldsymbol{W}_{s,:r}^{(h)})^\top\boldsymbol{H}_{j,:q}\right)\right] \\
&= (\boldsymbol{\alpha}_{s,3}^{(h)}\tau)^2 \sum_{t=1}^p \sum_{q=1}^p \boldsymbol{H}_{i,:t}^\top\boldsymbol{H}_{j,:q} \sum_{r=1}^m \boldsymbol{W}_{h,tr}\boldsymbol{W}_{h,qr}\sigma'\left(\boldsymbol{Z}_{i,tr}\right)\sigma'\left(\boldsymbol{Z}_{i,qr}\right).
\end{aligned}$$

Then we can obtain

$$\begin{aligned}
&|\boldsymbol{G}_{ij}^{hs}(k) - \boldsymbol{G}_{ij}^{hs}(0)| \\
&= (\boldsymbol{\alpha}_{s,3}^{(h)}\tau)^2 \left| \sum_{t=1}^p \sum_{q=1}^p (\boldsymbol{H}_{i,:t}(k))^\top\boldsymbol{H}_{j,:q}(k) \sum_{r=1}^m \boldsymbol{W}_{h,tr}(k)\boldsymbol{W}_{h,qr}(k)\sigma'\left(\boldsymbol{Z}_{i,tr}(k)\right)\sigma'\left(\boldsymbol{Z}_{j,qr}(k)\right) \right. \\
&\left. \qquad\qquad - \sum_{t=1}^p \sum_{q=1}^p (\boldsymbol{H}_{i,:t}(k))^\top\boldsymbol{H}_{j,:q}(k) \sum_{r=1}^m \boldsymbol{W}_{h,tr}(k)\boldsymbol{W}_{h,qr}(k)\sigma'\left(\boldsymbol{Z}_{i,tr}(k)\right)\sigma'\left(\boldsymbol{Z}_{j,qr}(k)\right) \right|.
\end{aligned}$$

For brevity, we define $\boldsymbol{A}_1$, $\boldsymbol{A}_2$ and $\boldsymbol{A}_3$ as follows:

$$\boldsymbol{A}_1 = \left| \sum_{t=1}^{p}\sum_{q=1}^{p} \left( (\boldsymbol{H}_{i,:t}(k))^\top \boldsymbol{H}_{j,:q}(k) - (\boldsymbol{H}_{i,:t}(0))^\top \boldsymbol{H}_{j,:q}(0) \right) \sum_{r=1}^{m} \boldsymbol{W}_{h,tr}(0)\boldsymbol{W}_{h,qr}(0)\sigma'\left(\boldsymbol{Z}_{i,tr}(k)\right)\sigma'\left(\boldsymbol{Z}_{j,qr}(k)\right) \right|,$$

$$\boldsymbol{A}_2 = \left| \sum_{t=1}^{p}\sum_{q=1}^{p}(\boldsymbol{H}_{i,:t}(0))^\top \boldsymbol{H}_{j,:q}(0)\sum_{r=1}^{m}\boldsymbol{W}_{h,tr}(0)\boldsymbol{W}_{h,qr}(0)\left(\sigma'(\boldsymbol{Z}_{i,tr}(k))\,\sigma'(\boldsymbol{Z}_{j,qr}(k)) - \sigma'(\boldsymbol{Z}_{i,tr}(0))\,\sigma'(\boldsymbol{Z}_{j,qr}(0))\right) \right|,$$

$$\boldsymbol{A}_3 = \left| \sum_{t=1}^{p}\sum_{q=1}^{p}(\boldsymbol{H}_{i,:t}(0))^\top \boldsymbol{H}_{j,:q}(0)\sum_{r=1}^{m}\left(\boldsymbol{W}_{h,tr}(k)\boldsymbol{W}_{h,qr}(k) - \boldsymbol{W}_{h,tr}(0)\boldsymbol{W}_{h,qr}(0)\right)\sigma'\left(\boldsymbol{Z}_{i,tr}(k)\right)\sigma'\left(\boldsymbol{Z}_{j,qr}(k)\right) \right|.$$

Then we have

$$\left| \boldsymbol{G}_{ij}^{hs}(k) - \boldsymbol{G}_{ij}^{hs}(0) \right| = (\boldsymbol{\alpha}_{s,3}^{(h)}\tau)^2 \left(\boldsymbol{A}_1 + \boldsymbol{A}_2 + \boldsymbol{A}_3\right).$$

The remaining work is to upper bound $\boldsymbol{A}_1$, $\boldsymbol{A}_2$ and $\boldsymbol{A}_3$. We first look at $\boldsymbol{A}_1$:

$$\boldsymbol{A}_1 = \left| \sum_{t=1}^{p}\sum_{q=1}^{p}\left(\boldsymbol{H}_{i,:t}(k)^\top \boldsymbol{H}_{j,:q}(k) - (\boldsymbol{H}_{i,:t}(0))^\top \boldsymbol{H}_{j,:q}(0)\right)\sum_{r=1}^{m}\boldsymbol{W}_{h,tr}(0)\boldsymbol{W}_{h,qr}(0)\sigma'(\boldsymbol{Z}_{i,tr}(k))\,\sigma'(\boldsymbol{Z}_{j,qr}(k)) \right|$$

$$\leq m\mu^2 c_{u0}^2 \left| \sum_{t=1}^{p}\sum_{q=1}^{p}\left((\boldsymbol{H}_{i,:t}(k))^\top \boldsymbol{H}_{j,:q}(k) - (\boldsymbol{H}_{i,:t}(0))^\top \boldsymbol{H}_{j,:q}(0)\right) \right|$$

$$\overset{\textcircled{\scriptsize 1}}{\leq} m\mu^2 c_{u0}^2 \sum_{t=1}^{p}\sum_{q=1}^{p}\left[ \left| (\boldsymbol{H}_{i,:t}(k) - \boldsymbol{H}_{i,:t}(0))^\top \boldsymbol{H}_{j,:q}(k) \right| + \left| (\boldsymbol{H}_{i,:t}(0))^\top (\boldsymbol{H}_{j,:q}(k) - \boldsymbol{H}_{j,:q}(0)) \right| \right]$$

$$\leq m\mu^2 c_{u0}^2 \sqrt{\sum_{t=1}^{p}\sum_{q=1}^{p}\|\boldsymbol{H}_{i,:t}(k) - (\boldsymbol{H}_{i,:t}(0)\|_2^2}\sqrt{\sum_{t=1}^{p}\sum_{q=1}^{p}\|\boldsymbol{H}_{j,:q}(k)\|_2^2}$$

$$+ m\mu^2 c_{u0}^2 \sqrt{\sum_{t=1}^{p}\sum_{q=1}^{p}\|\boldsymbol{H}_{j,:q}(k) - \boldsymbol{H}_{j,:q}(0)\|_2^2}\sqrt{\sum_{t=1}^{p}\sum_{q=1}^{p}\|\boldsymbol{H}_{i,:t}(0)\|_2^2}$$

$$\leq mp\mu^2 c_{u0}^2 \left(\|\boldsymbol{H}_i(k) - \boldsymbol{H}_i(0)\|_F\|\boldsymbol{H}_j(k)\|_F + \|\boldsymbol{H}_j(k) - \boldsymbol{H}_j(0)\|_F\|\boldsymbol{H}_i(k)\|_F\right)$$

$$\leq mp\mu^2 c_{u0}^2 \left(\|\boldsymbol{H}_i(k) - \boldsymbol{H}_i(0)\|_F\|\boldsymbol{H}_j(k)\|_F + \|\boldsymbol{H}_j(k) - \boldsymbol{H}_j(0)\|_F\|\boldsymbol{H}_i(k)\|_F\right)$$

where $\textcircled{\scriptsize 1}$ holds since the activation function $\sigma(\cdot)$ is $\mu$-Lipschitz and $\rho$-smooth and the assumption $\|\boldsymbol{W}_s\|_\infty \leq c_{u0}$. To bound $\|\boldsymbol{H}_i(k) - \boldsymbol{H}_i(0)\|_F\|\boldsymbol{H}_j(k)\|_F$, we first recall our existing results. Lemma 16 that

$$\|\boldsymbol{X}_i^{(l)}(k) - \boldsymbol{X}_i^{(l)}(0)\|_F \leq c(1 + 2\boldsymbol{\alpha}_3 c_{x0})\mu\sqrt{k_c}\widetilde{r},$$

where $c = \left(1 + \boldsymbol{\alpha}_2 + 2\boldsymbol{\alpha}_3\mu\sqrt{k_c}c_{w0}\right)^l$ with $\boldsymbol{\alpha}_2 = \max_{s,l}\boldsymbol{\alpha}_{s,2}^{(l)}$ and $\boldsymbol{\alpha}_3 = \max_{s,l}\boldsymbol{\alpha}_{s,3}^{(l)}$. Here $\widetilde{r}$ is given in Lemma 13. Based on this result, Lemma 15 shows that (16) holds. So we have

$$\|\boldsymbol{H}_i(k) - \boldsymbol{H}_i(0)\|_F \leq \|\Phi(\boldsymbol{X}_i^{(s)}(k)) - \Phi(\boldsymbol{X}_i^{(s)}(0))\|_F \leq \sqrt{k_c}\|\boldsymbol{X}_i^{(s)}(k) - \boldsymbol{X}_i^{(s)}(0)\|_F$$
$$\leq c(1 + 2\boldsymbol{\alpha}_3 c_{x0})\mu k_c\widetilde{r}, \tag{17}$$
$$\|\boldsymbol{H}_j(k)\|_F = \|\Phi(\boldsymbol{X}_j^{(s)}(k))\|_F \leq \sqrt{k_c}\|\boldsymbol{X}_j^{(s)}(k)\|_F \leq 2\sqrt{k_c}c_{w0},$$

which indicates

$$\left(\|\boldsymbol{H}_i(k) - \boldsymbol{H}_i(0)\|_F\|\boldsymbol{H}_j(k)\|_F + \|\boldsymbol{H}_j(k) - \boldsymbol{H}_j(0)\|_F\|\boldsymbol{H}_i(k)\|_F\right) \leq 4cc_{w0}(1 + 2\boldsymbol{\alpha}_3 c_{x0})\mu k_c^{1.5}\widetilde{r}.$$

Therefore, we can upper bound

$$\boldsymbol{A}_1 \leq 4cmp\mu^3 k_c^{1.5}c_{u0}^2 c_{w0}(1 + 2\boldsymbol{\alpha}_3 c_{x0})\widetilde{r}.$$

Then we consider to bound $\boldsymbol{A}_2$. To begin with, we have

$$\left| \sigma'\left(\boldsymbol{Z}_{i,tr}(k)\right)\sigma'\left(\boldsymbol{Z}_{j,qr}(k)\right) - \sigma'\left(\boldsymbol{Z}_{i,tr}(0)\right)\sigma'\left(\boldsymbol{Z}_{j,qr}(0)\right) \right|$$
$$\leq \left| \left(\sigma'\left(\boldsymbol{Z}_{i,tr}(k)\right) - \sigma'\left(\boldsymbol{Z}_{i,tr}(0)\right)\right)\sigma'\left(\boldsymbol{Z}_{j,qr}(k)\right) \right| + \left| \sigma'\left(\boldsymbol{Z}_{i,tr}(0)\right)\left(\sigma'\left(\boldsymbol{Z}_{j,qr}(k)\right) - \sigma'\left(\boldsymbol{Z}_{j,qr}(0)\right)\right) \right|$$
$$\overset{\textcircled{\scriptsize 1}}{\leq} \mu \left| \sigma'\left(\boldsymbol{Z}_{i,tr}(k)\right) - \sigma'\left(\boldsymbol{Z}_{i,tr}(0)\right) \right| + \mu \left| \sigma'\left(\boldsymbol{Z}_{j,qr}(k)\right) - \sigma'\left(\boldsymbol{Z}_{j,qr}(0)\right) \right|$$
$$\overset{\textcircled{\scriptsize 2}}{\leq} \mu\rho \left| \boldsymbol{Z}_{i,tr}(k) - \boldsymbol{Z}_{i,tr}(0) \right| + \mu\rho \left| \boldsymbol{Z}_{j,qr}(k) - \boldsymbol{Z}_{j,qr}(0) \right|,$$

where ① holds since the activation function $\sigma(\cdot)$ is $\mu$-Lipschitz; ② holds since the activation function $\sigma(\cdot)$ is $\rho$-smooth. Therefore, we can upper bound

$$\boldsymbol{A}_2 \leq \sum_{t=1}^{p}\sum_{q=1}^{p}\left|\boldsymbol{H}_{i,:t}(0)^{\top}\boldsymbol{H}_{j,:q}(0)\right|\sum_{r=1}^{m}|\boldsymbol{W}_{h,tr}(0)\boldsymbol{W}_{h,qr}(0)|$$

$$\cdot\left|\left(\sigma'\left(\boldsymbol{Z}_{i,tr}(k)\right)\sigma'\left(\boldsymbol{Z}_{j,qr}(k)\right)-\sigma'\left(\boldsymbol{Z}_{i,tr}(0)\right)\sigma'\left(\boldsymbol{Z}_{j,qr}(0)\right)\right)\right|$$

$$\leq\mu\rho\sum_{t=1}^{p}\sum_{q=1}^{p}\left|(\boldsymbol{H}_{i,:t}(0))^{\top}\boldsymbol{H}_{j,:q}(0)\right|\sum_{r=1}^{m}|\boldsymbol{W}_{h,tr}(0)\boldsymbol{W}_{h,qr}(0)|\left[|\boldsymbol{Z}_{i,tr}(k)-\boldsymbol{Z}_{i,tr}(0)|+|\boldsymbol{Z}_{j,qr}(k)-\boldsymbol{Z}_{j,qr}(0)|\right]$$

$$\leq\mu\rho\sqrt{\sum_{t=1}^{p}\sum_{q=1}^{p}\||\boldsymbol{H}_{i,:t}(0)\|_{2}^{2}\|\boldsymbol{H}_{j,:q}(0)\|_{2}^{2}}\left[\sqrt{\sum_{t=1}^{p}\sum_{q=1}^{p}\left(\sum_{r=1}^{m}|\boldsymbol{W}_{h,tr}(0)\boldsymbol{W}_{h,qr}(0)|\,|\boldsymbol{Z}_{i,tr}(k)-\boldsymbol{Z}_{i,tr}(0)|\right)^{2}}\right.$$

$$\left.+\sqrt{\sum_{t=1}^{p}\sum_{q=1}^{p}\left(\sum_{r=1}^{m}|\boldsymbol{W}_{h,tr}(0)\boldsymbol{W}_{h,qr}(0)|\,|\boldsymbol{Z}_{j,qr}(k)-\boldsymbol{Z}_{j,qr}(0)|\right)^{2}}\right]$$

$$\leq\mu\rho c_{u0}\sqrt{m}\,\|\boldsymbol{H}_i(0)\|_F\,\|\boldsymbol{H}_j(0)\|_F\cdot$$

$$\left[\sqrt{\sum_{t=1}^{p}\sum_{q=1}^{p}\sum_{r=1}^{m}|\boldsymbol{Z}_{i,tr}(k)-\boldsymbol{Z}_{i,tr}(0)|^{2}}+\sqrt{\sum_{t=1}^{p}\sum_{q=1}^{p}\sum_{r=1}^{m}|\boldsymbol{Z}_{j,tr}(k)-\boldsymbol{Z}_{j,tr}(0)|^{2}}\right]$$

$$\leq\mu\rho c_{u0}\sqrt{mp}\,\|\boldsymbol{H}_i(0)\|_F\,\|\boldsymbol{H}_j(0)\|_F\left[\|\boldsymbol{Z}_i(k)-\boldsymbol{Z}_i(0)\|_F+\|\boldsymbol{Z}_j(k)-\boldsymbol{Z}_j(0)\|_F\right].$$

From Eqn. (17), we have $\|\boldsymbol{H}_j(k)\|_F\leq 2\sqrt{k_c}c_{w0}$. Lemma 13 shows that Eqn. (15) holds. Based on this result and the fact that $\widetilde{r}\leq c_{w0}$, Lemma 11 shows

$$\left\|\boldsymbol{W}_s^{(l)}(k)\Phi(\boldsymbol{X}^{(s)}(k))-\boldsymbol{W}_s^{(l)}(0)\Phi(\boldsymbol{X}^{(s)}(0))\right\|_F\leq\frac{c}{\boldsymbol{\alpha}_3}\sqrt{k_c m}\widetilde{r}.$$

Therefore we can bound

$$\boldsymbol{A}_2\leq\frac{8cmk_c^{1.5}c_{w0}^2\mu\rho c_{u0}\sqrt{p}\widetilde{r}}{\boldsymbol{\alpha}_3}.$$

Now we bound $\boldsymbol{A}_3$ as follows:

$$\boldsymbol{A}_3=\left|\sum_{t=1}^{p}\sum_{q=1}^{p}(\boldsymbol{H}_{i,:t}(0))^{\top}\boldsymbol{H}_{j,:q}(0)\sum_{r=1}^{m}(\boldsymbol{W}_{h,tr}(k)\boldsymbol{W}_{h,qr}(k)-\boldsymbol{W}_{h,tr}(0)\boldsymbol{W}_{h,qr}(0))\,\sigma'(\boldsymbol{Z}_{i,tr}(k))\,\sigma'(\boldsymbol{Z}_{j,qr}(k))\right|$$

$$\leq\mu^2\left|\sum_{t=1}^{p}\sum_{q=1}^{p}(\boldsymbol{H}_{i,:t}(0))^{\top}\boldsymbol{H}_{j,:q}(0)\sum_{r=1}^{m}(\boldsymbol{W}_{h,tr}(k)\boldsymbol{W}_{h,qr}(k)-\boldsymbol{W}_{h,tr}(0)\boldsymbol{W}_{h,qr}(0))\right|$$

$$\leq\mu^2\sum_{t=1}^{p}\sum_{q=1}^{p}\left|(\boldsymbol{H}_{i,:t}(0))^{\top}\boldsymbol{H}_{j,:q}(0)\right|\sum_{r=1}^{m}(|\boldsymbol{W}_{h,tr}(k)-\boldsymbol{W}_{h,tr}(0)||\boldsymbol{W}_{h,qr}(k)|+|\boldsymbol{W}_{h,tr}(0)||\boldsymbol{W}_{h,qr}(k)-\boldsymbol{W}_{h,qr}(0)|)$$

$$\leq\mu^2\sum_{t=1}^{p}\sum_{q=1}^{p}\left|(\boldsymbol{H}_{i,:t}(0))^{\top}\boldsymbol{H}_{j,:q}(0)\right|(\|\boldsymbol{W}_{h,t:}(k)-\boldsymbol{W}_{h,t:}(0)\|_2\|\boldsymbol{W}_{h,q:}(k)\|_2+\|\boldsymbol{W}_{h,t:}(0)\|_2\|\boldsymbol{W}_{h,qr}(k)-\boldsymbol{W}_{h,qr}(0)\|_2)$$

$$\leq\mu^2\sqrt{\sum_{t=1}^{p}\sum_{q=1}^{p}\||\boldsymbol{H}_{i,:t}(0)\|_2^2\|\boldsymbol{H}_{j,:q}(0)\|_2^2}\left[\sqrt{\sum_{t=1}^{p}\sum_{q=1}^{p}\|\boldsymbol{W}_{h,t:}(k)-\boldsymbol{W}_{h,t:}(0)\|_2\|\boldsymbol{W}_{h,q:}(k)\|_2}\right.$$

$$\left.+\sqrt{\sum_{t=1}^{p}\sum_{q=1}^{p}\|\boldsymbol{W}_{h,t:}(k)-\boldsymbol{W}_{h,t:}(0)\|_2\|\boldsymbol{W}_{h,q:}(k)\|_2}\right]$$

$$\leq\mu^2\,\|\boldsymbol{H}_i(0)\|_F\,\|\boldsymbol{H}_j(0)\|_F\left[\|\boldsymbol{W}_h(k)-\boldsymbol{W}_h(0)\|_F\|\boldsymbol{W}_h(k)\|_F+\|\boldsymbol{W}_h(k)-\boldsymbol{W}_h(0)\|_F\|\boldsymbol{W}_h(k)\|_F\right]$$

$$\overset{①}{\leq}8k_c\mu^2 c_{w0}^3 m\widetilde{r},$$

where ① holds by using Eqn.s (15), (16), (17).

By combining the above results, we have that for $s=0,\cdots,h-1$

$$|\boldsymbol{G}^{hs}(k)-\boldsymbol{G}^{hs}(0)\|_2\leq\sqrt{n}|\boldsymbol{G}_{ij}^{hs}(k)-\boldsymbol{G}_{ij}^{hs}(0)|_\infty$$

$$\leq 4(\boldsymbol{\alpha}_{s,3}^{(h)})^2 k_c\mu c_{w0}n^{0.5}\widetilde{r}\left(cp\mu^2 k_c^{0.5}c_{u0}^2(1+2\boldsymbol{\alpha}_3 c_{x0})+\frac{2ck_c^{0.5}c_{w0}\rho c_{u0}\sqrt{p}}{\boldsymbol{\alpha}_3}+2\mu c_{w0}^2\right).$$

**Then we consider** $1 \leq l < h$**, namely bound of** $H_{ls}$ $(0 \leq s \leq h-1)$**.** For brevity, let $\boldsymbol{B}_i(k) = \frac{\partial \ell}{\partial \boldsymbol{X}_i^{(l)}(k)}$. Here we use the same strategy as above. Let

$$\boldsymbol{A}_1 = \sum_{t=1}^{p}\sum_{q=1}^{p}\left((\boldsymbol{H}_{i,:t}(k))^\top \boldsymbol{H}_{j,:q}(k) - (\boldsymbol{H}_{i,:t}(0))^\top \boldsymbol{H}_{j,:q}(0)\right)\sum_{r=1}^{m}\boldsymbol{B}_{i,tr}(0)\boldsymbol{B}_{j,qr}(0)\sigma'(\boldsymbol{Z}_{i,tr}(k))\,\sigma'(\boldsymbol{Z}_{j,qr}(k)),$$

$$\boldsymbol{A}_2 = \sum_{t=1}^{p}\sum_{q=1}^{p}\boldsymbol{H}_{i,:t}(0)^\top \boldsymbol{H}_{j,:q}(0)\sum_{r=1}^{m}\boldsymbol{B}_{i,tr}(0)\boldsymbol{B}_{j,qr}(0)\left(\sigma'(\boldsymbol{Z}_{i,tr}(k))\,\sigma'(\boldsymbol{Z}_{j,qr}(k)) - \sigma'(\boldsymbol{Z}_{i,tr}(0))\,\sigma'(\boldsymbol{Z}_{j,qr}(0))\right),$$

$$\boldsymbol{A}_{3,ij} = \sum_{t=1}^{p}\sum_{q=1}^{p}(\boldsymbol{H}_{i,:t}(0))^\top \boldsymbol{H}_{j,:q}(0)\sum_{r=1}^{m}\left(\boldsymbol{B}_{i,tr}(k)\boldsymbol{B}_{j,qr}(k) - \boldsymbol{B}_{i,tr}(0)\boldsymbol{B}_{j,qr}(0)\right)\sigma'\left(\boldsymbol{Z}_{i,tr}(k)\right)\sigma'\left(\boldsymbol{Z}_{j,qr}(k)\right).$$

By assuming $\|\boldsymbol{B}_i(k)\|_\infty \leq c_{u0}$, we can use the same method to bound $\boldsymbol{A}_1$ and $\boldsymbol{A}_2$ as follows:

$$|\boldsymbol{A}_1| \leq 4cmp\mu^3 k_c^{1.5} c_{u0}^2 c_{w0}(1 + 2\boldsymbol{\alpha}_3 c_{x0})\widetilde{r}, \quad |\boldsymbol{A}_2| \leq \frac{8cmk_c^{1.5}c_{w0}^2\mu\rho c_{u0}\sqrt{p\widetilde{r}}}{\boldsymbol{\alpha}_3}.$$

Then we need to carefully bound $\boldsymbol{A}_3$:

$$|\boldsymbol{A}_{3,ij}| = \left|\sum_{t=1}^{p}\sum_{q=1}^{p}(\boldsymbol{H}_{i,:t}(0))^\top \boldsymbol{H}_{j,:q}(0)\sum_{r=1}^{m}\left(\boldsymbol{B}_{i,tr}(k)\boldsymbol{B}_{j,qr}(k) - \boldsymbol{B}_{i,tr}(0)\boldsymbol{B}_{j,qr}(0)\right)\sigma'\left(\boldsymbol{Z}_{i,tr}(k)\right)\sigma'\left(\boldsymbol{Z}_{j,qr}(k)\right)\right|$$

$$\leq \mu^2\left|\sum_{t=1}^{p}\sum_{q=1}^{p}(\boldsymbol{H}_{i,:t}(0))^\top \boldsymbol{H}_{j,:q}(0)\sum_{r=1}^{m}\left(\boldsymbol{B}_{i,tr}(k)\boldsymbol{B}_{j,qr}(k) - \boldsymbol{B}_{i,tr}(0)\boldsymbol{B}_{j,qr}(0)\right)\right|$$

$$\leq \mu^2\sum_{t=1}^{p}\sum_{q=1}^{p}\left|(\boldsymbol{H}_{i,:t}(0))^\top \boldsymbol{H}_{j,:q}(0)\right|\sum_{r=1}^{m}\left(|\boldsymbol{B}_{i,tr}(k) - \boldsymbol{B}_{i,tr}(0)||\boldsymbol{B}_{j,qr}(k)| + |\boldsymbol{B}_{i,tr}(0)||\boldsymbol{B}_{j,qr}(k) - \boldsymbol{B}_{j,qr}(0)|\right)$$

$$\leq \mu^2\sum_{t=1}^{p}\sum_{q=1}^{p}\left|(\boldsymbol{H}_{i,:t}(0))^\top \boldsymbol{H}_{j,:q}(0)\right|\left(\|\boldsymbol{B}_{i,t:}(k) - \boldsymbol{B}_{i,t:}(0)\|_2\|\boldsymbol{B}_{j,q:}(k)\|_2 + \|\boldsymbol{B}_{i,t:}(0)\|_2\|\boldsymbol{B}_{j,q:}(k) - \boldsymbol{B}_{j,q:}(0)\|_2\right)$$

$$\leq \mu^2\sqrt{\sum_{t=1}^{p}\sum_{q=1}^{p}\||\boldsymbol{H}_{i,:t}(0)\|_2^2\,\|\boldsymbol{H}_{j,:q}(0)\|_2^2}\left[\sqrt{\sum_{t=1}^{p}\sum_{q=1}^{p}\|\boldsymbol{B}_{i,t:}(k) - \boldsymbol{B}_{i,t:}(0)\|_2^2\|\boldsymbol{B}_{j,q:}(k)\|_2^2}\right.$$

$$\left. + \sqrt{\sum_{t=1}^{p}\sum_{q=1}^{p}\|\boldsymbol{B}_{i,t:}(0)\|_2^2\|\boldsymbol{B}_{j,q:}(k) - \boldsymbol{B}_{j,q:}(0)\|_2^2}\right]$$

$$\leq \mu^2\,\||\boldsymbol{H}_i(0)\|_F\,\|\boldsymbol{H}_j(0)\|_F\left[\|\boldsymbol{B}_i(k) - \boldsymbol{B}_i(0)\|_F\|\boldsymbol{B}_j(k)\|_F + \|\boldsymbol{B}_j(k) - \boldsymbol{B}_j(0)\|_F\|\boldsymbol{B}_i(0)\|_F\right]$$

$$\overset{\text{①}}{\leq} 4\mu^2 c_{w0}^2\left[\|\boldsymbol{B}_i(k) - \boldsymbol{B}_i(0)\|_F\|\boldsymbol{B}_j(k)\|_F + \|\boldsymbol{B}_j(k) - \boldsymbol{B}_j(0)\|_F\|\boldsymbol{B}_i(0)\|_F\right],$$

where ① holds by using Eqn.s (15), (16), (17). Then when for $c_y = \frac{1}{\sqrt{n}}\|\boldsymbol{u}^t - \boldsymbol{y}\|_2$ and $c_u = \|\boldsymbol{W}_t\|_F$, Lemma 12 shows

$$\frac{1}{n}\sum_{i=1}^{n}\left\|\frac{\partial \ell}{\partial \boldsymbol{X}_i^{(l)}(t)}\right\|_F \leq \left(1 + \boldsymbol{\alpha}_2 + \boldsymbol{\alpha}_3\mu\sqrt{k_c}(r + c_{w0})\right)^l c_y c_u$$

$$\overset{\text{①}}{\leq} 2c\sqrt{m}c_{w0}\left(1 - \frac{\eta\lambda}{2}\right)^{t/2}\|\boldsymbol{u}^0 - \boldsymbol{y}\|_2,$$

where $c = \left(1 + \boldsymbol{\alpha}_2 + 2\boldsymbol{\alpha}_3\mu\sqrt{k_c}c_{w0}\right)^l$, $\boldsymbol{\alpha}_2 = \max_{s,l}\boldsymbol{\alpha}_{s,2}^{(l)}$ and $\boldsymbol{\alpha}_3 = \max_{s,l}\boldsymbol{\alpha}_{s,3}^{(l)}$. ① holds since $c_u = \|\boldsymbol{W}_t\|_F \leq \|\boldsymbol{W}_t - \boldsymbol{W}_0\|_F + \|\boldsymbol{W}_0\|_F \leq \sqrt{m}(\widetilde{r} + c_{w0}) \leq 2\sqrt{m}c_{w0}$ and $\|\boldsymbol{u}^t - \boldsymbol{y}\|_2 \leq \left(1 - \frac{\eta\lambda}{2}\right)^{t/2}\|\boldsymbol{u}^0 - \boldsymbol{y}\|_2$ in Theorem 19. Lemma 17 proves

$$\left\|\frac{\partial \ell}{\partial \boldsymbol{X}_i^{(l)}(k)} - \frac{\partial \ell}{\partial \boldsymbol{X}_i^{(l)}(0)}\right\|_F \leq c_1 c \boldsymbol{\alpha}_3 c_{w0}^2 c_{x0}\rho k_c m\widetilde{r},$$

where $c_1$ is a constant. The remaining work is to bound

$$\|\boldsymbol{B}_i(k) - \boldsymbol{B}_i(0)\|_F\|\boldsymbol{B}_j(k)\|_F \leq c_1 c\boldsymbol{\alpha}_3 c_{w0}^2 c_{x0}\rho k_c m\widetilde{r}\|\boldsymbol{B}_j(k)\|_F.$$

In this way, we have

$$\|\boldsymbol{A}_3\|_1 \le \sum_{j=1}^{n}\sum_{i=1}^{n}\|\boldsymbol{A}_{3,ij}\| \le 4\mu^2 c_{w0}^2 c_1 c \boldsymbol{\alpha}_3 c_{w0}^2 c_{x0}\rho k_c m \widetilde{r}\sum_{j=1}^{n}\sum_{i=1}^{n}\left(\|\boldsymbol{B}_j(k)\|_F + \boldsymbol{B}_i(k)\|_F\right)$$

$$\le 8c_1 n\mu^2 c^2 \boldsymbol{\alpha}_3 c_{w0}^5 c_{x0}\rho k_c m^{1.5}\widetilde{r}\left(1-\frac{\eta\lambda}{2}\right)^{t/2}\|\boldsymbol{u}^0 - \boldsymbol{y}\|_2.$$

Then combining all above results gives

$$\left\|\boldsymbol{G}^{hs}(k) - \boldsymbol{G}^{hs}(0)\right\|_2 = (\boldsymbol{\alpha}_{s,3}^{(h)}\tau)^2\|\boldsymbol{A}_1 + \boldsymbol{A}_2 + \boldsymbol{A}_3\|_2 \le (\boldsymbol{\alpha}_{s,3}^{(h)}\tau)^2\left(\|\boldsymbol{A}_1\|_2 + \|\boldsymbol{A}_2\|_2 + \|\boldsymbol{A}_3\|_2\right)$$

$$\le (\boldsymbol{\alpha}_{s,3}^{(h)}\tau)^2\sqrt{n}\left(\|\boldsymbol{A}_1\|_\infty + \|\boldsymbol{A}_2\|_\infty + \|\boldsymbol{A}_3\|_1\right)$$

$$\le 4(\boldsymbol{\alpha}_{s,3}^{(h)})^2 k_c\mu c_{w0}n^{0.5}\widetilde{r}\left(cp\mu^2 k_c^{0.5}c_{u0}^2(1+2\boldsymbol{\alpha}_3 c_{x0}) + \frac{2ck_c^{0.5}c_{w0}\rho c_{u0}\sqrt{p}}{\boldsymbol{\alpha}_3}\right)$$

$$+ 8(\boldsymbol{\alpha}_{s,3}^{(h)})^2 nc_1\mu^2 c^2 \boldsymbol{\alpha}_3 c_{w0}^5 c_{x0}\rho k_c m^{0.5}\widetilde{r}\left(1-\frac{\eta\lambda}{2}\right)^{t/2}\|\boldsymbol{u}^0 - \boldsymbol{y}\|_2.$$

In this way, we only need to upper bound $\left\|\boldsymbol{G}^0(k) - \boldsymbol{G}^0(0)\right\|_2$, $\left\|\boldsymbol{G}^{ls}(k) - \boldsymbol{G}^{ls}(0)\right\|_2$ and $\|\boldsymbol{G}^s(k) - \boldsymbol{G}^s(0)\|_2$.

**Step 3. Bound of $\left\|\bar{\boldsymbol{G}}^0(k) - \bar{\boldsymbol{G}}^0(0)\right\|_2$.**

Here we use the same method when we bound $\left\|\boldsymbol{G}^{ls}(k) - \boldsymbol{G}^{ls}(0)\right\|_2$ to bound $\left\|\boldsymbol{G}^0(k) - \boldsymbol{G}^0(0)\right\|_2$. Let $\boldsymbol{H}_i = \Phi(\boldsymbol{X}_i)$, $\boldsymbol{H}_{i,:t} = [\boldsymbol{H}_i]_{:,t}$, $\boldsymbol{H}_{i,tr} = [\boldsymbol{H}_i]_{t,r}$, $\boldsymbol{Z}_{i,tr} = (\boldsymbol{W}_{s,:r}^{(0)})^\top \boldsymbol{H}_{i,:t}$ and $\boldsymbol{B}_i(k) = \frac{\partial \ell}{\partial \boldsymbol{X}_i^{(l)}(k)}$. In this way, for $1 \le s \le h-1$ we can write $\boldsymbol{G}_{ij}^{hs}$ as Then we define

$$\boldsymbol{A}_1 = \sum_{t=1}^{p}\sum_{q=1}^{p}\left((\boldsymbol{H}_{i,:t}(k))^\top \boldsymbol{H}_{j,:q}(k) - (\boldsymbol{H}_{i,:t}(0))^\top \boldsymbol{H}_{j,:q}(0)\right)\sum_{r=1}^{m}\boldsymbol{B}_{i,tr}(0)\boldsymbol{B}_{j,qr}(0)\sigma'(\boldsymbol{Z}_{i,tr}(k))\,\sigma'(\boldsymbol{Z}_{j,qr}(k))\,,$$

$$\boldsymbol{A}_2 = \sum_{t=1}^{p}\sum_{q=1}^{p}(\boldsymbol{H}_{i,:t}(0))^\top \boldsymbol{H}_{j,:q}(0)\sum_{r=1}^{m}\boldsymbol{B}_{i,tr}(0)\boldsymbol{B}_{j,qr}(0)\left(\sigma'(\boldsymbol{Z}_{i,tr}(k))\,\sigma'(\boldsymbol{Z}_{j,qr}(k)) - \sigma'(\boldsymbol{Z}_{i,tr}(0))\,\sigma'(\boldsymbol{Z}_{j,qr}(0))\right),$$

$$\boldsymbol{A}_{3,ij} = \sum_{t=1}^{p}\sum_{q=1}^{p}(\boldsymbol{H}_{i,:t}(0))^\top \boldsymbol{H}_{j,:q}(0)\sum_{r=1}^{m}\left(\boldsymbol{B}_{i,tr}(k)\boldsymbol{B}_{j,qr}(k) - \boldsymbol{B}_{i,tr}(0)\boldsymbol{B}_{j,qr}(0)\right)\sigma'(\boldsymbol{Z}_{i,tr}(k))\,\sigma'(\boldsymbol{Z}_{j,qr}(k))\,.$$

Then by using the same method, we can prove

$$\left\|\bar{\boldsymbol{G}}^0(k) - \bar{\boldsymbol{G}}^0(0)\right\|_2 = \tau^2\|\boldsymbol{A}_1 + \boldsymbol{A}_2 + \boldsymbol{A}_3\|_2 \le (\boldsymbol{\alpha}_{s,3}^{(h)}\tau)^2\left(\|\boldsymbol{A}_1\|_2 + \|\boldsymbol{A}_2\|_2 + \|\boldsymbol{A}_3\|_2\right)$$

$$\le \tau^2\sqrt{n}\left(\|\boldsymbol{A}_1\|_\infty + \|\boldsymbol{A}_2\|_\infty + \|\boldsymbol{A}_3\|_1\right)$$

$$\le 4k_c\mu c_{w0}n^{0.5}\widetilde{r}\left(cp\mu^2 k_c^{0.5}c_{u0}^2(1+2\boldsymbol{\alpha}_3 c_{x0}) + \frac{2ck_c^{0.5}c_{w0}\rho c_{u0}\sqrt{p}}{\boldsymbol{\alpha}_3}\right)$$

$$+ 8c_1 n\mu^2 c^2 \boldsymbol{\alpha}_3 c_{w0}^5 c_{x0}\rho k_c m^{0.5}\widetilde{r}\left(1-\frac{\eta\lambda}{2}\right)^{k/2}\|\boldsymbol{u}^0 - \boldsymbol{y}\|_2.$$

**Step 4. Bound of $\|\boldsymbol{G}(k) - \boldsymbol{G}(0)\|_2$.**

By combining the above results and ignoring all constants for brevity, we can bound

$$\|\boldsymbol{G}(k) - \boldsymbol{G}(0)\|_2 \le \left\|\bar{\boldsymbol{G}}^0(k) - \bar{\boldsymbol{G}}^0(0)\right\|_2 + \sum_{l=0}^{h-1}\sum_{s=0}^{l-1}\left\|\boldsymbol{G}^{ls}(k) - \boldsymbol{G}^{ls}(0)\right\|_2 + \sum_{s=0}^{h-1}\|\boldsymbol{G}^s(k) - \boldsymbol{G}^s(0)\|_2$$

$$\le c_2 ch\mu k_c^{0.5}c_{x0}\widetilde{r}n^{0.5}\left(\rho h\mu^2 k_c c_{u0}^2 c_{w0} + \boldsymbol{\alpha}_3 c\rho h\mu k_c^{0.5}c_{w0}^5 n^{0.5}\right)$$

where $c = \left(1 + \boldsymbol{\alpha}_2 + 2\boldsymbol{\alpha}_3\mu\sqrt{k_c}c_{w0}\right)^h$ and $c_2$ is a constant. Considering

$$\widetilde{r} = \frac{8c_{x0}\|\boldsymbol{y} - \boldsymbol{u}(0)\|_2}{\lambda\sqrt{mn}}\max\left(1, 2\left(1+\boldsymbol{\alpha}_2 + 2\boldsymbol{\alpha}_3\mu\sqrt{k_c}c_{w0}\right)^h \boldsymbol{\alpha}_3\mu\sqrt{k_c}c_{w0}\right) \le c_{w0},$$

to achieve

$$\|\boldsymbol{G}(k) - \boldsymbol{G}(0)\|_2 \le \frac{\lambda}{2},$$

$m$ should be at the order of

$$m \ge \frac{c_3 \boldsymbol{\alpha}_3^2 \mu^2 k_c c_{x0}^2 c^2}{\lambda^2 n},$$

where $c_3$ is a constant, $c = \left(1 + \boldsymbol{\alpha}_2 + 2\boldsymbol{\alpha}_3 \mu \sqrt{k_c} c_{w0}\right)^h$, $\boldsymbol{\alpha}_2 = \max_{s,l} \boldsymbol{\alpha}_{s,2}^{(l)}$ and $\boldsymbol{\alpha}_3 = \max_{s,l} \boldsymbol{\alpha}_{s,3}^{(l)}$. The proof is completed. $\qquad\square$

### C.3.3 Proof of Lemma 21

*Proof.* Lemma 19 proves that when $m = \mathcal{O}\left(\frac{\rho k_c^2 c_{w0}^2 \|\boldsymbol{y} - \boldsymbol{u}(0)\|_2^2}{\lambda^2 n}\left(1 + \boldsymbol{\alpha}_2 + 2\boldsymbol{\alpha}_3 \mu \sqrt{k_c} c_{w0}\right)^{2h}\right)$, then with probability at least $1 - \delta/2$ we have

$$\|\boldsymbol{y} - \boldsymbol{u}(k)\|_2^2 \le \left(1 - \frac{\eta\lambda}{2}\right)\|\boldsymbol{y} - \boldsymbol{u}(k-1)\|_2^2 \le \left(1 - \frac{\eta\lambda}{2}\right)^k \|\boldsymbol{y} - \boldsymbol{u}(0)\|_2^2,$$

where $\lambda$ is smallest eigenvalue of the Gram matrix $\boldsymbol{G}(t)$ ($t = 1, \cdots, k-1$). Lemma 20 shows that if $m$ satisfies $m \ge \frac{c_3 \boldsymbol{\alpha}_3^2 \mu^2 k_c c_{x0}^2 c^2}{\lambda^2 n}$, where $c_3$ is a constant, $c = \left(1 + \boldsymbol{\alpha}_2 + 2\boldsymbol{\alpha}_3 \mu \sqrt{k_c} c_{w0}\right)^h$, $\boldsymbol{\alpha}_2 = \max_{s,l} \boldsymbol{\alpha}_{s,2}^{(l)}$ and $\boldsymbol{\alpha}_3 = \max_{s,l} \boldsymbol{\alpha}_{s,3}^{(l)}$, then we have

$$\|\boldsymbol{G}(k) - \boldsymbol{G}(0)\|_2 \le \frac{\lambda_{\min}(\boldsymbol{G}(0))}{2},$$

where $\lambda_{\min}(\boldsymbol{G}(0))$ is the smallest eigenvalue of $\boldsymbol{G}(0)$. So we have

$$\lambda_{\min}(\boldsymbol{G}(t)) \ge \frac{\lambda_{\min}(\boldsymbol{G}(0))}{2}.$$

So combining these results, we have

$$\|\boldsymbol{y} - \boldsymbol{u}(k)\|_2^2 \le \left(1 - \frac{\eta\lambda_{\min}(\boldsymbol{G}(0))}{4}\right)\|\boldsymbol{y} - \boldsymbol{u}(k-1)\|_2^2 \le \left(1 - \frac{\eta\lambda_{\min}(\boldsymbol{G}(0))}{4}\right)^k \|\boldsymbol{y} - \boldsymbol{u}(0)\|_2^2,$$

when $m$ satisfies $m \ge \frac{c_m' \rho c^2 k_c^2 c_{w0}^2 \mu^2}{\lambda^2 n}$ and $\eta \le \frac{c_\eta' \lambda}{\sqrt{m} \mu^4 h^3 k_c^2 c^4}$, where $c_m', c_\eta'$ are constants, $c = \left(1 + \boldsymbol{\alpha}_2 + 2\boldsymbol{\alpha}_3 \mu \sqrt{k_c} c_{w0}\right)^h$, $\boldsymbol{\alpha}_2 = \max_{s,l} \boldsymbol{\alpha}_{s,2}^{(l)}$ and $\boldsymbol{\alpha}_3 = \max_{s,l} \boldsymbol{\alpha}_{s,3}^{(l)}$. The proof is completed. $\qquad\square$

### C.4 Step 2 Lower Bound of Eigenvalue of Gram Matrix

Here we define some necessary notations for this subsection first. By Gaussian distribution $\mathcal{P}$ over a $q$-dimensional subspace $\mathcal{W}$, it means that for a basis $\{\boldsymbol{e}_1, \boldsymbol{e}_2, \cdots, \boldsymbol{e}_q\}$ of $\mathcal{W}$ and $(v_1, v_2, \cdots, v_q) \sim \mathcal{N}(0, \boldsymbol{I})$ such that $\sum_{i=1}^q v_i \boldsymbol{e}_i \sim \mathcal{P}$. Then we equip one Gaussian distribution $\mathcal{P}^{(i)}$ with each linear subspace $\mathcal{W}$. Based on these, we define a transform $\mathcal{W}$ as

$$\mathcal{W}_{tq}^{(ls)}(\boldsymbol{K}) = \begin{cases} \mathbb{E}_{\boldsymbol{W}_t^{(l)} \sim \mathcal{P}}[\boldsymbol{W}_t^{(l)} \boldsymbol{K}(\boldsymbol{W}_t^{(l)})^\top], & \text{if } l = s \text{ and } t = q \\ \mathbb{E}_{\boldsymbol{W}_t^{(l)} \sim \mathcal{P}, \boldsymbol{W}_q^{(s)} \sim \mathcal{P}}[\boldsymbol{W}_t^{(l)} \boldsymbol{K}(\boldsymbol{W}_q^{(s)})^\top], & \text{otherwise} \end{cases},$$

where $\boldsymbol{K} \in \mathbb{R}^{p \times p}$ and $\boldsymbol{W}_t^{(l)}$ denotes the parameters in convolution.

Then we define the population Gram matrix as follows. For brevity, let $\bar{\boldsymbol{X}} = \Phi(\boldsymbol{X}) \in \mathbb{R}^{k_c m \times p}$. We first define the case where $l = 0$:

$$\boldsymbol{b}_i^{(-1)} = \boldsymbol{0} \in \mathbb{R}^p, \qquad \boldsymbol{K}_{ij}^{(-1)} = \boldsymbol{X}_i^\top \boldsymbol{X}_i, \qquad \boldsymbol{Q}_{ij}^{(-1)} = \bar{\boldsymbol{X}}_i^\top \bar{\boldsymbol{X}}_i \in \mathbb{R}^{p \times p},$$

$$\boldsymbol{A}^{(00)} = \begin{bmatrix} \mathcal{W}^{(0)}(\boldsymbol{Q}_{ij}^{(-1)}), \mathcal{W}^{(0)}(\boldsymbol{Q}_{ij}^{(-1)}) \\ \mathcal{W}^{(0)}(\boldsymbol{Q}_{ji}^{(-1)}), \mathcal{W}^{(0)}(\boldsymbol{Q}_{jj}^{(-1)}) \end{bmatrix}, \qquad (\boldsymbol{M}^{(00)}, \boldsymbol{N}^{(00)}) \sim \mathcal{N}\left(\boldsymbol{0}, \boldsymbol{A}^{(00)}\right)$$

$$\boldsymbol{b}_i^{(0)} = \tau \mathbb{E}_{\boldsymbol{M}^{(00)}} \sigma(\boldsymbol{M}^{(00)}), \qquad \boldsymbol{K}_{ij}^{(00)} = \mathbb{E}_{(\boldsymbol{M}^{(00)}, \boldsymbol{N}^{(00)})}\left(\sigma(\boldsymbol{M}^{(00)})\sigma(\boldsymbol{N}^{(00)})^\top\right),$$

$$\boldsymbol{Q}_{ij,ab}^{(00)} = \text{Tr}\left(\boldsymbol{K}_{ij,S_a^{(l)}, S_b^{(s)}}^{(00)}\right),$$

where $\mathcal{W}^{(0)}(\boldsymbol{K}) = \mathbb{E}_{\boldsymbol{W}^{(0)}\sim\mathcal{P}}[\boldsymbol{W}^{(0)}\boldsymbol{K}(\boldsymbol{W}^{(0)})^\top]$, $\boldsymbol{Q}_{ij}^{(00)} \in \mathbb{R}^{p\times p}$, $\boldsymbol{K}_{ij,ab}^{(00)}$ denotes the $(a,b)$-th entry in $\boldsymbol{K}_{ij}^{(00)}$, and $S_a^{(0)} = \{j \mid \boldsymbol{X}_{:,j} \in \text{the } a - \text{th patch for convolution}\}$.

Then for $1 \leq l \leq h, 1 \leq s \leq l$, we can recurrently define

$$\boldsymbol{A}_{tq}^{(ls)} = \begin{bmatrix} \mathcal{W}_{tq}^{(ls)}(\boldsymbol{Q}_{ii}^{(tq)}), \mathcal{W}_{tq}^{(ls)}(\boldsymbol{Q}_{ij}^{(tq)}) \\ \mathcal{W}_{tq}^{(ls)}(\boldsymbol{Q}_{ji}^{(tq)}), \mathcal{W}_{tq}^{(ls)}(\boldsymbol{Q}_{jj}^{(tq)}) \end{bmatrix}, \quad (\boldsymbol{M}_{tq}^{(ls)}, \boldsymbol{N}_{tq}^{(ls)}) \sim \mathcal{N}\left(\boldsymbol{0}, \boldsymbol{A}_{tq}^{(ls)}\right), \qquad (0 \leq t, q \leq l-1),$$

$$\boldsymbol{b}_i^{(l)} = \sum_{t=1}^{l-1}\left(\boldsymbol{\alpha}_{t,2}^{(l)}\boldsymbol{b}_i^{(t)} + \tau\boldsymbol{\alpha}_{t,3}^{(l)}\mathbb{E}_{\boldsymbol{M}_{tt}^{(ll)}}\sigma(\boldsymbol{M}_{tt}^{(ll)})\right);$$

$$\boldsymbol{K}_{ij}^{(ls)} = \sum_{t=1}^{l-1}\sum_{q=1}^{s-1}\left[\boldsymbol{\alpha}_{t,2}^{(l)}\boldsymbol{\alpha}_{q,2}^{(s)}\boldsymbol{K}_{ij}^{(tq)} + \tau\mathbb{E}_{(\boldsymbol{M}_{tq}^{(ls)},\boldsymbol{N}_{tq}^{(ls)})}\left(\boldsymbol{\alpha}_{t,3}^{(l)}\boldsymbol{\alpha}_{q,2}^{(s)}\sigma(\boldsymbol{M}_{tq}^{(ls)})(\boldsymbol{b}_j^{(q)})^\top + \boldsymbol{\alpha}_{t,2}^{(l)}\boldsymbol{\alpha}_{q,3}^{(s)}\boldsymbol{b}_i^{(t)}\sigma(\boldsymbol{N}_{tq}^{(ls)})^\top\right.\right.$$

$$\left.\left. + \tau\boldsymbol{\alpha}_{t,3}^{(l)}\boldsymbol{\alpha}_{q,3}^{(s)}\sigma(\boldsymbol{M}_{tq}^{(ls)})\sigma(\boldsymbol{N}_{tq}^{(ls)})^\top\right)\right],$$

$$\boldsymbol{Q}_{ij,ab}^{(ls)} = \text{Tr}\left(\boldsymbol{K}_{ij,S_a^{(l)},S_b^{(s)}}^{(ls)}\right),$$

where $\boldsymbol{K}_{ij}^{(ls)} \in \mathbb{R}^{p\times p}$, $\boldsymbol{Q}_{ij,ab}^{(ls)}$ denotes the $(a,b)$-th entry in $\boldsymbol{Q}_{ij}^{(ls)}$, and $S_a^{(s)} = \{j \mid \boldsymbol{X}_{:,j}^{(s-1)} \in \text{the } a - \text{th patch for convolution}\}$. Finally, we define

$$\boldsymbol{A}^{(s)} = \begin{bmatrix} \mathcal{W}_{ss}^{(hh)}(\boldsymbol{Q}_{ii}^{(ss)}), \mathcal{W}_{ss}^{(hh)}(\boldsymbol{Q}_{ij}^{(ss)}) \\ \mathcal{W}_{ss}^{(hh)}(\boldsymbol{Q}_{ji}^{(ss)}), \mathcal{W}_{ss}^{(hh)}(\boldsymbol{Q}_{jj}^{(ss)}) \end{bmatrix},$$

$$\boldsymbol{Q}_{ij,ab}^{(s)} = \boldsymbol{Q}_{ij,ab}^{(ss)}\mathbb{E}_{((\boldsymbol{M},\boldsymbol{N})\sim\bar{\boldsymbol{A}}^{(s)})}\sigma'(\boldsymbol{M})\sigma'(\boldsymbol{N})^\top, \qquad \boldsymbol{K}_{ij,ab}^{(s)} = \text{Tr}\left(\boldsymbol{Q}_{ij}^{(s)}\right), \ (s=0, h-1).$$

For brevity, we first define

$$\widehat{\boldsymbol{K}}_{ij}^{(ls)} = \frac{1}{m}\sum_{t=1}^{m}\boldsymbol{X}_{i,t}^{(l)}(\boldsymbol{X}_{j,t}^{(s)})^\top, \qquad \widehat{\boldsymbol{b}}_i^{(l)} = \frac{1}{m}\sum_{t=1}^{m}\boldsymbol{X}_{i,t}^{(l)}.$$

Then we prove that $\boldsymbol{K}^{(s)}$ is very close to the randomly generated gram matrix $\widehat{\boldsymbol{K}}_{ij}^{(ls)}$.

**Lemma 22.** *With probability at least $1-\delta$ over the convolution parameters $\boldsymbol{W}$ in each layer, then for $0 \leq t \leq h, 0 \leq s \leq h$, it holds*

$$\left\|\frac{1}{m}\sum_{s=1}^{m}(\boldsymbol{X}_{i,s}^{(t)})^\top\boldsymbol{X}_{j,s}^{(q)} - \boldsymbol{K}_{ij}^{(tq)}\right\|_\infty \leq C\sqrt{\frac{\log(n^2p^2h^2/\delta)}{m}},$$

*and*

$$\left\|\frac{1}{m}\sum_{s=1}^{m}\boldsymbol{X}_{i,s}^{(t)} - \boldsymbol{b}_i^{(t)}\right\|_\infty \leq C\sqrt{\frac{\log(n^2p^2h^2/\delta)}{m}},$$

*where $C$ is a constant which depends on the activation function $\sigma(\cdot)$, namely $C \sim \sigma(0) + \sup_x \sigma'(x)$.*

See its proof in Appendix C.4.1.

**Lemma 23.** *Suppose Assumptions 1, 2 and 3 hold. Then if $m \geq \frac{c_4\mu^2p^2n^2\log(n/\delta)}{\lambda^2}$, we have*

$$\left\|\boldsymbol{G}^{hs}(0) - (\boldsymbol{\alpha}_{s,3}^{(h)})^2\boldsymbol{K}^{(s)}\right\|_{op} \leq \frac{\lambda}{4} \qquad (s=0,\cdots,h),$$

*where $c_4$ and $\lambda$ are constants.*

See its proof in Appendix C.4.2.

**Lemma 24.** *Suppose Assumptions 1, 2 and 3 hold. Suppose $\sigma$ is analytic and not a polynomial function. Consider data $\{\boldsymbol{X}_{i=1}^n\}_{i=1}^n$ are not parallel, namely $\text{vec}(\boldsymbol{X}_i) \notin \text{span}(\text{vec}(\boldsymbol{X}_j))$ for all $i \neq j$. Then if $m \geq \frac{c_4\mu^2p^2n^2\log(n/\delta)}{\lambda^2}$, it holds that with probability at least $1-\delta/2$, the smallest eigenvalue the matrix $\boldsymbol{G}$ satisfies*

$$\lambda_{\min}(\boldsymbol{G}(0)) \geq \frac{3c_\sigma}{4}\sum_{s=0}^{h-1}(\boldsymbol{\alpha}_{s,3}^{(h)})^2\left(\prod_{t=0}^{s-1}(\boldsymbol{\alpha}_{t,2}^{(s)})^2\right)\lambda_{\min}(\boldsymbol{K}).$$

*where* $\lambda = 3c_\sigma \sum_{s=0}^{h-1} (\boldsymbol{\alpha}_{s,3}^{(h)})^2 \left( \prod_{t=0}^{s-1} (\boldsymbol{\alpha}_{t,2}^{(s)})^2 \right) \lambda_{\min}(\boldsymbol{K})$, $c_\sigma$ *is a constant that only depends on $\sigma$ and the input data,* $\lambda_{\min}(\boldsymbol{K}) = \min_{i,j} \lambda_{\min}(\boldsymbol{K}_{ij})$ *is larger than zero in which* $\lambda_{\min}(\boldsymbol{K}_{ij})$ *is the the smallest eigenvalue of* $\boldsymbol{K}_{ij} = \begin{bmatrix} \boldsymbol{X}_i^\top \boldsymbol{X}_j, \boldsymbol{X}_i^\top \boldsymbol{X}_j \\ \boldsymbol{X}_j^\top \boldsymbol{X}_i, \boldsymbol{X}_j^\top \boldsymbol{X}_j \end{bmatrix}$.

See its proof in C.4.3.

### C.4.1    Proof of Lemma 22

*Proof.* We use mathematical induction to prove these results. For brevity, let $\bar{\boldsymbol{X}} = \Phi(\boldsymbol{X}) \in \mathbb{R}^{k_c m \times p}$ and $\boldsymbol{X}_{i,s} = \boldsymbol{X}_{i,s:}^\top \in \mathbb{R}^p$. For the first layer ($l = 0$), we have

$$\boldsymbol{X}_{i,s}^{(0)} = \tau \sigma \left( \sum_{t=1}^{m} \boldsymbol{W}_{ts}^{(0)} \bar{\boldsymbol{X}}_{i,t} \right) \tag{18}$$

Then let

$$\boldsymbol{A}_{i,s}^{(0)} = \sum_{t=1}^{m} \boldsymbol{W}_{ts}^{(0)} \bar{\boldsymbol{X}}_{i,t}. \tag{19}$$

Since the convolution parameter $\boldsymbol{W}$ satisfies Gaussian distribution, $\boldsymbol{A}_{i,s:}^{(0)}$ is a mean-zero Guassian variable with covariance matrix as follows

$$\mathbb{E}\left[ (\boldsymbol{A}_{i,s}^{(0)})^\top \boldsymbol{A}_{j,q}^{(0)} \right] = \mathbb{E} \sum_{t,t'} \boldsymbol{W}_{ts}^{(0)} \bar{\boldsymbol{X}}_{i,t}^{(0)} (\bar{\boldsymbol{X}}_{j,t'})^T (\boldsymbol{W}_{t'q}^{(0)})^T = \delta_{st} \mathcal{W}^{(0)} \left( \sum_t \bar{\boldsymbol{X}}_{i,t} \bar{\boldsymbol{X}}_{j,t}^\top \right) = \delta_{st} \mathcal{W}^{(0)} \left( \boldsymbol{Q}_{ij}^{(-1)} \right),$$

where $\delta_{st}$ is a random variable with $\delta_{st} = \pm 1$ with both probability 0.5. Therefore, we have

$$\mathbb{E}\left[ \frac{1}{m} \sum_{i=1}^{m} \boldsymbol{X}_{i,t}^{(0)} (\boldsymbol{X}_{j,t}^{(0)})^\top \right] = \boldsymbol{K}_{ij}^{(00)}, \quad \mathbb{E}\left[ \frac{1}{m} \sum_{i=1}^{m} \boldsymbol{X}_{i,t}^{(0)} \right] = \boldsymbol{b}_i^{(0)}.$$

In this way, following [4] we can apply Hoeffding and Bernstein bounds and obtain the following results:

$$\mathbb{P}\left( \max_{ij} \left\| \frac{1}{m} \sum_{t=1}^{m} \boldsymbol{X}_{i,t}^{(0)} (\boldsymbol{X}_{j,t}^{(0)})^T - \boldsymbol{K}_{ij}^{(00)} \right\|_\infty \leq \sqrt{\frac{16(1 + 2C_1^2/\sqrt{\pi})M^2 \log(4n^2p^2h^2/\delta))}{m}} \right) \geq 1 - \frac{\delta}{h^2},$$

where we use $\|\boldsymbol{X}_{i,t}^{(0)} (\boldsymbol{X}_{j,t}^{(0)})^\top\|_2 \leq \|\boldsymbol{X}_{i,t}^{(0)} (\boldsymbol{X}_{j,t}^{(0)})^\top\|_F \leq 0.5(\|\boldsymbol{X}_{i,t}^{(0)}\|_F^2 + \|\boldsymbol{X}_{j,t}^{(0)}\|_F^2) \overset{①}{\leq} c_{x0}^2$, $M_1 = 1 + 100 \max_{i,j,s,t,l} |\mathcal{W}^0(\boldsymbol{Q}_{ij}^{(-1)})_{st}|$. Here ① holds by using Lemma 10. Similarly, we can prove

$$\mathbb{P}\left( \left\| \frac{1}{m} \sum_{t=1}^{m} \boldsymbol{X}_{i,t}^{(1)} - \boldsymbol{b}_i^{(1)} \right\|_\infty \leq \sqrt{\frac{2C_1 M \log(2nph/\delta))}{m}} \right) \geq 1 - \delta/h^2.$$

Then we prove the results still hold when $l \geq 1, l \geq s \geq 0$. For brevity, we first define

$$\widehat{\boldsymbol{K}}_{ij}^{(ls)} = \frac{1}{m} \sum_{t=1}^{m} \boldsymbol{X}_{i,t}^{(l)} (\boldsymbol{X}_{j,t}^{(s)})^\top, \qquad \widehat{\boldsymbol{b}}_i^{(l)} = \frac{1}{m} \sum_{t=1}^{m} \boldsymbol{X}_{i,t}^{(l)}.$$

Suppose the results in our lemma holds for $0 \leq l \leq k, 0 \leq q \leq l$ with probability at least $1 - \frac{k^2}{h^2} \delta$. For $l = k + 1$, we need to prove the results still hold with probability at least $1 - \frac{2l-1}{h^2} \delta$. Toward this goal, we have

$$\boldsymbol{X}_{i,s}^{(l)} = \sum_{0 \leq q \leq l-1} \left[ \boldsymbol{X}_{i,s}^{(q)} + \tau \sigma \left( \sum_{t=1}^{m} \boldsymbol{W}_{q,ts}^{(l)} \bar{\boldsymbol{X}}_{i,t}^{(q)} \right) \right],$$

where $\tau = \frac{1}{\sqrt{m}}$. Then let

$$\boldsymbol{A}_{i,s}^{(lq)} = \sum_{t=1}^{m} \boldsymbol{W}_{q,ts}^{(l)} \bar{\boldsymbol{X}}_{i,t}^{(q)}.$$

Similarly, we can obtain $\boldsymbol{A}_{i,s}^{(lq)}$ is a mean-zero Guassian variable with covariance matrix

$$\mathbb{E}\left[ \boldsymbol{A}_{i,s}^{(lq)} (\boldsymbol{A}_{i,s}^{(lr)})^\top \right] = \delta_{st} \mathcal{W}_{qr}^{(l)} \left( \sum_t \bar{\boldsymbol{X}}_{i,t}^{(q)} (\bar{\boldsymbol{X}}_{j,t}^{(q)})^\top \right) = \delta_{st} \mathcal{W}_{qr}^{(l)} \left( \widehat{\boldsymbol{Q}}_{ij}^{qr} \right).$$

Note that since for convolution networks, each element in the output involves several elements in the input (implemented by the operation $\Phi(\cdot)$), we need to consider this by combining the involved elements. Therefore, we can conclude

$$\widehat{\boldsymbol{Q}}_{ij,ab}^{(ls)} = \mathrm{Tr}\left(\widehat{\boldsymbol{K}}_{ij,S_a^{(l)},S_b^{(s)}}^{(ls)}\right) \ (1 \leq s \leq l)$$

where $\widehat{\boldsymbol{K}}_{ij,ab}^{(ls)}$ denotes the $(a,b)$-th entry in $\widehat{\boldsymbol{K}}_{ij}^{(ls)}$, and $S_a^{(s)} = \{j \mid \boldsymbol{X}_{:,j}^{(s-1)} \in \text{the } a\text{-th patch}\}$. Moreover, we can easily obtain

$$\mathbb{E}\left[\widehat{\boldsymbol{b}}_i^{(l)}\right] = \sum_{t=1}^{l-1}\left(\boldsymbol{\alpha}_{t,2}^{(l)}\widehat{\boldsymbol{b}}_i^{(t)} + \tau\boldsymbol{\alpha}_{t,3}^{(l)}\mathbb{E}_{\widehat{\boldsymbol{M}}_{tt}^{(l)}}\sigma(\widehat{\boldsymbol{M}}_{tt}^{(l)})\right).$$

In this way, we can further obtain

$$\widehat{\boldsymbol{A}}_{tq}^{(l)} = \begin{bmatrix} \mathcal{W}_{tq}^{(l)}(\widehat{\boldsymbol{Q}}_{ii}^{(tq)}), \mathcal{W}_{tq}^{(l)}(\widehat{\boldsymbol{Q}}_{ij}^{(tq)}) \\ \mathcal{W}_{tq}^{(l)}(\widehat{\boldsymbol{Q}}_{ji}^{(tq)}), \mathcal{W}_{tq}^{(l)}(\widehat{\boldsymbol{Q}}_{jj}^{(tq)}) \end{bmatrix}, \quad (\widehat{\boldsymbol{M}}_{tq}^{(l)}, \widehat{\boldsymbol{N}}_{tq}^{(l)}) \sim \mathcal{N}\left(\boldsymbol{0}, \widehat{\boldsymbol{A}}_{tq}^{(l)}\right), \qquad (0 \leq t, q \leq l-1),$$

$$\mathbb{E}\left[\widehat{\boldsymbol{K}}_{ij}^{(ls)}\right] = \sum_{t=1}^{l-1}\sum_{q=1}^{s-1}\left[\boldsymbol{\alpha}_{t,2}^{(l)}\boldsymbol{\alpha}_{q,2}^{(s)}\widehat{\boldsymbol{K}}_{ij}^{(tq)} + \tau\mathbb{E}_{(\widehat{\boldsymbol{M}}_{tq}^{(l)},\widehat{\boldsymbol{N}}_{tq}^{(l)})}\left(\boldsymbol{\alpha}_{t,3}^{(l)}\boldsymbol{\alpha}_{q,2}^{(s)}\sigma(\widehat{\boldsymbol{M}}_{tq}^{(l)})(\widehat{\boldsymbol{b}}_j^{(q)})^\top + \boldsymbol{\alpha}_{t,2}^{(l)}\boldsymbol{\alpha}_{q,3}^{(s)}\widehat{\boldsymbol{b}}_i^{(t)}\sigma(\widehat{\boldsymbol{N}}_{tq}^{(l)})^\top\right.$$

$$\left.+\tau\boldsymbol{\alpha}_{t,3}^{(l)}\boldsymbol{\alpha}_{q,3}^{(s)}\sigma(\widehat{\boldsymbol{M}}_{tq}^{(l)})\sigma(\widehat{\boldsymbol{N}}_{tq}^{(l)})^\top\right)\right] \in \mathbb{R}^{p\times p}.$$

Then we also apply the concentration inequality and obtain that for $1 \leq s \leq l$

$$\mathbb{P}\left(\max_{ij}\left\|\frac{1}{m}\sum_{t=1}^{m}\boldsymbol{X}_{i,t}^{(l)}(\boldsymbol{X}_{j,t}^{(s)})^T - \mathbb{E}\widehat{\boldsymbol{K}}_{ij}^{(ls)}\right\|_\infty \leq \sqrt{\frac{16(1+2C_1^2/\sqrt{\pi})M^2\log(4n^2p^2h^2/\delta))}{m}}\right) \geq 1 - \delta/h^2$$

where we use $\|\boldsymbol{X}_{i,t}^{(0)}(\boldsymbol{X}_{j,t}^{(0)})^\top\|_2 \leq \|\boldsymbol{X}_{i,t}^{(0)}(\boldsymbol{X}_{j,t}^{(0)})^\top\|_F \leq 0.5(\|\boldsymbol{X}_{i,t}^{(0)}\|_F^2 + \|\boldsymbol{X}_{j,t}^{(0)\top}\|_F^2) \leq c_{x0}^2$, $M_1 = 1 + 100\max_{i,j,s,t,l}|\mathcal{W}^l(\bar{\boldsymbol{K}}_{ij}^{(l-1)})_{st}|$. Similarly, we can prove

$$\mathbb{P}\left(\left\|\frac{1}{m}\sum_{t=1}^{m}\boldsymbol{X}_{i,t}^{(l)} - \mathbb{E}\widehat{\boldsymbol{b}}_i^{(l)}\right\|_\infty \leq \sqrt{\frac{2C_1M\log(2nph/\delta))}{m}}\right) \geq 1 - \delta/h^2.$$

According to the definition

$$\widehat{\boldsymbol{K}}_{ij}^{(ls)} = \frac{1}{m}\sum_{t=1}^{m}\boldsymbol{X}_{i,t}^{(l)}(\boldsymbol{X}_{j,t}^{(s)})^\top, \qquad \widehat{\boldsymbol{b}}_i^{(l)} = \frac{1}{m}\sum_{t=1}^{m}\boldsymbol{X}_{i,t}^{(l)}.$$

we have

$$\left\|\frac{1}{m}\sum_{t=1}^{m}\boldsymbol{X}_{i,t}^{(l)}(\boldsymbol{X}_{j,t}^{(s)})^\top - \boldsymbol{K}_{ij}^{(ls)}\right\|_\infty \leq \left\|\frac{1}{m}\sum_{t=1}^{m}\boldsymbol{X}_{i,t}^{(l)}(\boldsymbol{X}_{j,t}^{(s)})^\top - \mathbb{E}\widehat{\boldsymbol{K}}_{ij}^{(ls)}\right\|_\infty + \left\|\mathbb{E}\widehat{\boldsymbol{K}}_{ij}^{(ls)} - \boldsymbol{K}_{ij}^{(ls)}\right\|_\infty,$$

$$\left\|\frac{1}{m}\sum_{t=1}^{m}\boldsymbol{X}_{i,t}^{(l)} - \boldsymbol{b}_i^{(l)}\right\|_\infty \leq \left\|\frac{1}{m}\sum_{t=1}^{m}\boldsymbol{X}_{i,t}^{(l)} - \mathbb{E}\widehat{\boldsymbol{b}}_i^{(l)}\right\|_\infty + \left\|\mathbb{E}\widehat{\boldsymbol{b}}_i^{(l)} - \boldsymbol{b}_i^{(l)}\right\|_\infty.$$

Then we only need to bound

$$\left\|\mathbb{E}\widehat{\boldsymbol{K}}_{ij}^{(ls)} - \boldsymbol{K}_{ij}^{(ls)}\right\|_\infty \quad \text{and} \quad \left\|\mathbb{E}\widehat{\boldsymbol{b}}_i^{(l)} - \boldsymbol{b}_i^{(l)}\right\|_\infty.$$

In the following content, we bound these two terms in turn. To begin with, we have

$$\left\|\mathbb{E}\widehat{\boldsymbol{K}}_{ij}^{(ls)} - \boldsymbol{K}_{ij}^{(ls)}\right\|_\infty = \left\|\mathrm{Tr}\left(\widehat{\boldsymbol{Q}}_{ij,S_a^{(s)},S_b^{(ls)}}^{(ls)}\right) - \mathrm{Tr}\left(\boldsymbol{Q}_{ij,S_a^{(s)},S_b^{(ls)}}^{(ls)}\right)\right\|_\infty \leq \left\|\widehat{\boldsymbol{Q}}_{ij}^{(l)} - \boldsymbol{Q}_{ij}^{(l)}\right\|_\infty$$

$$\leq \sum_{t=1}^{l-1}\sum_{q=1}^{s-1}\left[\boldsymbol{\alpha}_{t,2}^{(l)}\boldsymbol{\alpha}_{q,2}^{(s)}\left\|\widehat{\boldsymbol{K}}_{ij}^{(tq)} - \boldsymbol{K}_{ij}^{(tq)}\right\|_\infty\right.$$

$$+ \tau\boldsymbol{\alpha}_{t,3}^{(l)}\boldsymbol{\alpha}_{q,2}^{(s)}\left\|\mathbb{E}_{((\widehat{\boldsymbol{M}}^{(tq)},\widehat{\boldsymbol{N}}^{(tq)}))}\sigma(\widehat{\boldsymbol{M}}^{(tq)})(\widehat{\boldsymbol{b}}_j^{(q)})^\top - \mathbb{E}_{((\boldsymbol{M}^{(tq)},\boldsymbol{N}^{(tq)}))}\sigma(\boldsymbol{M}^{(tq)})(\boldsymbol{b}_j^{(q)})^\top\right\|_\infty$$

$$+ \tau\boldsymbol{\alpha}_{t,2}^{(l)}\boldsymbol{\alpha}_{q,3}^{(s)}\left\|\mathbb{E}_{((\widehat{\boldsymbol{M}}^{(tq)},\widehat{\boldsymbol{N}}^{(tq)}))}\widehat{\boldsymbol{b}}_i^{(t)}\sigma(\widehat{\boldsymbol{N}}^{(tq)})^\top - \mathbb{E}_{((\boldsymbol{M}^{(tq)},\boldsymbol{N}^{(tq)}))}\boldsymbol{b}_i^{(t)}\sigma(\boldsymbol{N}^{(tq)})^\top\right\|_\infty$$

$$+\tau\boldsymbol{\alpha}_{t,3}^{(l)}\boldsymbol{\alpha}_{q,3}^{(s)}\left\|\mathbb{E}_{((\widehat{\boldsymbol{M}}^{(tq)},\widehat{\boldsymbol{N}}^{(tq)}))}\sigma(\widehat{\boldsymbol{M}}^{(tq)})\sigma(\widehat{\boldsymbol{N}}^{(tq)})^\top - \mathbb{E}_{((\boldsymbol{M}^{(tq)},\boldsymbol{N}^{(tq)}))}\sigma(\boldsymbol{M}^{(tq)})\sigma(\boldsymbol{N}^{(tq)})^\top\right\|_\infty\right]$$

Then we bound

$$\left\|\mathbb{E}_{((\widehat{\boldsymbol{M}}^{(tq)},\widehat{\boldsymbol{N}}^{(tq)}))}\sigma(\widehat{\boldsymbol{M}}^{(tq)})(\widehat{\boldsymbol{b}}_j^{(q)})^\top - \mathbb{E}_{((\boldsymbol{M}^{(tq)},\boldsymbol{N}^{(tq)}))}\sigma(\boldsymbol{M}^{(tq)})(\boldsymbol{b}_j^{(q)})^\top\right\|_\infty$$

$$=\left\|\mathbb{E}_{((\boldsymbol{M},\boldsymbol{N})\sim\widehat{\boldsymbol{A}}^{(tq)})}\sigma(\boldsymbol{M})(\widehat{\boldsymbol{b}}_j^{(q)})^\top - \mathbb{E}_{((\boldsymbol{M},\boldsymbol{N})\sim\boldsymbol{A}^{(tq)})}\sigma(\boldsymbol{M})(\boldsymbol{b}_j^{(q)})^\top\right\|_\infty$$

$$\leq\left\|\mathbb{E}_{((\boldsymbol{M},\boldsymbol{N})\sim\widehat{\boldsymbol{A}}^{(tq)})}\sigma(\boldsymbol{M})(\widehat{\boldsymbol{b}}_j^{(q)}-\boldsymbol{b}_j^{(q)})^\top\right\|_\infty + \left\|\left[\mathbb{E}_{((\boldsymbol{M},\boldsymbol{N})\sim\widehat{\boldsymbol{A}}^{(tq)})}\sigma(\boldsymbol{M})-\mathbb{E}_{((\boldsymbol{M},\boldsymbol{N})\sim\boldsymbol{A}^{(tq)})}\sigma(\boldsymbol{M})\right](\boldsymbol{b}_j^{(q)})^\top\right\|_\infty$$

Next, we bound the above inequality by bound each term:

$$\left\|\left[\mathbb{E}_{((\boldsymbol{M},\boldsymbol{N})\sim\widehat{\boldsymbol{A}}^{(tq)})}\sigma(\boldsymbol{M})-\mathbb{E}_{((\boldsymbol{M},\boldsymbol{N})\sim\boldsymbol{A}^{(tq)})}\sigma(\boldsymbol{M})\right](\boldsymbol{b}_j^{(q)})^\top\right\|_\infty$$

$$\leq\max_i\|\boldsymbol{b}_j^{(q)}\|_\infty(\sigma(0)+\sup_x\sigma'(x))\|\widehat{\boldsymbol{A}}^{(tq)}-\boldsymbol{A}^{(tq)}\|_\infty$$

$$\leq c_1 c_2 c_3\|\widehat{\boldsymbol{Q}}_{ij}^{(tq)}-\boldsymbol{Q}_{ij}^{(tq)}\|_\infty$$

$$=c_1 c_2 c_3\max_{a,b}\left\|\mathrm{Tr}\left(\widehat{\boldsymbol{K}}_{ij,S_a^{(l)},S_b^{(s)}}^{(ls)}\right)-\mathrm{Tr}\left(\boldsymbol{K}_{ij,S_a^{(l)},S_b^{(s)}}^{(ls)}\right)\right\|_\infty$$

$$\leq c_1 c_2 c_3 q\left\|\widehat{\boldsymbol{K}}_{ij}^{(l)}-\boldsymbol{K}_{ij}^{(l)}\right\|_\infty,$$

where $c_1=\max_l 1+\|\mathcal{W}_{tq}^{(l)}\|_{L^\infty\to L^\infty}$, $c_2=\sigma(0)+\sup_x\sigma'(x)$, $c_3=\max_{i,q}\|\boldsymbol{b}_i^{(q)}\|_\infty$. Similarly, we can bound

$$\left\|\mathbb{E}_{((\boldsymbol{M},\boldsymbol{N})\sim\widehat{\boldsymbol{A}}^{(tq)})}\sigma(\boldsymbol{M})(\widehat{\boldsymbol{b}}_j^{(q)}-\boldsymbol{b}_j^{(q)})^\top\right\|_\infty \leq c_2\sqrt{c_1 c_4}\|\boldsymbol{b}_j^{(q)}-\widehat{\boldsymbol{b}}_j^{(q)}\|_\infty$$

where $c_4=\max_{ij}\|\widehat{\boldsymbol{Q}}_{ij}^{(tq)})\|_\infty\leq q\max_{ij}\|\widehat{\boldsymbol{K}}_{ij}^{(tq)})\|_\infty\leq qc_{x0}^2$ and $1\leq q\leq l-1$. Therefore we have

$$\left\|\mathbb{E}_{((\widehat{\boldsymbol{M}}^{(tq)},\widehat{\boldsymbol{N}}^{(tq)}))}\sigma(\widehat{\boldsymbol{M}}^{(tq)})(\widehat{\boldsymbol{b}}_j^{(q)})^\top - \mathbb{E}_{((\boldsymbol{M}^{(tq)},\boldsymbol{N}^{(tq)}))}\sigma(\boldsymbol{M}^{(tq)})(\boldsymbol{b}_j^{(q)})^\top\right\|_\infty$$

$$=(c_1 c_2 c_3 q+c_2\sqrt{c_1 c_4})\max\left(\|\widehat{\boldsymbol{K}}_{ij}^{(tq)}-\boldsymbol{K}_{ij}^{(tq)}\|_\infty,\|\boldsymbol{b}_j^{(q)}-\widehat{\boldsymbol{b}}_j^{(q)}\|_\infty\right).$$

By using the same method, we can upper bound

$$\left\|\mathbb{E}_{((\widehat{\boldsymbol{M}}^{(tq)},\widehat{\boldsymbol{N}}^{(tq)}))}\widehat{\boldsymbol{b}}_i^{(t)}\sigma(\widehat{\boldsymbol{N}}^{(tq)})^\top - \mathbb{E}_{((\boldsymbol{M}^{(tq)},\boldsymbol{N}^{(tq)}))}\boldsymbol{b}_i^{(t)}\sigma(\boldsymbol{N}^{(tq)})^\top\right\|_\infty$$

$$=(c_1 c_2 c_3 q+c_2\sqrt{c_1 c_4})\max\left(\|\widehat{\boldsymbol{K}}_{ij}^{(tq)}-\boldsymbol{K}_{ij}^{(tq)}\|_\infty,\|\boldsymbol{b}_j^{(q)}-\widehat{\boldsymbol{b}}_j^{(q)}\|_\infty\right).$$

Next, we can upper bound

$$\left\|\mathbb{E}_{((\widehat{\boldsymbol{M}}^{(tq)},\widehat{\boldsymbol{N}}^{(tq)}))}\sigma(\widehat{\boldsymbol{M}}^{(tq)})\sigma(\widehat{\boldsymbol{N}}^{(tq)})^\top - \mathbb{E}_{((\boldsymbol{M}^{(tq)},\boldsymbol{N}^{(tq)}))}\sigma(\boldsymbol{M}^{(tq)})\sigma(\boldsymbol{N}^{(tq)})^\top\right\|_\infty$$

$$=\left\|\mathbb{E}_{((\boldsymbol{M},\boldsymbol{N})\sim\widehat{\boldsymbol{A}}^{(tq)})}\sigma(\boldsymbol{M}^{(tq)})\sigma(\boldsymbol{N}^{(tq)})^\top - \mathbb{E}_{((\boldsymbol{M},\boldsymbol{N})\sim\boldsymbol{A}^{(tq)})}\sigma(\boldsymbol{M}^{(tq)})\sigma(\boldsymbol{N}^{(tq)})^\top\right\|_\infty$$

$$\leq c_\sigma\|\widehat{\boldsymbol{A}}^{(tq)}-\boldsymbol{A}^{(tq)}\|_\infty\leq c_\sigma c_1\|\widehat{\boldsymbol{Q}}_{ij}^{(tq)})-\bar{\boldsymbol{Q}}_{ij}^{(tq)})\|_\infty\leq c_\sigma c_1 q\|\widehat{\boldsymbol{K}}_{ij}^{(tq)})-\bar{\boldsymbol{K}}_{ij}^{(tq)})\|_\infty,$$

where $c_\sigma$ is a constant that only depends on $\sigma$. Combing all results yields

$$\left\|\mathbb{E}\widehat{\boldsymbol{K}}_{ij}^{(ls)}-\boldsymbol{K}_{ij}^{(ls)}\right\|_\infty$$

$$\leq\sum_{t=1}^{l-1}\sum_{q=1}^{s-1}\Big[(\boldsymbol{\alpha}_{t,2}^{(l)}\boldsymbol{\alpha}_{q,2}^{(s)}+\tau^2\boldsymbol{\alpha}_{t,3}^{(l)}\boldsymbol{\alpha}_{q,3}^{(s)}c_\sigma c_1 q)\|\widehat{\boldsymbol{K}}_{ij}^{(tq)})-\boldsymbol{K}_{ij}^{(tq)})\|_\infty$$

$$+\tau(\boldsymbol{\alpha}_{t,2}^{(l)}\boldsymbol{\alpha}_{q,2}^{(s)}+\boldsymbol{\alpha}_{t,3}^{(l)}\boldsymbol{\alpha}_{q,2}^{(s)})(c_1 c_2 c_3 q+c_2\sqrt{c_1 c_4})\max\left(\|\widehat{\boldsymbol{K}}_{ij}^{(tq)}-\boldsymbol{K}_{ij}^{(tq)}\|_\infty,\|\boldsymbol{b}_j^{(q)}-\widehat{\boldsymbol{b}}_j^{(q)}\|_\infty\right)\Big]$$

$$\leq c\max_{1\leq t\leq l-1,1\leq q\leq l-1}\left(\|\widehat{\boldsymbol{K}}_{ij}^{(tq)}-\boldsymbol{K}_{ij}^{(tq)}\|_\infty,\|\boldsymbol{b}_j^{(q)}-\widehat{\boldsymbol{b}}_j^{(q)}\|_\infty\right)$$

where $c_l=\sum_{t=1}^{l-1}\sum_{q=1}^{s-1}\left[\boldsymbol{\alpha}_{t,2}^{(l)}\boldsymbol{\alpha}_{q,2}^{(s)}+\tau^2\boldsymbol{\alpha}_{t,3}^{(l)}\boldsymbol{\alpha}_{q,3}^{(s)}c_\sigma c_1 q+\tau(\boldsymbol{\alpha}_{t,2}^{(l)}\boldsymbol{\alpha}_{q,2}^{(s)}+\boldsymbol{\alpha}_{t,3}^{(l)}\boldsymbol{\alpha}_{q,2}^{(s)})(c_1 c_2 c_3 q+c_2\sqrt{c_1 c_4})\right]$.
Since we have assumed that with probability $1-(l-1)^2\delta/h^2$ for $0\leq t\leq l-1,0\leq s\leq l-1$, it holds

$$\max\left(\left\|\frac{1}{m}\sum_{s=1}^m(\boldsymbol{X}_{i,s}^{(t)})^\top\boldsymbol{X}_{j,s}^{(q)}-\boldsymbol{K}_{ij}^{(tq)}\right\|_\infty,\left\|\frac{1}{m}\sum_{s=1}^m\boldsymbol{X}_{i,s}^{(t)}-\boldsymbol{b}_i^{(t)}\right\|_\infty\right)\leq C_{l-1}\sqrt{\frac{\log(n^2p^2h^2/\delta)}{m}},$$

where $C$ is a constant. Then with probability $1 - (l-1)^2\delta/h^2$, we have for all $0 \le s \le l$

$$\left\|\mathbb{E}\widehat{\boldsymbol{K}}_{ij}^{(ls)} - \boldsymbol{K}_{ij}^{(ls)}\right\|_\infty \le c_l C_{l-1}\sqrt{\frac{\log(n^2 p^2 h^2/\delta)}{m}}.$$

Thus, with probability $(1 - (l-1)^2\delta/h^2)(1 - \delta/h^2) \ge 1 - l^2\delta/h^2 \ge 1 - \delta$, we have for all for $0 \le t \le h, 0 \le s \le h$

$$\left\|\frac{1}{m}\sum_{s=1}^m (\boldsymbol{X}_{i,s}^{(t)})^\top \boldsymbol{X}_{j,s}^{(q)} - \boldsymbol{K}_{ij}^{(tq)}\right\|_\infty \le C\sqrt{\frac{\log(n^2 p^2 h^2/\delta)}{m}},$$

where $C = C_0 \prod_{l=1}^h c_l$ is a constant.

Now we consider to bound

$$\left\|\mathbb{E}\widehat{\boldsymbol{b}}_i^{(l)} - \boldsymbol{b}_i^{(l)}\right\|_\infty$$

$$= \left\|\sum_{t=1}^{l-1}\left(\boldsymbol{\alpha}_{t,2}^{(l)}(\widehat{\boldsymbol{b}}_i^{(t)} - \boldsymbol{b}_i^{(t)}) + \tau\boldsymbol{\alpha}_{t,3}^{(l)}\left(\mathbb{E}_{\boldsymbol{M}\sim\widehat{\boldsymbol{A}}^{lt}}\sigma(\boldsymbol{M}) - \mathbb{E}_{\boldsymbol{M}\sim\boldsymbol{A}^{lt}}\sigma(\boldsymbol{M})\right)\right)\right\|_\infty$$

$$\le \sum_{t=1}^{l-1}\left(\boldsymbol{\alpha}_{t,2}^{(l)}\left\|\widehat{\boldsymbol{b}}_i^{(t)} - \boldsymbol{b}_i^{(t)}\right\|_\infty + \tau\boldsymbol{\alpha}_{t,3}^{(l)}\left\|\left(\mathbb{E}_{\boldsymbol{M}\sim\widehat{\boldsymbol{A}}^{(l-1)t}}\sigma(\boldsymbol{M}) - \mathbb{E}_{\boldsymbol{M}\sim\boldsymbol{A}^{(l-1)t}}\sigma(\boldsymbol{M})\right)\right\|_\infty\right)$$

$$\le \sum_{t=1}^{l-1}\left(\boldsymbol{\alpha}_{t,2}^{(l)}\left\|\widehat{\boldsymbol{b}}_i^{(t)} - \boldsymbol{b}_i^{(t)}\right\|_\infty + \tau\boldsymbol{\alpha}_{t,3}^{(l)}c_\sigma\left\|\widehat{\boldsymbol{A}}^{(l-1)t} - \boldsymbol{A}^{(l-1)t}\right\|_\infty\right)$$

$$\le \sum_{t=1}^{l-1}\left(\boldsymbol{\alpha}_{t,2}^{(l)}\left\|\widehat{\boldsymbol{b}}_i^{(t)} - \boldsymbol{b}_i^{(t)}\right\|_\infty + \tau\boldsymbol{\alpha}_{t,3}^{(l)}c_\sigma\left\|\widehat{\boldsymbol{Q}}^{(l-1)t} - \boldsymbol{Q}^{(l-1)t}\right\|_\infty\right)$$

$$\le \sum_{t=1}^{l-1}\left(\boldsymbol{\alpha}_{t,2}^{(l)} + \tau\boldsymbol{\alpha}_{t,3}^{(l)}c_\sigma c_1 q\right)\max\left(\left\|\widehat{\boldsymbol{b}}_i^{(t)} - \boldsymbol{b}_i^{(t)}\right\|_\infty, \left\|\widehat{\boldsymbol{K}}^{(l-1)t} - \boldsymbol{K}^{(l-1)t}\right\|_\infty\right)$$

where $c_l' = \sum_{t=1}^{l-1}\left(\boldsymbol{\alpha}_{t,2}^{(l)} + \tau\boldsymbol{\alpha}_{t,3}^{(l)}c_\sigma c_1 q\right)$. Then with probability $(1 - (l-1)^2\delta/h)(1 - \delta/h) \ge 1 - \delta$, we have for all for $0 \le t \le h$

$$\left\|\frac{1}{m}\sum_{s=1}^m \boldsymbol{X}_{i,s}^{(t)} - \boldsymbol{b}_i^{(t)}\right\|_\infty \le C\sqrt{\frac{\log(n^2 p^2 h^2/\delta)}{m}},$$

where $C = C_0\prod_{l=1}^h \max(c_l, c_l')$ is a constant. The proof is completed. $\qquad\square$

### C.4.2 Proof of Lemma 23

*Proof.* For brevity, here we just use $\boldsymbol{X}_i^{(s)}$, $\boldsymbol{W}_s^{(h)}$, $\boldsymbol{W}_h$, $\bar{\boldsymbol{X}}_i^{(s)})$ to respectively denote $Xmii(s)i(0)$ $\boldsymbol{W}_s^{(h)}(0)$, $\boldsymbol{W}_h(0)$, $\Phi(\boldsymbol{X}_i^{(s)})$, since here we only involve the initialization and does not update the variables. Let $\bar{\boldsymbol{X}}_{i,t}^{(s)}) = (\bar{\boldsymbol{X}}_{i,:t}^{(s)})^\top$ and $\boldsymbol{Z}_{i,tr} = (\boldsymbol{W}_{s,:r}^{(h)})^\top \bar{\boldsymbol{X}}_{i,t}^{(s)}$. Firstly according to the definition, we have

$$\boldsymbol{G}_{ij}^{hs}(0) = \left\langle \frac{\partial\ell_i}{\partial\boldsymbol{W}_s^{(h)}(0)}, \frac{\partial\ell_j}{\partial\boldsymbol{W}_s^{(h)}(0)}\right\rangle$$

$$= (\boldsymbol{\alpha}_{s,3}^{(h)}\tau)^2\left\langle \Phi(\boldsymbol{X}_i^{(s)})\left(\sigma'\left(\boldsymbol{W}_s^{(l)}\Phi(\boldsymbol{X}_i^{(s)})\right) \odot \boldsymbol{W}_h\right)^\top, \Phi(\boldsymbol{X}_j^{(s)})\left(\sigma'\left(\boldsymbol{W}_s^{(l)}\Phi(\boldsymbol{X}_j^{(s)})\right) \odot \boldsymbol{W}_h\right)^\top\right\rangle$$

$$= (\boldsymbol{\alpha}_{s,3}^{(h)}\tau)^2\sum_{t=1}^p\sum_{q=1}^p \bar{\boldsymbol{X}}_{i,t}^{(s)})(\bar{\boldsymbol{X}}_{j,q}^{(s)})^\top\sum_{r=1}^m \boldsymbol{W}_{h,tr}\boldsymbol{W}_{h,qr}\sigma'(\boldsymbol{Z}_{i,tr})\sigma'(\boldsymbol{Z}_{j,qr}).$$

Then by taking expectation on $\boldsymbol{W} \sim \mathcal{N}(0, \boldsymbol{I})$ and $\boldsymbol{U} \sim \mathcal{N}(0, \boldsymbol{I})$, we have

$$\boldsymbol{G}_{ij}^{hs}(0) = (\boldsymbol{\alpha}_{s,3}^{(h)}\tau)^2\sum_{t=1}^p\sum_{q=1}^p \bar{\boldsymbol{X}}_{i,t}^{(s)})(\bar{\boldsymbol{X}}_{j,q}^{(s)})^\top\sum_{r=1}^m \mathbb{E}_{\boldsymbol{W}_h}\left[\boldsymbol{W}_{h,tr}\boldsymbol{W}_{h,qr}\right]\mathbb{E}_{\boldsymbol{W}_s^{(h)}}\left[\sigma'(\boldsymbol{Z}_{i,tr})\sigma'(\boldsymbol{Z}_{j,qr})\right]$$

$$= (\boldsymbol{\alpha}_{s,3}^{(h)}\tau)^2\sum_{t=1}^p \bar{\boldsymbol{X}}_{i,t}^{(s)})(\bar{\boldsymbol{X}}_{j,t}^{(s)})^\top\sum_{r=1}^m \mathbb{E}_{\boldsymbol{W}_s^{(h)}}\left[\sigma'(\boldsymbol{Z}_{i,tr})\sigma'(\boldsymbol{Z}_{j,qr})\right]$$

(20)

where ① holds since $\mathbb{E}_{\boldsymbol{W}_h}[\boldsymbol{W}_{h,tr}\boldsymbol{W}_{h,qr}] = 1$ if $t = q$ and $\mathbb{E}_{\boldsymbol{W}_h}[\boldsymbol{W}_{h,tr}\boldsymbol{W}_{h,qr}] = 0$ if $t \neq q$.

$$\boldsymbol{Z}_{i,r} = \sum_{t=1}^{m} (\boldsymbol{W}_{s,tr}^{(h)})^{\top} \bar{\boldsymbol{X}}_{i,t}^{(s)}).$$

Since the convolution parameter $\boldsymbol{W}_s^{(h)}$ satisfies Gaussian distribution, $\boldsymbol{Z}_{i,r}$ is a mean-zero Guassian variable with covariance matrix as follows

$$
\begin{aligned}
\mathbb{E}\left[(\boldsymbol{Z}_{i,r})^{\top}\boldsymbol{Z}_{j,q}\right] &= \mathbb{E}\sum_{t,t'}(\boldsymbol{W}_{s,t}^{(h)})^{\top}\bar{\boldsymbol{X}}_{i,t}^{(s)}(\bar{\boldsymbol{X}}_{j,t'}^{(s)})^{\top}(\boldsymbol{W}_{s,t'q}^{(h)})^{\top} = \delta_{st}\mathcal{W}^{(hs)}\left(\sum_{t}\bar{\boldsymbol{X}}_{i,t}^{(s)}(\bar{\boldsymbol{X}}_{j,t}^{(s)})^{\top}\right) \\
&= \delta_{st}\mathcal{W}^{(hs)}\left(\widehat{\boldsymbol{Q}}_{ij}^{(s)}\right),
\end{aligned}
\tag{21}
$$

where $\delta_{st}$ is a random variable with $\delta_{st} = \pm 1$ with both probability 0.5, and

$$\widehat{\boldsymbol{K}}_{ij}^{(ss)} = \frac{1}{m}\sum_{t=1}^{m}\boldsymbol{X}_{i,t}^{(s)}(\boldsymbol{X}_{j,t}^{(s)})^{\top}, \qquad \widehat{\boldsymbol{Q}}_{ij}^{(ss)} = \frac{1}{m}\sum_{t=1}^{m}\bar{\boldsymbol{X}}_{i,t}^{(s)}(\bar{\boldsymbol{X}}_{j,t}^{(s)})^{\top}.$$

According to this definition, we actually have

$$\widehat{\boldsymbol{Q}}_{ij,ab}^{(ss)} = \mathrm{Tr}\left(\widehat{\boldsymbol{K}}_{ij,S_a^{(s)},S_b^{(s)}}^{(ss)}\right),$$

where $\widehat{\boldsymbol{K}}_{ij}^{(ss)} \in \mathbb{R}^{p\times p}$, $\widehat{\boldsymbol{Q}}_{ij,ab}^{(ss)}$ denotes the $(a,b)$-th entry in $\widehat{\boldsymbol{Q}}_{ij}^{(ss)}$, and $S_a^{(s)} = \{j \mid \boldsymbol{X}_{:,j}^{(s-1)} \in$ the $a-$ th patch for convolution$\}$. Then according to the following definitions

$$
\begin{aligned}
\widehat{\boldsymbol{A}}^{(s)} &= \begin{bmatrix} \mathcal{W}_{ss}^{(h)}(\widehat{\boldsymbol{Q}}_{ii}^{(ss)}), \mathcal{W}_{ss}^{(h)}(\widehat{\boldsymbol{Q}}_{ij}^{(ss)}) \\ \mathcal{W}_{ss}^{(h)}(\widehat{\boldsymbol{Q}}_{ji}^{(ss)}), \mathcal{W}_{ss}^{(h)}(\widehat{\boldsymbol{Q}}_{jj}^{(ss)}) \end{bmatrix}, \\
\widehat{\boldsymbol{Q}}_{ij,ab}^{(s)} &= \widehat{\boldsymbol{Q}}_{ij,ab}^{(ss)}\mathbb{E}_{((\boldsymbol{M},\boldsymbol{N})\sim\widehat{\boldsymbol{A}}^{(s)})}\sigma'(\boldsymbol{M})\sigma'(\boldsymbol{N})^{\top}, \qquad \widehat{\boldsymbol{K}}_{ij,ab}^{(s)} = \mathrm{Tr}\left(\widehat{\boldsymbol{Q}}_{ij}^{(s)}\right), \ (s=0,h-1).
\end{aligned}
$$

and Eqns. (20) and (21), we have

$$\mathbb{E}\left[\boldsymbol{G}_{ij}^{hs}(0)\right] = (\boldsymbol{\alpha}_{s,3}^{(h)})^2\widehat{\boldsymbol{K}}_{ij}^{(s)}, \qquad \mathbb{E}\left[\boldsymbol{G}^{hs}(0)\right] = (\boldsymbol{\alpha}_{s,3}^{(h)})^2\widehat{\boldsymbol{K}}^{(s)}.$$

In this way, we can apply the Hoeffding inequality and obtain that if $m \geq \mathcal{O}\left(\frac{n^2\log(n/\delta)}{\lambda^2}\right)$

$$\left\|\boldsymbol{G}^{hs}(0) - (\boldsymbol{\alpha}_{s,3}^{(h)})^2\widehat{\boldsymbol{K}}^{(s)}\right\|_{\mathrm{op}} \leq \frac{\lambda}{8}.$$

On the other hand, Lemma 22 shows that with probability at least $1 - \delta$

$$\left\|\widehat{\boldsymbol{K}}_{ij}^{(ss)} - \boldsymbol{K}_{ij}^{(ss)}\right\|_{\infty} \leq C\sqrt{\frac{\log(n^2p^2h^2/\delta)}{m}} \overset{①}{\leq} \frac{C_3\lambda}{n},$$

where ① holds by setting $m \geq \mathcal{O}\left(\frac{C_3^2 n^2\log(n^2p^2h^2/\delta)}{\lambda^2}\right)$. Moreover, Lemma 10 shows

$$\frac{1}{c_{x0}} \leq \|\boldsymbol{X}^{(l)}(0)\|_F \leq c_{x0}.$$

where $c_{x0} \geq 1$ is a constant. So $\|\widehat{\boldsymbol{K}}_{ij}^{(ss)}\|_{\infty}$ is upper bounded by $c_{x0}^2$.

Next, Lemma 7 shows if each diagonal entry in $\boldsymbol{A}$ and $\boldsymbol{B}$ is upper bounded by c and lower upper bounded by $1/c$, then

$$|g(\boldsymbol{A}) - g(\boldsymbol{B})| \leq c\|\boldsymbol{A} - \boldsymbol{B}\|_F \leq 2C_1\|\boldsymbol{A} - \boldsymbol{B}\|_{\infty},$$

where $g(\boldsymbol{A}) = \mathbb{E}_{(u,v)\sim\mathcal{N}(0,\boldsymbol{A})}\sigma(u)\sigma(v)$, $C_1$ is a constant that only depends on $c$ and the Lipschitz and smooth parameter of $\sigma(\cdot)$. By applying this lemma, we can obtain

$$|\widehat{\boldsymbol{Q}}^{(ss)}_{ij,rq}\mathbb{E}_{(\boldsymbol{M},\boldsymbol{N})\sim\widehat{\boldsymbol{A}}^{(s)}}\left[\sigma'(\boldsymbol{M}_r))\sigma'(\boldsymbol{N}_q)\right] - \boldsymbol{Q}^{(ss)}_{ij,rq}\mathbb{E}_{(\boldsymbol{M},\boldsymbol{N})\sim\bar{\boldsymbol{A}}^{(s)}}\left[\sigma'(\boldsymbol{M}_r))\sigma'(\boldsymbol{N}_q)\right]|$$

$$\leq|\widehat{\boldsymbol{Q}}^{(ss)}_{ij,rq}\left(\mathbb{E}_{(\boldsymbol{M},\boldsymbol{N})\sim\widehat{\boldsymbol{A}}^{(s)}}\left[\sigma'(\boldsymbol{M}_r))\sigma'(\boldsymbol{N}_q)\right] - \mathbb{E}_{(\boldsymbol{M},\boldsymbol{N})\sim\bar{\boldsymbol{A}}^{(s)}}\left[\sigma'(\boldsymbol{M}_r))\sigma'(\boldsymbol{N}_q)\right]\right)|$$

$$+|(\widehat{\boldsymbol{Q}}^{(ss)}_{ij,rq} - \boldsymbol{Q}^{(ss)}_{ij,rq})\mathbb{E}_{(\boldsymbol{M},\boldsymbol{N})\sim\bar{\boldsymbol{A}}^{(s)}}\left[\sigma'(\boldsymbol{M}_r))\sigma'(\boldsymbol{N}_q)\right]|$$

$$\leq C_1 c_{x0}^2|\widehat{\boldsymbol{A}}^{(s)} - \boldsymbol{A}^{(s)}| + \mu^2|\widehat{\boldsymbol{Q}}^{(ss)}_{ij,rq} - \boldsymbol{Q}^{(ss)}_{ij,rq}|$$

$$\leq C_1 C_2 c_{x0}^2 \max_{i,j}|\widehat{\boldsymbol{Q}}^{(ss)}_{ij,rq} - \bar{\boldsymbol{Q}}^{(ss)}_{ij,rq}| + \mu^2|\widehat{\boldsymbol{Q}}^{(ss)}_{ij,rq} - \boldsymbol{Q}^{(ss)}_{ij,rq}|$$

$$\leq(C_1 C_2 c_{x0}^2 + \mu^2)\|\widehat{\boldsymbol{Q}}^{(ss)}_{ij} - \boldsymbol{Q}^{(ss)}_{ij}\|_\infty$$

$$\leq(C_1 C_2 c_{x0}^2 + \mu^2)\max_{a,b}\left\|\mathrm{Tr}\left(\widehat{\boldsymbol{K}}^{(ss)}_{ij,S_a^{(s)},S_b^{(s)}}\right) - \mathrm{Tr}\left(\boldsymbol{K}^{(ss)}_{ij,S_a^{(s)},S_b^{(s)}}\right)\right\|_\infty$$

$$\leq(C_1 C_2 c_{x0}^2 + \mu^2)p\left\|\widehat{\boldsymbol{K}}^{(ss)}_{ij} - \boldsymbol{K}^{(ss)}_{ij}\right\|_\infty,$$

where $C_2 = 1 + \|\mathcal{W}^{(h)}_{ss}\|_{L^\infty\to L^\infty}$.

Then we can bound

$$\|\widehat{\boldsymbol{K}}^{(s)} - \bar{\boldsymbol{K}}^{(s)}\|_{op} \leq \|\widehat{\boldsymbol{K}}^{(s)} - \bar{\boldsymbol{K}}^{(s)}\|_F = \sqrt{\sum_{i=1}^n\sum_{j=1}^n\left[\mathrm{Tr}\left(\widehat{\boldsymbol{Q}}^{(s)}_{ij}\right) - \mathrm{Tr}\left(\boldsymbol{Q}^{(s)}_{ij}\right)\right]^2}$$

$$\leq\sqrt{\sum_{i=1}^n\sum_{j=1}^n p\sum_{r=1}^p\left[\widehat{\boldsymbol{Q}}^{(s)}_{ij,rr} - \boldsymbol{Q}^{(s)}_{ij,rr}\right]^2}$$

$$\leq\sqrt{\sum_{i=1}^n\sum_{j=1}^n p\sum_{r=1}^p\left[\widehat{\boldsymbol{Q}}^{(ss)}_{ij,rr}\mathbb{E}_{((\boldsymbol{M},\boldsymbol{N})\sim\widehat{\boldsymbol{A}}^{(s)})}\sigma'(\boldsymbol{M}_r)\sigma'(\boldsymbol{N}_r)^\top - \boldsymbol{Q}^{(ss)}_{ij,rr}\mathbb{E}_{((\boldsymbol{M},\boldsymbol{N})\sim\bar{\boldsymbol{A}}^{(s)})}\sigma'(\boldsymbol{M}_r)\sigma'(\boldsymbol{N}_r)^\top\right]^2}$$

$$\leq\sqrt{\sum_{i=1}^n\sum_{j=1}^n p^2\sum_{r=1}^p(C_1 C_2 c_{x0}^2 + \mu^2)^2\|\widehat{\boldsymbol{K}}^{(ss)}_{ij} - \bar{\boldsymbol{K}}^{(ss)}_{ij}\|_\infty^2}$$

$$\leq(C_1 C_2 c_{x0}^2 + \mu^2)C_3 p^2 \lambda$$

$$\overset{\text{\textcircled{1}}}{\leq} \frac{\lambda}{8},$$

where ① holds by setting $C_3 \leq \frac{1}{(C_1 C_2 c_{x0}^2 + \mu^2)p^2}$. In this way, we have

$$\left\|\boldsymbol{G}^{hs}(0) - (\boldsymbol{\alpha}^{(h)}_{s,3})^2\bar{\boldsymbol{K}}^{(s)}\right\|_{op} \leq \left\|\boldsymbol{G}^{hs}(0) - (\boldsymbol{\alpha}^{(h)}_{s,3})^2\widehat{\boldsymbol{K}}^{(s)}\right\|_{op} + (\boldsymbol{\alpha}^{(h)}_{s,3})^2\left\|\widehat{\boldsymbol{K}}^{(s)} - \bar{\boldsymbol{K}}^{(s)}\right\|_{op} \leq \frac{\lambda}{4}.$$

The proof is completed. $\qquad\square$

### C.4.3 Proof of Lemma 24

*Proof.* To begin with, according to the definition, we have

$$\boldsymbol{K}^{(ls)}_{ij} - \boldsymbol{b}^{(l)}_i(\boldsymbol{b}^{(s)}_i)^\top = \sum_{t=1}^{l-1}\sum_{q=1}^{s-1}\left[\boldsymbol{\alpha}^{(l)}_{t,2}\boldsymbol{\alpha}^{(s)}_{q,2}\left(\boldsymbol{K}^{(tq)}_{ij} - \boldsymbol{b}^{(t)}_i(\boldsymbol{b}^{(q)}_i)^\top\right)\right.$$

$$\left.+\tau^2\boldsymbol{\alpha}^{(l)}_{t,3}\boldsymbol{\alpha}^{(s)}_{q,3}\left[\mathbb{E}_{(\boldsymbol{M}^{(ls)}_{tq},\boldsymbol{N}^{(ls)}_{tq})}\sigma(\boldsymbol{M}^{(ls)}_{tq})\sigma(\boldsymbol{N}^{(ls)}_{tq})^\top - \mathbb{E}_{\boldsymbol{M}^{(ls)}_{tq}}\sigma(\boldsymbol{M}^{(ls)}_{tq})\mathbb{E}_{\boldsymbol{N}^{(ls)}_{tq}}\sigma(\boldsymbol{N}^{(ls)}_{tq})^\top\right]\right].$$

By defining

$$\boldsymbol{R}^{(ls)}_{tq} := \mathbb{E}_{(\boldsymbol{M}^{(ls)}_{tq},\boldsymbol{N}^{(ls)}_{tq})}\begin{bmatrix}\sigma(\boldsymbol{M}^{(ls)}_{tq})\sigma(\boldsymbol{M}^{(ls)}_{tq})^\top, & \sigma(\boldsymbol{M}^{(ls)}_{tq})\sigma(\boldsymbol{N}^{(ls)}_{tq})^\top \\ \sigma(\boldsymbol{N}^{(ls)}_{tq})\sigma(\boldsymbol{M}^{(ls)}_{tq})^\top, & \sigma(\boldsymbol{N}^{(ls)}_{tq})\sigma(\boldsymbol{N}^{(ls)}_{tq})^\top\end{bmatrix}$$

$$- \mathbb{E}_{(\boldsymbol{M}^{(ls)}_{tq},\boldsymbol{N}^{(ls)}_{tq})}\begin{bmatrix}\sigma(\boldsymbol{M}^{(ls)}_{tq}) \\ \sigma(\boldsymbol{N}^{(ls)}_{tq})\end{bmatrix}\mathbb{E}_{(\boldsymbol{M}^{(ls)}_{tq},\boldsymbol{N}^{(ls)}_{tq})}\left[(\sigma(\boldsymbol{M}^{(ls)}_{tq})^\top, \sigma(\boldsymbol{N}^{(ls)}_{tq})^\top\right],$$

we can further obtain

$$
\begin{bmatrix} \boldsymbol{K}_{ii}^{(ls)}, \boldsymbol{K}_{ij}^{(ls)} \\ \boldsymbol{K}_{ji}^{(ls)}, \boldsymbol{K}_{jj}^{(ls)} \end{bmatrix} - \begin{bmatrix} \boldsymbol{b}_i^{(l)} \\ \boldsymbol{b}_j^{(l)} \end{bmatrix} \left[ (\boldsymbol{b}_i^{(s)})^\top, (\boldsymbol{b}_j^{(s)})^\top \right]
$$
$$
= \sum_{t=1}^{l-1} \sum_{q=1}^{s-1} \left[ \boldsymbol{\alpha}_{t,2}^{(l)} \boldsymbol{\alpha}_{q,2}^{(s)} \left[ \begin{bmatrix} \boldsymbol{K}_{ii}^{(tq)}, \boldsymbol{K}_{ij}^{(tq)} \\ \boldsymbol{K}_{ji}^{(tq)}, \boldsymbol{K}_{jj}^{(tq)} \end{bmatrix} - \begin{bmatrix} \boldsymbol{b}_i^{(t)} \\ \boldsymbol{b}_j^{(t)} \end{bmatrix} \left[ (\boldsymbol{b}_i^{(q)})^\top, (\boldsymbol{b}_j^{(q)})^\top \right] \right] + \tau^2 \boldsymbol{\alpha}_{t,3}^{(l)} \boldsymbol{\alpha}_{q,3}^{(l)} \boldsymbol{R}_{tq}^{(ls)} \right].
$$

Let

$$
\bar{\boldsymbol{R}}_{tq}^{(ls)} = \begin{bmatrix} \sigma(\boldsymbol{M}_{tq}^{(ls)}) \\ \sigma(\boldsymbol{N}_{tq}^{(ls)}) \end{bmatrix} - \mathbb{E}_{(\boldsymbol{M}_{tq}^{(ls)}, \boldsymbol{N}_{tq}^{(ls)})} \begin{bmatrix} \sigma(\boldsymbol{M}_{tq}^{(ls)}) \\ \sigma(\boldsymbol{N}_{tq}^{(ls)}) \end{bmatrix}.
$$

Then we have

$$
\boldsymbol{R}_{tq}^{(ls)} = \mathbb{E}_{(\boldsymbol{M}_{tq}^{(ls)}, \boldsymbol{N}_{tq}^{(ls)})} \left[ \bar{\boldsymbol{R}}_{tq}^{(ls)} (\bar{\boldsymbol{R}}_{tq}^{(ls)})^\top \right] \succeq \boldsymbol{0}.
$$

Therefore, by induction, we can conclude

$$
\begin{bmatrix} \boldsymbol{K}_{ii}^{(ls)}, \boldsymbol{K}_{ij}^{(ls)} \\ \boldsymbol{K}_{ji}^{(ls)}, \boldsymbol{K}_{jj}^{(ls)} \end{bmatrix} - \begin{bmatrix} \boldsymbol{b}_i^{(l)} \\ \boldsymbol{b}_j^{(l)} \end{bmatrix} \left[ (\boldsymbol{b}_i^{(s)})^\top, (\boldsymbol{b}_j^{(s)})^\top \right] \succeq a \left[ \begin{bmatrix} \boldsymbol{K}_{ii}^{(-1)}, \boldsymbol{K}_{ij}^{(-1)} \\ \boldsymbol{K}_{ji}^{(-1)}, \boldsymbol{K}_{jj}^{(-1)} \end{bmatrix} - \begin{bmatrix} \boldsymbol{b}_i^{(-1)} \\ \boldsymbol{b}_j^{(-1)} \end{bmatrix} \left[ (\boldsymbol{b}_i^{-1})^\top, (\boldsymbol{b}_j^{(-1)})^\top \right] \right]
$$
$$
\succeq a \begin{bmatrix} \boldsymbol{K}_{ii}^{(-1)}, \boldsymbol{K}_{ij}^{(-1)} \\ \boldsymbol{K}_{ji}^{(-1)}, \boldsymbol{K}_{jj}^{(-1)} \end{bmatrix} \overset{①}{\succ} \boldsymbol{0},
$$

where $a$ is a constant that depends on $\boldsymbol{\alpha}_{t,2}^{(l)}$ ($\forall l, t$), ① holds by using Lemma 5 which shows that $\boldsymbol{K}_{ii}^{(00)} \succ 0$. Based on this result, we can estimate

$$
\begin{bmatrix} \boldsymbol{K}_{ii}^{(ll)}, \boldsymbol{K}_{ij}^{(ll)} \\ \boldsymbol{K}_{ji}^{(ll)}, \boldsymbol{K}_{jj}^{(ll)} \end{bmatrix} - \begin{bmatrix} \boldsymbol{b}_i^{(l)} \\ \boldsymbol{b}_j^{(l)} \end{bmatrix} \left[ (\boldsymbol{b}_i^{(l)})^\top, (\boldsymbol{b}_j^{(l)})^\top \right]
$$
$$
= \sum_{t=1}^{l-1} \sum_{q=1}^{l-1} \left[ \boldsymbol{\alpha}_{t,2}^{(l)} \boldsymbol{\alpha}_{q,2}^{(s)} \left[ \begin{bmatrix} \boldsymbol{K}_{ii}^{(tq)}, \boldsymbol{K}_{ij}^{(tq)} \\ \boldsymbol{K}_{ji}^{(tq)}, \boldsymbol{K}_{jj}^{(tq)} \end{bmatrix} - \begin{bmatrix} \boldsymbol{b}_i^{(t)} \\ \boldsymbol{b}_j^{(t)} \end{bmatrix} \left[ (\boldsymbol{b}_i^{q})^\top, (\boldsymbol{b}_j^{(q)})^\top \right] \right] + \tau^2 \boldsymbol{\alpha}_{t,3}^{(l)} \boldsymbol{\alpha}_{q,3}^{(l)} \boldsymbol{R}_{tq}^{(ls)} \right]
$$
$$
\succeq \sum_{t=1}^{l-1} \left[ (\boldsymbol{\alpha}_{t,2}^{(l)})^2 \left[ \begin{bmatrix} \boldsymbol{K}_{ii}^{(tt)}, \boldsymbol{K}_{ij}^{(tt)} \\ \boldsymbol{K}_{ji}^{(tt)}, \boldsymbol{K}_{jj}^{(tt)} \end{bmatrix} - \begin{bmatrix} \boldsymbol{b}_i^{(t)} \\ \boldsymbol{b}_j^{(t)} \end{bmatrix} \left[ (\boldsymbol{b}_i^t)^\top, (\boldsymbol{b}_j^{(t)})^\top \right] \right] + \tau^2 (\boldsymbol{\alpha}_{t,3}^{(l)})^2 \boldsymbol{R}_{tt}^{(ll)} \right]
$$
$$
\succeq \left( \prod_{t=1}^{l-1} (\boldsymbol{\alpha}_{t,2}^{(l)})^2 \right) \left[ \begin{bmatrix} \boldsymbol{K}_{ii}^{(-1)}, \boldsymbol{K}_{ij}^{(-1)} \\ \boldsymbol{K}_{ji}^{(-1)}, \boldsymbol{K}_{jj}^{(-1)} \end{bmatrix} - \begin{bmatrix} \boldsymbol{b}_i^{(-1)} \\ \boldsymbol{b}_j^{(-1)} \end{bmatrix} \left[ (\boldsymbol{b}_i^{-1})^\top, (\boldsymbol{b}_j^{(-1)})^\top \right] \right]
$$
$$
\succeq \left( \prod_{t=1}^{l-1} (\boldsymbol{\alpha}_{t,2}^{(l)})^2 \right) \begin{bmatrix} \boldsymbol{K}_{ii}^{(-1)}, \boldsymbol{K}_{ij}^{(-1)} \\ \boldsymbol{K}_{ji}^{(-1)}, \boldsymbol{K}_{jj}^{(-1)} \end{bmatrix}.
$$

Then there must exit a constant $c$ such that

$$
\lambda_{\min}(\boldsymbol{K}^{(ll)}) \geq \left( \prod_{t=0}^{l-1} (\boldsymbol{\alpha}_{t,2}^{(l)})^2 \right) \lambda_{\min}(\widehat{K}).
$$

where $\widehat{K} = \begin{bmatrix} \boldsymbol{K}_{ii}^{(-1)}, \boldsymbol{K}_{ij}^{(-1)} \\ \boldsymbol{K}_{ji}^{(-1)}, \boldsymbol{K}_{jj}^{(-1)} \end{bmatrix}$. On the other hand, we have

$$
\boldsymbol{Q}_{ij,ab}^{(ll)} = \mathrm{Tr}\left( \boldsymbol{K}_{ij, S_a^{(l)}, S_b^{(l)}}^{(ll)} \right),
$$

where $S_a^{(s)} = \{j \mid \boldsymbol{X}_{:,j}^{(s-1)} \in$ the $a-$th patch for convolution$\}$. This actually means that we can obtain $\boldsymbol{Q}_{ij}^{(ll)}$ by using (adding) linear transformation on $\boldsymbol{K}_{ij}^{(ll)}$. Since for all $\boldsymbol{Q}_{ij}^{(ll)}$ we use the same linear transformation which means that $\boldsymbol{Q}^{(ll)}$ by using linear transformation on $\boldsymbol{K}^{(ll)}$. Since linear transformation does not change the eigenvalue property of a matrix, we can further obtain

$$
\lambda_{\min}(\boldsymbol{Q}^{(ll)}) \geq \left( \prod_{t=0}^{l-1} (\boldsymbol{\alpha}_{t,2}^{(l)})^2 \right) \lambda_{\min}(\widehat{K}).
$$

Finally, let $\mathbf{Q} = \mathbf{B}\mathbf{S}\mathbf{B}^\top$ be the SVD of $\mathbf{Q}$ and $\mathbf{Z} = \mathbf{S}^{1/2}\mathbf{B}^\top$ denotes $n$ samples (each column denotes one). Since $\mathbf{Q}$ is full rank, the samples in $\mathbf{Z}$ are not parallel. In this way, we can apply Lemma 5 and obtain that $\mathbf{Q}^{(s)}$ which is defined below, is full rank

$$\mathbf{A}^{(l)} = \begin{bmatrix} \mathcal{W}_{ll}^{(h)}(\mathbf{Q}_{ii}^{(ll)}), \mathcal{W}_{ll}^{(h)}(\mathbf{Q}_{ij}^{(ll)}) \\ \mathcal{W}_{ll}^{(h)}(\mathbf{Q}_{ji}^{(ll)}), \mathcal{W}_{ll}^{(h)}(\mathbf{Q}_{jj}^{(ll)}) \end{bmatrix},$$

$$\mathbf{Q}_{ij,ab}^{(l)} = \mathbf{Q}_{ij,ab}^{(ll)} \mathbb{E}_{((\mathbf{M},\mathbf{N})\sim \bar{\mathbf{A}}^{(l)})} \sigma'(\mathbf{M})\,\sigma'(\mathbf{N})^\top, \qquad \mathbf{K}_{ij,ab}^{(l)} = \mathrm{Tr}\left(\mathbf{Q}_{ij}^{(s)}\right),\ (s = l, \cdots, h-1).$$

Recall that Lemma 10 shows

$$\frac{1}{c_{x0}} \le \|\mathbf{X}^{(l)}(0)\|_F \le c_{x0}.$$

where $c_{x0} \ge 1$ is a constant. Therefore, we have $\mathbf{K}_{ii}^{ll} = \langle \mathbf{X}^{(l)}(0), \mathbf{X}^{(l)}(0)\rangle \in [1/c_{x0}^2, c_{x0}^2]$ and thus $\mathbf{Q}_{ii}^{ll} = \langle \Phi(\mathbf{X}^{(l)}(0)), \Phi(\mathbf{X}^{(l)}(0))\rangle \ge \langle \mathbf{X}^{(l)}(0), \mathbf{X}^{(l)}(0)\rangle \ge 1/c_{x0}^2$ and $\mathbf{Q}_{ii}^{ll} = \langle \Phi(\mathbf{X}^{(l)}(0)), \Phi(\mathbf{X}^{(l)}(0))\rangle \le k_c \langle \mathbf{X}^{(l)}(0), \mathbf{X}^{(l)}(0)\rangle \ge k_c/c_{x0}^2$. Then we have

$$\mathbf{Q}_{ij}^{(l)} = \mathbf{Q}_{ij}^{ll}\mathbb{E}_{(\mathbf{M}\sim\mathcal{N}0,\mathbf{I})}\sigma'(\mathbf{M}\mathbf{Z}_i)\,\sigma'(\mathbf{M}\mathbf{Z}_j)^\top$$

where $\mathbf{Z} = \mathbf{S}^{1/2}\mathbf{B}^\top$ and $\mathbf{Z}_i = \mathbf{Z}_{:i}$ in which $\mathbf{Q}^{ll} = \mathbf{B}\mathbf{S}\mathbf{B}^\top$ is the SVD of $\mathbf{Q}^{ll}$. Since Since $\mathbf{Q}^{ll}$ is full rank, the samples in $\mathbf{Z}$ are not parallel. Then we can apply Lemma 6 and obtain

$$\lambda_{\min}(\mathbf{Q}^{(l)}) \ge c_\sigma \left(\prod_{t=0}^{l-1} (\boldsymbol{\alpha}_{t,2}^{(l)})^2\right) \lambda_{\min}(\widehat{K}),$$

where $c_\sigma$ is a constant that only depends on $\sigma$ and input data. Since

$$\mathbf{K}_{ij,ab}^{(s)} = \mathrm{Tr}\left(\mathbf{Q}_{ij}^{(s)}\right),\ (s = 0, h-1)$$

which means that $\mathbf{K}^{(s)}$ can be obtained by using adding linear transformation on $\mathbf{Q}^{(s)}$. So the eigenvalue of $\mathbf{K}^{(s)}$ also satisfies

$$\lambda_{\min}(\mathbf{K}^{(l)}) \ge c_\sigma \left(\prod_{t=0}^{l-1} (\boldsymbol{\alpha}_{t,2}^{(l)})^2\right) \lambda_{\min}(\widehat{K}),$$

In this way, we can further establish

$$\lambda_{\min}(\mathbf{G}(0)) \ge \sum_{s=0}^{h-1} \lambda_{\min}\left(\mathbf{G}^{hs}(0)\right) \overset{\textcircled{\scriptsize 1}}{\ge} \sum_{s=0}^{h-1} (\boldsymbol{\alpha}_{s,3}^{(h)})^2 \lambda_{\min}\left(\mathbf{K}^{(s)}(0)\right) - \frac{\lambda}{4}$$

$$\ge \frac{3c_\sigma}{4} \sum_{s=0}^{h-1} (\boldsymbol{\alpha}_{s,3}^{(h)})^2 \left(\prod_{t=0}^{s-1} (\boldsymbol{\alpha}_{t,2}^{(s)})^2\right) \lambda_{\min}(\widehat{K}),$$

where $\textcircled{\scriptsize 1}$ holds since we set $\lambda = c_\sigma \sum_{s=0}^{h-1} (\boldsymbol{\alpha}_{s,3}^{(h)})^2 \left(\prod_{t=0}^{s-1} (\boldsymbol{\alpha}_{t,2}^{(s)})^2\right) \lambda_{\min}(\widehat{K})$ and Lemma 23 shows

$$\left\|\mathbf{G}^{hs}(0) - (\boldsymbol{\alpha}_{s,3}^{(h)})^2 \mathbf{K}^{(s)}\right\|_{\mathrm{op}} \le \frac{\lambda}{4} \qquad (s = 0, \cdots, h).$$

where $\lambda$ is a constant. The proof is completed. $\qquad\square$

# D   Proofs of Results in Sec. 4

## D.1   Proof of Theorem 2

*Proof.* We first prove the first result. Suppose except one gate $\mathbf{g}_{s,t}^{(l)}$, all remaining stochastic gates $\mathbf{g}_{s',t}^{(l')}$ are fixed. Then we discuss the type of the gate $\mathbf{g}_{s,t}^{(l)}$. Note $\mathbf{g}_{s,t}^{(l)}$ denotes one operation in the operation set $\mathcal{O} = \{O_t\}_{t=1}^s$, including zero operation, skip connection, pooling, and convolution with any kernel size, between nodes $\mathbf{X}^{(s)}$ and $\mathbf{X}^{(l)}$. Now we discuss different kinds of operations.

If the gate $\mathbf{g}_{s,t}^{(l)}$ is for zero operation, it is easily to check that the loss $F_{\mathrm{val}}(\mathbf{W}^*(\boldsymbol{\beta}), \boldsymbol{\beta})$ in (2) will not change, since zero operation does not delivery any information to subsequent node $\mathbf{X}^{(l)}$.

If the gate $g_{s,t}^{(l)}$ is for skip connection, there are two cases. Firstly, increasing the weight $g_{s,t}^{(l)}$ gives smaller loss. For this case, it directly obtain our result. Secondly, increasing the weight $g_{s,t}^{(l)}$ gives larger loss. For this case, suppose we increase $g_{s,t}^{(l)}$ to $g_{s,t}^{(l)} + \epsilon$. Then node $\boldsymbol{X}^{(l)}$ will become $\boldsymbol{X}^{(l)} + \epsilon \boldsymbol{X}^{(s)} = \boldsymbol{X}_{\text{conv}}^{(l)} + \boldsymbol{X}_{\text{nonconv}}^{(l)} + \epsilon \boldsymbol{X}^{(s)}$ if we fix the remaining operations, where $\boldsymbol{X}_{\text{conv}}^{(l)}$ denotes the output of convolution and $\boldsymbol{X}_{\text{nonconv}}^{(l)}$ denotes the sum of all remaining operations. Now suppose the convolution operation between node $\boldsymbol{X}^{(l)}$ and $\boldsymbol{X}^{(s)}$ is $g_{s,t}^{(l)}\text{conv}(\boldsymbol{W}_s^{(l)}; \boldsymbol{X}^{(s)}) = g_{s,t}^{(l)}\sigma(\boldsymbol{W}_s^{(l)}\Phi(\boldsymbol{X}^{(s)}))$ where $t$ denotes the index of convolution in the operation set . Then we consider a function

$$g_{s,t}^{(l)}\sigma(\bar{\boldsymbol{W}}_s^{(l)}\Phi(\boldsymbol{X}^{(s)})) = -\epsilon \boldsymbol{X}^{(s)}. \tag{22}$$

Since for the almost activation functions are monotone increasing, this means that $\sigma()$ does not change the rank of $\bar{\boldsymbol{W}}_s^{(l)}\Phi(\boldsymbol{X}^{(s)})$. At the same time, the linear transformation $\Phi(\boldsymbol{X}^{(s)})$ has the same rank as $\boldsymbol{X}^{(s)}$. Then when $g_{s,t}^{(l)} \neq 0$ there exist a $\bar{\boldsymbol{W}}_s^{(l)}$ such that Eqn. (22) holds. On the other hand, we already have

$$g_{s,t}^{(l)}\sigma(\boldsymbol{W}_s^{(l)}\Phi(\boldsymbol{X}^{(s)})) = \boldsymbol{X}_{\text{conv}}^{(l)}.$$

Since we assume the function $\sigma()$ is Lipschitz and smooth and the constant $\epsilon$ is sufficient small, then by using mean value theorem, there must exist $g_{s,t}^{(l)}\sigma(\widetilde{\boldsymbol{W}}_s^{(l)}\Phi(\boldsymbol{X}^{(s)})) = \boldsymbol{X}_{\text{conv}}^{(l)} - \epsilon \boldsymbol{X}^{(s)}$. So the convolution can counteract the increment $\epsilon \boldsymbol{X}^{(s)}$ brought by increasing the weight of skip connection. In this way, the whole network remains the same, leading the same loss. When the weight of convolution satisfies $g_{s,t}^{(l)} = 0$, we only need to increase $g_{s,t}^{(l)}$ to a positive constant, then we use the same method and can prove the same result. In this case, we actually increase the weights of skip connection and convolution at the same time, which also accords with our results in the Proposition 2.

If the gate $g_{s,t}^{(l)}$ is for pooling connection, we can use the same method for skip connection to prove our result, since pooling operation is also a linear transformation.

If the gate $g_{s,t}^{(l)}$ is for convolution, then we increase it to $g_{s,t}^{(l)} + \epsilon g_{s,t}^{(l)}$ and obtain the new output $(1+\epsilon)\boldsymbol{X}_{\text{conv}}^{(l)}$ because of $g_{s,t}^{(l)}\sigma(\boldsymbol{W}_s^{(l)}\Phi(\boldsymbol{X}^{(s)})) = \boldsymbol{X}_{\text{conv}}^{(l)}$. If the new feature map can lead to smaller loss, then we directly obtain our results. If the new feature map can lead to larger loss we only need to find a new parameter $\widetilde{\boldsymbol{W}}_s^{(l)}$ such that $g_{s,t}^{(l)}\sigma(\widetilde{\boldsymbol{W}}_s^{(l)}\Phi(\boldsymbol{X}^{(s)})) = \frac{1}{1+\epsilon}\boldsymbol{X}_{\text{conv}}^{(l)}$. Since for most activation $\sigma(0) = 0$, we have $g_{s,t}^{(l)}\sigma(\bar{\boldsymbol{W}}_s^{(l)}\Phi(\boldsymbol{X}^{(s)})) = 0$ when $\bar{\boldsymbol{W}}_s^{(l)} = 0$. On the other hand, we have $g_{s,t}^{(l)}\sigma(\boldsymbol{W}_s^{(l)}\Phi(\boldsymbol{X}^{(s)})) = \boldsymbol{X}_{\text{conv}}^{(l)}$. Moreover since we assume the function $\sigma()$ is Lipschitz and smooth and the constant $\epsilon$ is sufficient small, then by using mean value theorem, there must exist $\widetilde{\boldsymbol{W}}_s^{(l)}$ such that $g_{s,t}^{(l)}\sigma(\widetilde{\boldsymbol{W}}_s^{(l)}\Phi(\boldsymbol{X}^{(s)})) = \frac{1}{1+\epsilon}\boldsymbol{X}_{\text{conv}}^{(l)}$.

Then we prove the results in the second part. From Theorem 1, we know that for the $k$-th iteration in the search phase, increasing the weights $g_{s,t_1}^{(l)}$ ($l \neq h$) of skip connects and the weights $g_{s,t_2}^{(h)}$ of convolutions can reduce the loss $F_{\text{train}}(\boldsymbol{W}^*(\boldsymbol{\beta}), \boldsymbol{\beta})$ in (2), where $t_1$ and $t_2$ respectively denote the indexes of skip connection and convolution in the operation set $\mathcal{O} = \{O_t\}_{t=1}^s$. Specifically, Theorem 1 proves for the training loss

$$\|\boldsymbol{y} - \boldsymbol{u}(k)\|_2^2 \leq \left(1 - \frac{\eta\lambda}{4}\right)^k \|\boldsymbol{y} - \boldsymbol{u}(0)\|_2^2,$$

where $\lambda = \frac{3c_\sigma}{4}\lambda_{\min}(\widehat{\boldsymbol{K}})\sum_{s=0}^{h-1}(\boldsymbol{\alpha}_{s,3}^{(h)})^2\prod_{t=0}^{s-1}(\boldsymbol{\alpha}_{t,2}^{(s)})^2$. Moreover, since $F(\boldsymbol{\Omega}) = \frac{1}{2n}\sum_{i=1}^n(u_i - y_i)^2 = \frac{1}{2n}\|\boldsymbol{u} - \boldsymbol{y}\|_2^2$, increasing the weights $g_{s,t_1}^{(l)}$ ($l \neq h$) of skip connects and the weights $g_{s,t_2}^{(h)}$ of convolutions can reduce the loss $F_{\text{train}}(\boldsymbol{W}^*(\boldsymbol{\beta}), \boldsymbol{\beta})$. Since the samples for training and validation are drawn from the same distribution which means that $\mathbb{E}[F_{\text{train}}(\boldsymbol{\Omega})] = \mathbb{E}[F_{\text{val}}(\boldsymbol{\Omega})]$, increasing weights of skip connections and convolution can reduce $F_{\text{val}}(\boldsymbol{\Omega})$ in expectation. Then by using first-order extension, we can obtain

$$\mathbb{E}\left[F_{\text{val}}(g_{s,t_1}^{(l)} + \epsilon) - F_{\text{val}}(g_{s,t_1}^{(l)})\right] = \epsilon \mathbb{E}\left[\nabla_{\bar{g}_{s,t_1}^{(l)}} F_{\text{val}}(g_{s,t_1}^{(l)})\right].$$

where $g_{s,t_1}^{(l)} \in \bar{g}_{s,t_1}^{(l)} \leq g_{s,t_1}^{(l)} + \epsilon$. Since as above analysis, increasing the weights $g_{s,t_1}^{(l)}$ ($l \neq h$) of skip connects will reduce the current loss $F_{\text{val}}(g_{s,t_1}^{(l)})$ in expectation, which means that $\mathbb{E}\left[\nabla_{g_{s,t_1}^{(l)}} F_{\text{val}}(g_{s,t_1}^{(l)})\right]$ is positive. Since when the algorithm does not converge, we have $0 < C \leq \mathbb{E}\left[\nabla_{g_{s,t_1}^{(l)}} F_{\text{val}}(g_{s,t_1}^{(l)})\right]$. In

this way, we have

$$\mathbb{E}\left[F_{\text{val}}(\boldsymbol{g}_{s,t_1}^{(l)} + \epsilon) - F_{\text{val}}(\boldsymbol{g}_{s,t_1}^{(l)})\right] \geq C\epsilon.$$

Similarly, for convolution we can obtain

$$\mathbb{E}\left[F_{\text{val}}(\boldsymbol{g}_{s,t_2}^{(l)} + \epsilon) - F_{\text{val}}(\boldsymbol{g}_{s,t_2}^{(l)})\right] \geq C\epsilon.$$

The proof is completed. $\qquad\square$

## D.2 Proof of Theorem 3

*Proof.* For the results in the first part, it is easily to check according to the definitions. Now we focus on proving the results in the second part. When $\tilde{\boldsymbol{g}}_{s,t}^{(l)} \leq -\frac{a}{b-a}$, then $\boldsymbol{g}_{s,t}^{(l)} = 0$. Meanwhile, the cumulative distribution of $\tilde{\boldsymbol{g}}_{s,t}^{(l)}$ is $\Theta\big(\tau(\ln\delta - \ln(1-\delta)) - \boldsymbol{\beta}_{s,t}^{(l)}\big)$ [7]. In this way, we can easily compute

$$
\begin{aligned}
\mathbb{P}\left(\boldsymbol{g}_{s,t}^{(l)} \neq 0\right) &= 1 - \mathbb{P}\left(\tilde{\boldsymbol{g}}_{s,t}^{(l)} \leq -\frac{a}{b-a}\right) \\
&= 1 - \Theta\left(\tau\left(\ln\left(-\frac{a}{b-a}\right) - \ln\left(1 + \frac{a}{b-a}\right)\right) - \boldsymbol{\beta}_{s,t}^{(l)}\right) \\
&= \Theta\left(\boldsymbol{\beta}_{s,t}^{(l)} - \tau\ln\frac{-a}{b}\right).
\end{aligned}
$$

The proof is completed. $\qquad\square$

## D.3 Proof of Theorem 4

*Proof.* Here we first prove the convergence rate of the shallow network with two branches. The proof is very similar to Theorem C.1. By using the totally same method, we can follow Lemma 21 to prove

$$\|\boldsymbol{y} - \boldsymbol{u}(k)\|_2^2 \leq \left(1 - \frac{\eta\lambda_{\min}(\boldsymbol{G}(0))}{4}\right)\|\boldsymbol{y} - \boldsymbol{u}(k-1)\|_2^2.$$

Here $\boldsymbol{G}(0)$ denotes the Gram matrix of the shallow network and have the same definition as the Gram matrix of deep network with one branch. Please refer to the definition of Gram matrix in Appendix B.

The second step is to prove the smallest least eigenvalue of $\boldsymbol{G}(0)$ is lower bounded. For this step, the analysis method is also the same as the method to lower bounding smallest least eigenvalue of $\boldsymbol{G}(0)$ in DARTS. Specifically, by following Lemma 24, we can obtain

$$\lambda_{\min}(\boldsymbol{G}(0)) \geq \frac{3c_\sigma}{4}\left[\sum_{s=1}^{\frac{h}{2}-1}(\boldsymbol{\alpha}_{s,3}^{(h/2)})^2\left(\prod_{t=0}^{s-1}(\boldsymbol{\alpha}_{t,2}^{(s)})^2\right) + \sum_{s=\frac{h}{2}}^{h-1}(\boldsymbol{\alpha}_{s,3}^h)^2\left(\prod_{t=0}^{s-1}(\boldsymbol{\alpha}_{t,2}^{(s)})^2\right)\right]\lambda_{\min}(\boldsymbol{K}).$$

where $c_\sigma$ is a constant that only depends on $\sigma$ and the input data, $\lambda_{\min}(\boldsymbol{K}) > 0$ is given in Theorem 1.

From Theorem 1, we know that for deep cell with one branch, the loss satisfies

$$\|\boldsymbol{y} - \boldsymbol{u}(k)\|_2^2 \leq \left(1 - \frac{\eta\lambda}{4}\right)^k\|\boldsymbol{y} - \boldsymbol{u}(0)\|_2^2,$$

where $\lambda = \frac{3c_\sigma}{4}\lambda_{\min}(\boldsymbol{K})\sum_{s=0}^{h-1}(\boldsymbol{\alpha}_{s,3}^{(h)})^2\prod_{t=0}^{s-1}(\boldsymbol{\alpha}_{t,2}^{(s)})^2$.

Since all weights $\boldsymbol{\alpha}_{s,t}^{(l)}$ belong to the range $[0,1]$, by comparison, the convergence rate $\lambda'$ of shallow cell with two branch is large than the convergence rate $\lambda$ of shallow cell with two branch:

$$
\begin{aligned}
\lambda' &= \frac{3c_\sigma}{4}\left[\sum_{s=1}^{\frac{h}{2}-1}(\boldsymbol{\alpha}_{s,3}^{(h/2)})^2\left(\prod_{t=0}^{s-1}(\boldsymbol{\alpha}_{t,2}^{(s)})^2\right) + \sum_{s=\frac{h}{2}}^{h-1}(\boldsymbol{\alpha}_{s,3}^h)^2\left(\prod_{t=0}^{s-1}(\boldsymbol{\alpha}_{t,2}^{(s)})^2\right)\right]\lambda_{\min}(\boldsymbol{K}) \\
&> \lambda = \frac{3c_\sigma}{4}\lambda_{\min}(\boldsymbol{K})\sum_{s=0}^{h-1}(\boldsymbol{\alpha}_{s,3}^{(h)})^2\prod_{t=0}^{s-1}(\boldsymbol{\alpha}_{t,2}^{(s)})^2.
\end{aligned}
$$

This completes the proof. $\qquad\square$

# E  Proofs of Auxiliary Lemmas

## E.1  Proof of Lemma 8

*Proof.* We use chain rule to obtain the following gradients:

$$\frac{\partial \ell}{\partial \boldsymbol{X}^{(h-1)}} = (u-y)\boldsymbol{W}_h \in \mathbb{R}^{m \times p};$$

$$\frac{\partial \ell}{\partial \boldsymbol{X}^{(l)}} = (u-y)\boldsymbol{W}_l + \sum_{s=l+1}^{h} \frac{\partial \ell}{\partial \boldsymbol{X}^{(s)}} \frac{\partial \boldsymbol{X}^{(s)}}{\partial \boldsymbol{X}^{(l)}} \ (l=0,\cdots,h-2)$$

$$= (u-y)\boldsymbol{W}_l + \sum_{s=l+1}^{h} \left( \boldsymbol{\alpha}_{l,2}^{(s)} \frac{\partial \ell}{\partial \boldsymbol{X}^{(s)}} + \boldsymbol{\alpha}_{l,3}^{(s)} \tau \Psi \left( (\boldsymbol{W}_l^{(s)})^\top \left( \sigma' \left( \boldsymbol{W}_l^{(s)} \Phi(\boldsymbol{X}^{(l)}) \right) \odot \frac{\partial \ell}{\partial \boldsymbol{X}^{(s)}} \right) \right) \right) \in \mathbb{R}^{m \times p};$$

$$\frac{\partial \ell}{\partial \boldsymbol{X}} = \frac{\partial \ell}{\partial \boldsymbol{X}^{(1)}} \frac{\partial \boldsymbol{X}^{(1)}}{\partial \boldsymbol{X}^{(0)}} = \tau \Psi \left( (\boldsymbol{W}^{(0)})^\top \left( \sigma' \left( \boldsymbol{W}^{(0)} \Phi(\boldsymbol{X}) \right) \odot \frac{\partial \ell}{\partial \boldsymbol{X}^{(0)}} \right) \right) \in \mathbb{R}^{m \times p},$$

$$\frac{\partial \ell}{\partial \boldsymbol{W}_s^{(l)}} = \frac{\partial \ell}{\partial \boldsymbol{X}^{(l)}} \frac{\partial \boldsymbol{X}^{(l)}}{\partial \boldsymbol{W}_s^{(l)}} = \boldsymbol{\alpha}_{s,3}^{(l)} \tau \Phi(\boldsymbol{X}^{(s)}) \left( \sigma' \left( \boldsymbol{W}_s^{(l)} \Phi(\boldsymbol{X}^{(s)}) \right) \odot \frac{\partial \ell}{\partial \boldsymbol{X}^{(l)}} \right)^\top \in \mathbb{R}^{m \times p}$$

$$(1 \le l \le h, 1 \le s \le l-1);$$

$$\frac{\partial \ell}{\partial \boldsymbol{W}^{(0)}} = \frac{\partial \ell}{\partial \boldsymbol{X}^{(0)}} \frac{\partial \boldsymbol{X}^{(0)}}{\partial \boldsymbol{W}^{(0)}} = \tau \Phi(\boldsymbol{X}) \left( \sigma' \left( \boldsymbol{W}^{(0)} \Phi(\boldsymbol{X}) \right) \odot \frac{\partial \ell}{\partial \boldsymbol{X}^{(0)}} \right)^\top \in \mathbb{R}^{m \times p},$$

$$\frac{\partial \ell}{\partial \boldsymbol{W}_s} = (u-y)\boldsymbol{X}^{(l)} \in \mathbb{R}^{m \times p},$$

where $\odot$ denotes the dot product. $\qquad \square$

## E.2  Proof of Lemma 9

*Proof.* We use chain rule to obtain the following gradients:

$$\frac{\partial u}{\partial \boldsymbol{X}^{(h-1)}} = \boldsymbol{W}_{h-1} \in \mathbb{R}^{m \times p};$$

$$\frac{\partial u}{\partial \boldsymbol{X}^{(l)}} = \boldsymbol{W}_l + \sum_{s=l+1}^{h} \frac{\partial u}{\partial \boldsymbol{X}^{(s)}} \frac{\partial \boldsymbol{X}^{(s)}}{\partial \boldsymbol{X}^{(l)}} \ (l=0,\cdots,h-2)$$

$$= \boldsymbol{W}_l + \sum_{s=l+1}^{h} \left( \boldsymbol{\alpha}_{l,2}^{(s)} \frac{\partial u}{\partial \boldsymbol{X}^{(s)}} + \boldsymbol{\alpha}_{l,3}^{(s)} \tau \Psi \left( (\boldsymbol{W}_l^{(s)})^\top \left( \sigma' \left( \boldsymbol{W}_l^{(s)} \Phi(\boldsymbol{X}^{(l)}) \right) \odot \frac{\partial u}{\partial \boldsymbol{X}^{(s)}} \right) \right) \right) \in \mathbb{R}^{m \times p};$$

$$(0 \le l \le h-1, 0 \le s \le l-1),$$

$$\frac{\partial u}{\partial \boldsymbol{X}} = \frac{\partial u}{\partial \boldsymbol{X}^{(1)}} \frac{\partial \boldsymbol{X}^{(1)}}{\partial \boldsymbol{X}^{(0)}} = \tau \Psi \left( (\boldsymbol{W}^{(0)})^\top \left( \sigma' \left( \boldsymbol{W}^{(0)} \Phi(\boldsymbol{X}) \right) \odot \frac{\partial u}{\partial \boldsymbol{X}^{(0)}} \right) \right) \in \mathbb{R}^{m \times p},$$

$$\frac{\partial u}{\partial \boldsymbol{W}_s^{(l)}} = \frac{\partial u}{\partial \boldsymbol{X}^{(l)}} \frac{\partial \boldsymbol{X}^{(l)}}{\partial \boldsymbol{W}_s^{(l)}} = \boldsymbol{\alpha}_{s,3}^{(l)} \tau \Phi(\boldsymbol{X}^{(s)}) \left( \sigma' \left( \boldsymbol{W}_s^{(l)} \Phi(\boldsymbol{X}^{(s)}) \right) \odot \frac{\partial u}{\partial \boldsymbol{X}^{(l)}} \right)^\top \in \mathbb{R}^{m \times p}$$

$$(0 \le l \le h-1, 1 \le s \le l-1);$$

$$\frac{\partial u}{\partial \boldsymbol{W}^{(0)}} = \frac{\partial u}{\partial \boldsymbol{X}^{(0)}} \frac{\partial \boldsymbol{X}^{(0)}}{\partial \boldsymbol{W}^{(0)}} = \tau \Phi(\boldsymbol{X}) \left( \sigma' \left( \boldsymbol{W}^{(0)} \Phi(\boldsymbol{X}) \right) \odot \frac{\partial u}{\partial \boldsymbol{X}^{(0)}} \right)^\top \in \mathbb{R}^{m \times p},$$

where $\odot$ denotes the dot product. $\qquad \square$

## E.3  Proof of Lemma 10

*Proof.* We each layer in turn. Our proof follows the proof framework in [4]. Note for notation simplicity, we have assumed that the input $\boldsymbol{X}$ is of size $m \times p$ in Sec. B. To begin with, we look at the

first layer. For brevity, let $\boldsymbol{H} = \Phi(\boldsymbol{X})$. According to the definition, we have

$$
\begin{aligned}
\mathbb{E}\left[\|\boldsymbol{X}^{(0)}(0)\|_F^2\right] =&\tau^2\mathbb{E}\left[\|\sigma(\boldsymbol{W}^{(0)}(0)\Phi(\boldsymbol{X}))\|_F^2\right] = \tau^2 \sum_{i=1}^{m}\sum_{j=1}^{p}\mathbb{E}\left[\sigma^2(\boldsymbol{W}_{i:}^{(0)}(0)\boldsymbol{H}_{:j})\right] \\
&\overset{①}{=} \sum_{j=1}^{p}\mathbb{E}_{\omega\sim\mathcal{N}(0,1)}\left[\sigma^2(\|\boldsymbol{H}_{:j}\|_F\omega)\right] \overset{②}{\geq} \mathbb{E}_{\omega\sim\mathcal{N}(0,1)}\left[\sigma^2(\|\boldsymbol{H}_{:j'}\|_F\omega)\right] \\
&\geq \mathbb{E}_{\omega\sim\mathcal{N}(0,\frac{1}{\sqrt{p}})}\left[\sigma^2(\omega)\right] := c > 0,
\end{aligned}
$$

where ① holds since $\tau = 1/\sqrt{m}$ and the entries in $\boldsymbol{W}^{(0)}(0)$ obeys i.i.d. Gaussian distribution which gives $\sum_{i=1}^{n}a_i\omega_i \sim \mathcal{N}(0,\sum_{i=1}^{n}a_i^2)$ with $\omega_i \sim \mathcal{N}(0,1)$; ② holds since $\|\boldsymbol{X}\| = 1$ which means there must exist one $j'$ such that $\|\boldsymbol{H}_{:j'}\|_F \geq \frac{1}{\sqrt{p}}$.

Next, we can bound the variance

$$
\begin{aligned}
&\mathsf{Var}\left[\|\boldsymbol{X}^{(0)}(0)\|_F^2\right] \\
=&\tau^4\mathsf{Var}\left[\|\sigma(\boldsymbol{W}^{(0)}(0)\Phi(\boldsymbol{X}))\|_F^2\right] = \tau^4\mathsf{Var}\left[\sum_{i=1}^{m}\sum_{j=1}^{p}\mathbb{E}\left[\sigma^2(\boldsymbol{W}_{i:}^{(0)}(0)\boldsymbol{H}_{:j})\right]\right] \\
\overset{①}{=}&\tau^2\mathsf{Var}\left[\sum_{j=1}^{p}\mathbb{E}\left[\sigma^2(\boldsymbol{W}_{i:}^{(0)}(0)\boldsymbol{H}_{:j})\right]\right] \overset{②}{\leq} \tau^2\mathbb{E}_{\omega\sim\mathcal{N}(0,1)}\left[\left(\left(\sum_{j=1}^{p}(\sigma(0) + \|\boldsymbol{H}_{:j}\|\|\omega|)^2\right)^2\right]\right. \\
\leq&\frac{p^2}{m}c_1,
\end{aligned}
$$

where ① holds since $\tau = 1/\sqrt{m}$ and the entries in $\boldsymbol{W}^{(0)}(0)$ obeys i.i.d. Gaussian distribution, ② holds since $\mathsf{Var}(x) \leq \mathbb{E}[x^2] - [\mathbb{E}(x)]^2$, ③ holds since $\|\boldsymbol{H}_{:j}\| \leq 1$ and $c_1 = \sigma^4(0) + 4|\sigma^3(0)|\mu\sqrt{2/\pi} + 8|\sigma(0)|\mu^3\sqrt{2/\pi} + 32\mu^4$. Then by using Chebyshev's inequality in Lemma 1, we have

$$
\mathbb{P}\left(|\|\boldsymbol{X}^{(0)}(0)\|_F^2 - \mathbb{E}[\|\boldsymbol{X}^{(0)}(0)\|_F^2]| \geq \frac{c}{2}\right) \leq \frac{4\mathsf{Var}(\|\boldsymbol{X}^{(0)}(0)\|_F^2)}{c^2} \leq \frac{4p^2}{mc^2}c_1.
$$

By setting $m \geq \frac{4c_1np^2}{c^2\delta}$, we have with probability at least $1 - \frac{\delta}{n}$,

$$
\|\boldsymbol{X}^{(0)}(0)\|_F^2 \geq \frac{c}{2}.
$$

Meanwhile, we can upper bound $\|\boldsymbol{X}^{(0)}(0)\|_F^2$ as follows:

$$
\|\boldsymbol{X}^{(0)}(0)\|_F^2 \leq \tau^2\|\sigma(\boldsymbol{W}^{(0)}(0)\Phi(\boldsymbol{X}))\|_F^2 \leq \tau^2\mu^2\|\boldsymbol{W}^{(0)}(0)\Phi(\boldsymbol{X})\|_F^2 \overset{①}{\leq} \mu^2 c_{w0}^2\|\Phi(\boldsymbol{X})\|_F^2 \overset{②}{\leq} k_c\mu^2 c_{w0}^2,
$$

where ① holds since $\|\boldsymbol{W}_s^{(l)}(0)\|_2 \leq \sqrt{m}c_{w0}$, and ② uses $\|\Phi(\boldsymbol{X})\|_F^2 \leq k_c\|\boldsymbol{X}\|_F^2$.

Next we consider the cases where $l \geq 1$. According to the definition, we can obtain

$$
\begin{aligned}
\|\boldsymbol{X}^{(l)}(0)\|_F =&\left\|\sum_{s=0}^{l-1}\left(\boldsymbol{\alpha}_{s,2}^{(l)}\boldsymbol{X}^{(s)}(0) + \boldsymbol{\alpha}_{s,3}^{(l)}\tau\sigma(\boldsymbol{W}_s^{(l)}(0)\Phi(\boldsymbol{X}^{(s)}(0)))\right)\right\|_F \\
\leq&\sum_{s=0}^{l-1}\left(\boldsymbol{\alpha}_{s,2}^{(l)}\|\boldsymbol{X}^{(s)}(0)\|_F + \boldsymbol{\alpha}_{s,3}^{(l)}\tau\|\sigma(\boldsymbol{W}_s^{(l)}(0)\Phi(\boldsymbol{X}^{(s)}(0)))\|_F\right) \\
\overset{①}{\leq}&\left(\boldsymbol{\alpha}_{s,2}^{(l)} + \boldsymbol{\alpha}_{s,3}^{(l)}\sqrt{k_c}\mu c_{w0}\right)\sum_{s=0}^{l-1}\|\boldsymbol{X}^{(s)}(0)\|_F \\
\overset{②}{\leq}&\frac{c_2^{l+1} - 1}{c_2 - 1}c_2\sqrt{k_c}\mu c_{w0},
\end{aligned}
$$

where ① uses the fact that $\|\sigma(\boldsymbol{W}_s^{(l)}(0)\Phi(\boldsymbol{X}^{(s)}(0)))\|_F \leq \mu\|\boldsymbol{W}_s^{(l)}(0)\Phi(\boldsymbol{X}^{(s)}(0))\|_F \leq \sqrt{m}\mu c_{w0}\|\Phi(\boldsymbol{X}^{(s)}(0))\|_F \leq \sqrt{m}\mu\sqrt{k_c}c_{w0}\|\boldsymbol{X}^{(s)}(0)\|_F$, ② holds by setting $c_2 = \boldsymbol{\alpha}_{s,2}^{(l)} + \boldsymbol{\alpha}_{s,3}^{(l)}\sqrt{k_c}\mu c_{w0}$.

Similarly, we can obtain

$$
\begin{aligned}
\|\boldsymbol{X}^{(l)}(0)\|_F &= \left\|\sum_{s=0}^{l-1}\left(\boldsymbol{\alpha}_{s,2}^{(l)}\boldsymbol{X}^{(s)}(0) + \boldsymbol{\alpha}_{s,3}^{(l)}\tau\sigma(\boldsymbol{W}_s^{(l)}(0)\Phi(\boldsymbol{X}^{(s)}(0)))\right)\right\|_F \\
&\geq \min_{0\leq s\leq l-1}\left|\boldsymbol{\alpha}_{s,2}^{(l)}\|\boldsymbol{X}^{(s)}(0)\|_F - \boldsymbol{\alpha}_{s,3}^{(l)}\tau\|\sigma(\boldsymbol{W}_s^{(l)}(0)\Phi(\boldsymbol{X}^{(s)}(0)))\|_F\right| \\
&\geq \min_{0\leq s\leq l-1}\left|\boldsymbol{\alpha}_{s,2}^{(l)} - \boldsymbol{\alpha}_{s,3}^{(l)}\sqrt{k_c}\mu c_{w0}\right|\|\boldsymbol{X}^{(s)}(0)\|_F \\
&\geq \left|\boldsymbol{\alpha}_{s,2}^{(l)} - \boldsymbol{\alpha}_{s,3}^{(l)}\sqrt{k_c}\mu c_{w0}\right|^{l-1}\sqrt{k_c}\mu c_{w0} > 0.
\end{aligned}
$$

Therefore, we can obtain that there exists a constant $c_{x0}$ such that for all $l \in [0, 1, \cdots, h-1]$,

$$
\frac{1}{c_{x0}} \leq \|\boldsymbol{X}^{(l)}(0)\|_F \leq c_{x0}.
$$

The proof is completed. $\qquad\square$

### E.4 Proof of Lemma 11

*Proof.* For this proof, we will respectively bound each layer. We first consider the first layer, namely $l = 1$.

**Step 1. Case where $l = 0$: upper bound of $\|\boldsymbol{X}^{(0)}(k) - \boldsymbol{X}^{(0)}(0)\|_F$.** According to the definition, we have $\boldsymbol{X}^{(0)}(k) = \tau\sigma(\boldsymbol{W}^{(0)}(k)\Phi(\boldsymbol{X}))$ which yields

$$
\begin{aligned}
\|\boldsymbol{X}^{(0)}(k) - \boldsymbol{X}^{(0)}(0)\|_F &= \tau\|\sigma(\boldsymbol{W}^{(0)}(k)\Phi(\boldsymbol{X})) - \sigma(\boldsymbol{W}^{(0)}(k)\Phi(\boldsymbol{X}))\|_F \\
&\overset{\textcircled{1}}{\leq} \tau\mu\|\boldsymbol{W}^{(0)}(k)\Phi(\boldsymbol{X}) - \boldsymbol{W}^{(0)}(0)\Phi(\boldsymbol{X})\|_F \\
&\overset{\textcircled{2}}{\leq} \tau\mu\sqrt{k_c}\|\boldsymbol{W}^{(0)}(k) - \boldsymbol{W}^{(0)}(0)\|_F \\
&\overset{\textcircled{3}}{\leq} \mu\sqrt{k_c}r,
\end{aligned}
$$

where $\textcircled{1}$ uses the $\mu$-Lipschitz of $\sigma(\cdot)$, $\textcircled{2}$ uses $\|\Phi(\boldsymbol{X})\| \leq \sqrt{k_c}\|\boldsymbol{X}\| \leq \sqrt{k_c}$, $\textcircled{3}$ uses the assumption $\|\boldsymbol{W}^{(0)}(k) - \boldsymbol{W}^{(0)}(0)\|_2 \leq \sqrt{m}r$.

**Step 2. Case where $l \geq 1$: upper bound of $\|\boldsymbol{X}^{(l)}(k) - \boldsymbol{X}^{(l)}(0)\|_F$.** According to the definition, we have

$$
\begin{aligned}
&\|\boldsymbol{X}^{(l)}(k) - \boldsymbol{X}^{(l)}(0)\|_F \\
&= \left\|\sum_{s=0}^{l-1}\left[\boldsymbol{\alpha}_{s,2}^{(l)}\left(\boldsymbol{X}^{(s)}(k) - \boldsymbol{X}^{(s)}(0)\right) + \boldsymbol{\alpha}_{s,3}^{(l)}\tau\left(\sigma(\boldsymbol{W}_s^{(l)}(k)\Phi(\boldsymbol{X}^{(s)}(k))) - \sigma(\boldsymbol{W}_s^{(l)}(0)\Phi(\boldsymbol{X}^{(s)}(0)))\right)\right]\right\|_F \\
&= \sum_{s=0}^{l-1}\left[\boldsymbol{\alpha}_{s,2}^{(l)}\left\|\boldsymbol{X}^{(s)}(k) - \boldsymbol{X}^{(s)}(0)\right\|_F + \boldsymbol{\alpha}_{s,3}^{(l)}\tau\left\|\sigma(\boldsymbol{W}_s^{(l)}(k)\Phi(\boldsymbol{X}^{(s)}(k))) - \sigma(\boldsymbol{W}_s^{(l)}(0)\Phi(\boldsymbol{X}^{(s)}(0)))\right\|_F\right] \\
&\leq \sum_{s=0}^{l-1}\left[\boldsymbol{\alpha}_{s,2}^{(l)}\left\|\boldsymbol{X}^{(s)}(k) - \boldsymbol{X}^{(s)}(0)\right\|_F + \boldsymbol{\alpha}_{s,3}^{(l)}\tau\mu\left\|\boldsymbol{W}_s^{(l)}(k)\Phi(\boldsymbol{X}^{(s)}(k)) - \boldsymbol{W}_s^{(l)}(0)\Phi(\boldsymbol{X}^{(s)}(0))\right\|_F\right]
\end{aligned}
$$

Then we first bound the second term as follows:

$$
\begin{aligned}
&\left\|\boldsymbol{W}_s^{(l)}(k)\Phi(\boldsymbol{X}^{(s)}(k)) - \boldsymbol{W}_s^{(l)}(0)\Phi(\boldsymbol{X}^{(s)}(0))\right\|_F \\
&\leq \left\|\boldsymbol{W}_s^{(l)}(k)\Phi(\boldsymbol{X}^{(s)}(k)) - \boldsymbol{W}_s^{(l)}(k)\Phi(\boldsymbol{X}^{(s)}(0))\right\|_F + \left\|\boldsymbol{W}_s^{(l)}(k)\Phi(\boldsymbol{X}^{(s)}(0)) - \boldsymbol{W}_s^{(l)}(0)\Phi(\boldsymbol{X}^{(s)}(0))\right\|_F \\
&\leq \|\boldsymbol{W}_s^{(l)}(k)\|\left\|\Phi(\boldsymbol{X}^{(s)}(k)) - \Phi(\boldsymbol{X}^{(s)}(0))\right\|_F + \left\|\boldsymbol{W}_s^{(l)}(k) - \boldsymbol{W}_s^{(l)}(0)\right\|_F\|\Phi(\boldsymbol{X}^{(s)}(0))\|_F \\
&\leq \sqrt{k_c}\|\boldsymbol{W}_s^{(l)}(k)\|\left\|\boldsymbol{X}^{(s)}(k) - \boldsymbol{X}^{(s)}(0)\right\|_F + \sqrt{k_c}\left\|\boldsymbol{W}_s^{(l)}(k) - \boldsymbol{W}_s^{(l)}(0)\right\|_F\|\boldsymbol{X}^{(s)}(0)\|_F \\
&\overset{\textcircled{1}}{\leq} \sqrt{k_c}\sqrt{m}\left(r + c_{w0}\right)\left\|\boldsymbol{X}^{(s)}(k) - \boldsymbol{X}^{(s)}(0)\right\|_F + \sqrt{k_c}m c_{x0}\widetilde{r},
\end{aligned}
$$

where in $\textcircled{1}$ we use $\|\boldsymbol{W}_s^{(l)}(k)\|_F \leq \|\boldsymbol{W}_s^{(l)}(k) - \boldsymbol{W}_s^{(l)}(0)\|_F + \|\boldsymbol{W}_s^{(l)}(0)\|_F \leq \sqrt{m}(r + c_{w0})$, $\left\|\boldsymbol{W}_s^{(l)}(k) - \boldsymbol{W}_s^{(l)}(0)\right\|_F \leq \sqrt{m}\widetilde{r}$, and the results in Lemma 10 that $\frac{1}{c_{x0}} \leq \|\boldsymbol{X}^{(l)}(0)\|_F \leq c_{x0}$. Plugging

this result into the above inequality gives

$$\|\boldsymbol{X}^{(l)}(k) - \boldsymbol{X}^{(l)}(0)\|_F$$

$$\leq \sum_{s=0}^{l-1} \left[ \boldsymbol{\alpha}_{s,2}^{(l)} \left\| \boldsymbol{X}^{(s)}(k) - \boldsymbol{X}^{(s)}(0) \right\|_F + \boldsymbol{\alpha}_{s,3}^{(l)} \tau \mu \left\| \boldsymbol{W}_s^{(l)}(k)\Phi(\boldsymbol{X}^{(s)}(k)) - \boldsymbol{W}_s^{(l)}(0)\Phi(\boldsymbol{X}^{(s)}(0)) \right\|_F \right]$$

$$\leq \sum_{s=0}^{l-1} \left[ \left( \boldsymbol{\alpha}_{s,2}^{(l)} + \boldsymbol{\alpha}_{s,3}^{(l)}\mu\sqrt{k_c}\left(r + c_{w0}\right) \right) \left\| \boldsymbol{X}^{(s)}(k) - \boldsymbol{X}^{(s)}(0) \right\|_F + \boldsymbol{\alpha}_{s,3}^{(l)}\mu\sqrt{k_c}c_{x0}\widetilde{r} \right]$$

$$\leq \sum_{s=0}^{l-1} \left[ \left( \boldsymbol{\alpha}_{s,2}^{(l)} + \boldsymbol{\alpha}_{s,3}^{(l)}\mu\sqrt{k_c}\left(r + c_{w0}\right) \right) \left\| \boldsymbol{X}^{(s)}(k) - \boldsymbol{X}^{(s)}(0) \right\|_F + \boldsymbol{\alpha}_{s,3}^{(l)}\mu\sqrt{k_c}c_{x0}\widetilde{r} \right] \tag{23}$$

$$\leq \sum_{s=0}^{l-1} \left[ \left( \boldsymbol{\alpha}_2 + \boldsymbol{\alpha}_3\mu\sqrt{k_c}\left(r + c_{w0}\right) \right) \left\| \boldsymbol{X}^{(s)}(k) - \boldsymbol{X}^{(s)}(0) \right\|_F + \boldsymbol{\alpha}_{s,3}^{(l)}\mu\sqrt{k_c}c_{x0}\widetilde{r} \right]$$

$$\leq \left( 1 + \boldsymbol{\alpha}_2 + \boldsymbol{\alpha}_3\mu\sqrt{k_c}\left(r + c_{w0}\right) \right) \|\boldsymbol{X}^{(l-1)}(k) - \boldsymbol{X}^{(l-1)}(0)\|_F$$

$$\leq \left( 1 + \boldsymbol{\alpha}_2 + \boldsymbol{\alpha}_3\mu\sqrt{k_c}\left(r + c_{w0}\right) \right)^l \|\boldsymbol{X}^{(0)}(k) - \boldsymbol{X}^{0)}(0)\|_F$$

$$\leq \left( 1 + \boldsymbol{\alpha}_2 + \boldsymbol{\alpha}_3\mu\sqrt{k_c}\left(r + c_{w0}\right) \right)^l \mu\sqrt{k_c}r,$$

where $\boldsymbol{\alpha}_2 = \max_{s,l} \boldsymbol{\alpha}_{s,2}^{(l)}$ and $\boldsymbol{\alpha}_3 = \max_{s,l} \boldsymbol{\alpha}_{s,3}^{(l)}$.

By using Eqn. (23), we have

$$\left\| \boldsymbol{W}_s^{(l)}(k)\Phi\left(\boldsymbol{X}^{(s)}(k)\right) - \boldsymbol{W}_s^{(l)}(0)\Phi\left(\boldsymbol{X}^{(s)}(0)\right) \right\|_F \leq \frac{1}{\boldsymbol{\alpha}_3}\left( 1 + \boldsymbol{\alpha}_2 + \boldsymbol{\alpha}_3\mu\sqrt{k_c}\left(r + c_{w0}\right) \right)^l \sqrt{k_c}mr,$$

The proof is completed. $\qquad\qquad\qquad\qquad\qquad\qquad\qquad\qquad\qquad\qquad\qquad\square$

## E.5 Proof of Lemma 12

*Proof.* According to definition, we have

$$\frac{1}{n}\sum_{i=1}^n \left\| \frac{\partial \ell}{\partial \boldsymbol{X}_i^{(h)}(t)} \right\|_F = \frac{1}{n}\sum_{i=1}^n \|(u_i(t) - y_i)\boldsymbol{W}_h(t)\|_F \overset{\text{①}}{\leq} \frac{1}{\sqrt{n}}\|\boldsymbol{u}(t) - \boldsymbol{y}\|_F \|\boldsymbol{W}_l(t)\|_F \overset{\text{②}}{\leq} c_y c_u, \tag{24}$$

where ① holds since $\sum_{i=1}^n |u_i - y_i| \leq \sqrt{n}\|\boldsymbol{u} - \boldsymbol{y}\|_2 = \sqrt{n}\sqrt{\sum_i(u_i - y_i)^2}$, ② holds by assuming $\frac{1}{\sqrt{n}}\|\boldsymbol{u}(t) - \boldsymbol{y}\|_F = c_y$ and $\|\boldsymbol{W}_h(t)\|_F \leq c_u$.

Then for $0 \leq l < h$, we have

$$\frac{1}{n}\sum_{i=1}^n \left\| \frac{\partial \ell}{\partial \boldsymbol{X}_i^{(l)}(t)} \right\|_F = \frac{1}{n}\sum_{i=1}^n \|(u_i(t) - y_i)\boldsymbol{W}_l(t)$$

$$+ \sum_{s=l+1}^{h-1} \left( \boldsymbol{\alpha}_{l,2}^{(s)}\frac{\partial \ell}{\partial \boldsymbol{X}_i^{(s)}(t)} + \boldsymbol{\alpha}_{l,3}^{(s)}\tau\Psi\left( (\boldsymbol{W}_l^{(s)}(t))^\top \left( \sigma'\left( \boldsymbol{W}_l^{(s)}(t)\Phi(\boldsymbol{X}_i^{(l)}(t)) \right) \odot \frac{\partial \ell}{\partial \boldsymbol{X}_i^{(s)}(t)} \right) \right) \right) \bigg\|_F$$

$$\leq \frac{1}{n}\sum_{i=1}^n \|(u_i(t) - y_i)\boldsymbol{W}_l(t)\|_F$$

$$+ \sum_{s=l+1}^{h-1} \frac{1}{n}\sum_{i=1}^n \left\| \boldsymbol{\alpha}_{l,2}^{(s)}\frac{\partial \ell}{\partial \boldsymbol{X}_i^{(s)}(t)} + \boldsymbol{\alpha}_{l,3}^{(s)}\tau\Psi\left( (\boldsymbol{W}_l^{(s)}(t))^\top \left( \sigma'\left( \boldsymbol{W}_l^{(s)}(t)\Phi(\boldsymbol{X}_i^{(l)}(t)) \right) \odot \frac{\partial \ell}{\partial \boldsymbol{X}_i^{(s)}(t)} \right) \right) \right\|_F$$

The main task is to bound

$$
\left\| \boldsymbol{\alpha}_{l,2}^{(s)} \frac{\partial \ell}{\partial \boldsymbol{X}_i^{(s)}(t)} + \boldsymbol{\alpha}_{l,3}^{(s)} \tau \Psi\left( (\boldsymbol{W}_l^{(s)}(t))^\top \left( \sigma'\left( \boldsymbol{W}_l^{(s)}(t)\Phi(\boldsymbol{X}_i^{(l)}(t)) \right) \odot \frac{\partial \ell}{\partial \boldsymbol{X}_i^{(s)}(t)} \right) \right) \right\|_F
$$

$$
\leq \boldsymbol{\alpha}_{l,2}^{(s)} \left\| \frac{\partial \ell}{\partial \boldsymbol{X}_i^{(s)}(t)} \right\|_F + \boldsymbol{\alpha}_{l,3}^{(s)} \tau \left\| \Psi\left( (\boldsymbol{W}_l^{(s)}(t))^\top \left( \sigma'\left( \boldsymbol{W}_l^{(s)}(t)\Phi(\boldsymbol{X}_i^{(l)}(t)) \right) \odot \frac{\partial \ell}{\partial \boldsymbol{X}_i^{(s)}(t)} \right) \right) \right\|_F
$$

$$
\overset{①}{\leq} \boldsymbol{\alpha}_{l,2}^{(s)} \left\| \frac{\partial \ell}{\partial \boldsymbol{X}_i^{(s)}(t)} \right\|_F + \boldsymbol{\alpha}_{l,3}^{(s)} \tau \mu \sqrt{k_c} \| \boldsymbol{W}_l^{(s)}(t) \|_F \left\| \frac{\partial \ell}{\partial \boldsymbol{X}_i^{(s)}(t)} \right\|_F
$$

$$
\overset{①}{\leq} \left( \boldsymbol{\alpha}_{l,2}^{(s)} + \boldsymbol{\alpha}_{l,3}^{(s)} \mu \sqrt{k_c}(c_{w0} + r) \right) \left\| \frac{\partial \ell}{\partial \boldsymbol{X}_i^{(s)}(t)} \right\|_F,
$$

where ① holds since $\|\Psi(\boldsymbol{X})\|_F \leq \sqrt{k_c}\|\boldsymbol{X}\|_F$ and the activation function $\sigma(\cdot)$ is $\mu$-Lipschitz, ② holds since $\|\boldsymbol{W}_l^{(s)}(t)\|_F \leq \|\boldsymbol{W}_l^{(s)}(t) - \boldsymbol{W}_l^{(s)}(0)\|_F + \|\boldsymbol{W}_l^{(s)}(0)\|_F \leq \sqrt{m}(c_{w0} + r)$. Similar to (24), we can prove

$$
\frac{1}{n}\sum_{i=1}^n \|(u_i(t) - y_i)\boldsymbol{W}_l(t)\|_F \leq \frac{1}{\sqrt{n}} \|\boldsymbol{u}(t) - \boldsymbol{y}\|_F \|\boldsymbol{W}_l(t)\|_F \leq c_y c_u,
$$

Combining the above results yields

$$
\frac{1}{n}\sum_{i=1}^n \left\| \frac{\partial \ell}{\partial \boldsymbol{X}_i^{(l)}(t)} \right\|_F \leq c_y c_u + \sum_{s=l+1}^{h-1} \left( \boldsymbol{\alpha}_{l,2}^{(s)} + \boldsymbol{\alpha}_{l,3}^{(s)} \mu \sqrt{k_c}(c_{w0} + r) \right) \frac{1}{n}\sum_{i=1}^n \left\| \frac{\partial \ell}{\partial \boldsymbol{X}_i^{(s)}(t)} \right\|_F
$$

$$
\overset{①}{\leq} c_y c_u + \sum_{s=l+1}^{h-1} \left( \boldsymbol{\alpha}_2 + \boldsymbol{\alpha}_3 \mu \sqrt{k_c}(c_{w0} + r) \right) \frac{1}{n}\sum_{i=1}^n \left\| \frac{\partial \ell}{\partial \boldsymbol{X}_i^{(s)}(t)} \right\|_F
$$

$$
\leq \left( 1 + \boldsymbol{\alpha}_2 + \boldsymbol{\alpha}_3 \mu \sqrt{k_c}(c_{w0} + r) \right) \frac{1}{n}\sum_{i=1}^n \left\| \frac{\partial \ell}{\partial \boldsymbol{X}_i^{(l-1)}(t)} \right\|_F
$$

$$
\leq \left( 1 + \boldsymbol{\alpha}_2 + \boldsymbol{\alpha}_3 \mu \sqrt{k_c}(c_{w0} + r) \right)^l \frac{1}{n}\sum_{i=1}^n \left\| \frac{\partial \ell}{\partial \boldsymbol{X}_i^{(0)}(t)} \right\|_F
$$

$$
\leq \left( 1 + \boldsymbol{\alpha}_2 + \boldsymbol{\alpha}_3 \mu \sqrt{k_c}(c_{w0} + r) \right)^l c_y c_u,
$$

where ① uses $\boldsymbol{\alpha}_2 = \max_{s,l} \boldsymbol{\alpha}_{s,2}^{(l)}$ and $\boldsymbol{\alpha}_3 = \max_{s,l} \boldsymbol{\alpha}_{s,3}^{(l)}$. The proof is completed. □

### E.6 Proof of Lemma 13

*Proof.* Here we use mathematical induction to prove these results in turn. We first consider $t = 0$. The following results hold:

$$
\|\boldsymbol{W}_s^{(l)}(t) - \boldsymbol{W}_s^{(l)}(0)\|_F \leq \sqrt{m}\widetilde{r}, \quad \|\boldsymbol{W}_s(t) - \boldsymbol{W}_s(0)\|_F \leq \sqrt{m}\widetilde{r}. \tag{25}
$$

Now we assume (25) holds for $t = 1, \cdots, k$. We only need to prove it hold for $t + 1$. According to the definitions, we can establish

$$
\|\boldsymbol{W}_s^{(l)}(t+1) - \boldsymbol{W}_s^{(l)}(t)\|_F = \eta \boldsymbol{\alpha}_{s,3}^{(l)} \tau \left\| \frac{1}{n}\sum_{i=1}^n \Phi(\boldsymbol{X}_i^{(s)}(t)) \left( \sigma'\left( \boldsymbol{W}_s^{(l)}(t)\Phi(\boldsymbol{X}_i^{(s)}(t)) \right) \odot \frac{\partial \ell}{\partial \boldsymbol{X}_i^{(l)}(t)} \right)^\top \right\|_F
$$

$$
\leq \eta \boldsymbol{\alpha}_{s,3}^{(l)} \tau \frac{1}{n}\sum_{i=1}^n \left\| \Phi(\boldsymbol{X}_i^{(s)}(t)) \left( \sigma'\left( \boldsymbol{W}_s^{(l)}(t)\Phi(\boldsymbol{X}_i^{(s)}(t)) \right) \odot \frac{\partial \ell}{\partial \boldsymbol{X}_i^{(l)}(t)} \right)^\top \right\|_F
$$

$$
\overset{①}{\leq} \eta \boldsymbol{\alpha}_{s,3}^{(l)} \tau \sqrt{k_c} \frac{1}{n}\sum_{i=1}^n \|\boldsymbol{X}_i^{(s)}(t)\| \left\| \sigma'\left( \boldsymbol{W}_s^{(l)}(t)\Phi(\boldsymbol{X}_i^{(s)}(t)) \right) \odot \frac{\partial \ell}{\partial \boldsymbol{X}_i^{(l)}(t)} \right\|_F
$$

$$
\overset{②}{\leq} 2\eta \boldsymbol{\alpha}_{s,3}^{(l)} \tau \sqrt{k_c} c_{x0} \frac{1}{n}\sum_{i=1}^n \left\| \sigma'\left( \boldsymbol{W}_s^{(l)}(t)\Phi(\boldsymbol{X}_i^{(s)}(t)) \right) \odot \frac{\partial \ell}{\partial \boldsymbol{X}_i^{(l)}(t)} \right\|_F
$$

where ① holds since $\|\Phi(\boldsymbol{X}^{(s)})\|_F \leq \sqrt{k_c}\|\boldsymbol{X}^{(s)}\|_F$; ② holds since in Lemma 11 and Lemma 10, we have

$$
\begin{aligned}
\|\boldsymbol{X}^{(l)}(t)\| \leq & \|\boldsymbol{X}^{(l)}(t) - \boldsymbol{X}^{(l)}(0)\|_F + \|\boldsymbol{X}^{(l)}(0)\|_F \\
\leq & c_{x0} + \left(1 + \boldsymbol{\alpha}_2 + \boldsymbol{\alpha}_3 \mu\sqrt{k_c}\,(r + c_{w0})\right)^l \mu\sqrt{k_c}\,r \\
\overset{①}{\leq} & 2c_{x0},
\end{aligned}
\tag{26}
$$

where $\boldsymbol{\alpha}_2 = \max_{s,l} \boldsymbol{\alpha}_{s,2}^{(l)}$ and $\boldsymbol{\alpha}_3 = \max_{s,l} \boldsymbol{\alpha}_{s,3}^{(l)}$, and $c_{x0} \geq 1$ is given in Lemma 10. The inequality holds by setting $r$ small enough, namely $r \leq \min(\frac{c_{x0}}{\left(1+\boldsymbol{\alpha}_2+2\boldsymbol{\alpha}_3\mu\sqrt{k_c}c_{w0}\right)^l \mu\sqrt{k_c}}, c_{w0})$. This condition will be satisfied by setting enough large $m$ and will be discussed later.

Since the activation function $\sigma(\cdot)$ is $\mu$-Lipschitz, we have

$$
\left\| \sigma'\left(\boldsymbol{W}_s^{(l)}(t)\Phi(\boldsymbol{X}^{(s)}(t))\right) \odot \frac{\partial\ell}{\partial\boldsymbol{X}^{(l)}(t)} \right\|_F \leq \mu \left\| \frac{\partial\ell}{\partial\boldsymbol{X}^{(l)}(t)} \right\|_F.
$$

So the remaining task is to upper bound $\left\| \frac{\partial\ell}{\partial\boldsymbol{X}^{(l)}(t)} \right\|_F$. Towards this goal, we have $\frac{1}{\sqrt{n}}\|\boldsymbol{u}(t) - \boldsymbol{y}\|_F \leq c_y = \frac{1}{\sqrt{n}}(1 - \frac{\eta\lambda}{2})^{t/2}\|\boldsymbol{y} - \boldsymbol{u}(0)\|_2$, $\|\boldsymbol{W}_h(t)\|_F \leq \|\boldsymbol{W}_h(t) - \boldsymbol{W}_h(0)\|_F + \|\boldsymbol{W}_h(0)\|_F \leq c_u = \sqrt{m}(\widetilde{r} + c_{w0})$, $\|\boldsymbol{W}_l^{(s)}(t) - \boldsymbol{W}_l^{(s)}(0)\|_F \leq \sqrt{m}r$, and $\|\boldsymbol{W}_l^{(s)}(0)\|_F \leq c_{w0}$. In this way, we can use Lemma Lemma 12 and obtain

$$
\frac{1}{n}\sum_{i=1}^n \left\| \frac{\partial\ell}{\partial\boldsymbol{X}_i^{(l)}(t)} \right\|_F \leq c_1 c_y c_u = \frac{c_1(\widetilde{r} + c_{w0})}{\sqrt{n}} \left(1 - \frac{\eta\lambda}{2}\right)^{t/2} \|\boldsymbol{y} - \boldsymbol{u}(0)\|_2,
$$

where $c_1 = \left(1 + \boldsymbol{\alpha}_2 + \boldsymbol{\alpha}_3\tau\mu\sqrt{k_c}(\widetilde{r} + c_{w0})\right)^l$ with $\boldsymbol{\alpha}_2 = \max_{s,l} \boldsymbol{\alpha}_{s,2}^{(l)}$ and $\boldsymbol{\alpha}_3 = \max_{s,l} \boldsymbol{\alpha}_{s,3}^{(l)}$.

By combining the above results, we can directly obtain

$$
\begin{aligned}
\|\boldsymbol{W}_s^{(l)}(t+1) - \boldsymbol{W}_s^{(l)}(t)\|_F \leq & \frac{2c_1\eta\boldsymbol{\alpha}_{s,3}^{(l)}\mu\sqrt{k_c}c_{x0}(\widetilde{r} + c_{w0})}{\sqrt{n}} \|\boldsymbol{u}(t) - \boldsymbol{y}\|_F \\
\leq & \frac{2c_1\eta\boldsymbol{\alpha}_{s,3}^{(l)}\mu\sqrt{k_c}c_{x0}(\widetilde{r} + c_{w0})}{\sqrt{n}} \left(1 - \frac{\eta\lambda}{2}\right)^{t/2} \|\boldsymbol{y} - \boldsymbol{u}(0)\|_2.
\end{aligned}
$$

Therefore, we have

$$
\begin{aligned}
\|\boldsymbol{W}_s^{(l)}(t+1) - \boldsymbol{W}_s^{(l)}(0)\|_F \leq & \|\boldsymbol{W}_s^{(l)}(t+1) - \boldsymbol{W}_s^{(l)}(t)\|_F + \|\boldsymbol{W}_s^{(l)}(t) - \boldsymbol{W}_s^{(l)}(0)\|_F \\
\leq & \frac{8c_1\boldsymbol{\alpha}_{s,3}^{(l)}\mu\sqrt{k_c}c_{x0}(\widetilde{r} + c_{w0})}{\lambda\sqrt{n}} \|\boldsymbol{y} - \boldsymbol{u}(0)\|_2 \overset{①}{\leq} \sqrt{m}\widetilde{r},
\end{aligned}
$$

where ① holds by setting $\widetilde{r} = \frac{16\left(1+\boldsymbol{\alpha}_2+2\boldsymbol{\alpha}_3\mu\sqrt{k_c}c_{w0}\right)^l \boldsymbol{\alpha}_{s,3}^{(l)}\mu\sqrt{k_c}c_{x0}c_{w0}}{\lambda\sqrt{mn}}\|\boldsymbol{y} - \boldsymbol{u}(0)\|_2 \leq c_{w0}$. By using the same way, we can prove

$$
\|\boldsymbol{W}^{(0)}(t+1) - \boldsymbol{W}^{(0)}(t)\|_F \leq \frac{2c_1\eta\mu\sqrt{k_c}c_{x0}(\widetilde{r} + c_{w0})}{\sqrt{n}} \left(1 - \frac{\eta\lambda}{2}\right)^{t/2} \|\boldsymbol{y} - \boldsymbol{u}(0)\|_2,
$$

$$
\|\boldsymbol{W}_s^{(l)}(t+1) - \boldsymbol{W}_s^{(l)}(0)\|_F \leq \sqrt{m}\widetilde{r}.
$$

Then similarly, we can obtain

$$
\begin{aligned}
\|\boldsymbol{W}_s(t+1) - \boldsymbol{W}_s(t)\|_F = & \eta \left\| \frac{1}{n}\sum_{i=1}^n (u_i - y_i)\boldsymbol{X}_i^{(s)}(t) \right\|_F \leq \eta\frac{1}{n}\sum_{i=1}^n |u_i(t) - y_i| \left\| \boldsymbol{X}_i^{(s)}(t) \right\|_F \\
\overset{①}{\leq} & \frac{2\eta c_{x0}}{\sqrt{n}} \|\boldsymbol{u}(t) - \boldsymbol{y}\|_2 \leq \frac{2\eta c_{x0}}{\sqrt{n}} \left(1 - \frac{\eta\lambda}{2}\right)^{t/2} \|\boldsymbol{y} - \boldsymbol{u}(0)\|_2,
\end{aligned}
$$

where ① holds since $\sum_{i=1}^n |u_i - y_i| \leq \sqrt{n}\|\boldsymbol{u} - \boldsymbol{y}\|_2$, and $\left\| \boldsymbol{X}_i^{(s)}(t) \right\|_F \leq 2c_{x0}$ in (E.7). Then we establish

$$
\begin{aligned}
\|\boldsymbol{W}_s(t+1) - \boldsymbol{W}_s(0)\|_F \leq & \|\boldsymbol{W}_s(t+1) - \boldsymbol{W}_s(t)\|_F + \|\boldsymbol{W}_s(t) - \boldsymbol{W}_s(0)\|_F \\
\leq & \frac{8c_{x0}\|\boldsymbol{y} - \boldsymbol{u}(0)\|_2}{\lambda\sqrt{n}} \overset{①}{\leq} \sqrt{m}\widetilde{r},
\end{aligned}
$$

where ① holds by setting $\widetilde{r} = \frac{8c_{x0}\|\boldsymbol{y}-\boldsymbol{u}(0)\|_2}{\lambda\sqrt{mn}}$. Finally, combining the value of $\widetilde{r}$, we have $\widetilde{r} = \max\left(\frac{8c_{x0}\|\boldsymbol{y}-\boldsymbol{u}(0)\|_2}{\lambda\sqrt{mn}}, \frac{16\left(1+\boldsymbol{\alpha}_2+2\boldsymbol{\alpha}_3\mu\sqrt{k_c}c_{w0}\right)^l\boldsymbol{\alpha}_{s,3}^{(l)}\mu\sqrt{k_c}c_{x0}c_{w0}}{\lambda\sqrt{mn}}\|\boldsymbol{y}-\boldsymbol{u}(0)\|_2\right) \le c_{w0}$. Under this setting, we have

$$
\begin{aligned}
\|\boldsymbol{W}_s^{(l)}(t+1) - \boldsymbol{W}_s^{(l)}(t)\|_F &\le \frac{4c\eta\boldsymbol{\alpha}_{s,3}^{(l)}\mu c_{x0}c_{w0}\sqrt{k_c}}{\sqrt{n}}\|\boldsymbol{u}(t)-\boldsymbol{y}\|_F \\
&\le \frac{4c\eta\boldsymbol{\alpha}_{s,3}^{(l)}\mu c_{x0}c_{w0}\sqrt{k_c}}{\sqrt{n}}\left(1-\frac{\eta\lambda}{2}\right)^{t/2}\|\boldsymbol{y}-\boldsymbol{u}(0)\|_2, \\
\|\boldsymbol{W}^{(0)}(t+1) - \boldsymbol{W}^{(0)}(t)\|_F &\le \frac{4c\eta\mu c_{x0}c_{w0}\sqrt{k_c}}{\sqrt{n}}\|\boldsymbol{u}(t)-\boldsymbol{y}\|_F \\
&\le \frac{4c\eta\mu c_{x0}c_{w0}\sqrt{k_c}}{\sqrt{n}}\left(1-\frac{\eta\lambda}{2}\right)^{t/2}\|\boldsymbol{y}-\boldsymbol{u}(0)\|_2,
\end{aligned}
$$

where $c = \left(1+\boldsymbol{\alpha}_2+2\boldsymbol{\alpha}_3\mu\sqrt{k_c}c_{w0}\right)^l$ with $\boldsymbol{\alpha}_2 = \max_{s,l}\boldsymbol{\alpha}_{s,2}^{(l)}$ and $\boldsymbol{\alpha}_3 = \max_{s,l}\boldsymbol{\alpha}_{s,3}^{(l)}$. The proof is completed. $\qquad\square$

### E.7  Proof of Lemma 14

*Proof.* We use mathematical induction to prove the results. We first consider $h = 0$. According to the definition, we have

$$
\begin{aligned}
\left\|\boldsymbol{X}^{(0)}(k+1) - \boldsymbol{X}^{(0)}(k)\right\|_F &= \tau\left\|\sigma(\boldsymbol{W}^{(0)}(k+1)\Phi(\boldsymbol{X})) - \sigma(\boldsymbol{W}^{(0)}(k)\Phi(\boldsymbol{X}))\right\|_F \\
&\le \tau\mu\left\|\boldsymbol{W}^{(0)}(k+1) - \boldsymbol{W}^{(0)}(k)\right\|_F\|\Phi(\boldsymbol{X})\|_F \\
&\overset{①}{\le} \tau\mu\sqrt{k_c}\left\|\boldsymbol{W}^{(0)}(k+1) - \boldsymbol{W}^{(0)}(k)\right\|_F \\
&\overset{②}{\le} \frac{4c\tau\eta\mu^2 c_{x0}c_{w0}k_c}{\sqrt{n}}\|\boldsymbol{u}(k)-\boldsymbol{y}\|_F,
\end{aligned}
$$

where ① uses $\|\Phi(\boldsymbol{X})\|_F \le \sqrt{k_c}\|\boldsymbol{X}\|_F \le \sqrt{k_c}$ where the sample $\boldsymbol{X}$ obeys $\|\boldsymbol{X}\|_F = 1$; ② uses the result in Lemma 13 that $\|\boldsymbol{W}^{(0)}(t+1) - \boldsymbol{W}^{(0)}(t)\|_F \le \frac{4c\eta\mu c_{x0}c_{w0}\sqrt{k_c}}{\sqrt{n}}\|\boldsymbol{u}(t)-\boldsymbol{y}\|_F$.

Then we first consider $h \ge 1$.

$$
\begin{aligned}
&\left\|\boldsymbol{X}^{(l)}(k+1) - \boldsymbol{X}^{(l)}(k)\right\|_F \\
&= \left\|\sum_{s=0}^{l-1}\left(\boldsymbol{\alpha}_{s,2}^{(l)}(\boldsymbol{X}^{(s)}(k+1)-\boldsymbol{X}^{(s)}(k))+\boldsymbol{\alpha}_{s,3}^{(l)}\tau\left(\sigma(\boldsymbol{W}_s^{(l)}(k+1)\Phi(\boldsymbol{X}^{(s)}(k+1)))-\sigma(\boldsymbol{W}_s^{(l)}(k)\Phi(\boldsymbol{X}^{(s)}(k)))\right)\right)\right\|_F \\
&\le \sum_{s=0}^{l-1}\left[\boldsymbol{\alpha}_{s,2}^{(l)}\left\|\boldsymbol{X}^{(s)}(k+1)-\boldsymbol{X}^{(s)}(k)\right\|_F + \boldsymbol{\alpha}_{s,3}^{(l)}\tau\left\|\sigma(\boldsymbol{W}_s^{(l)}(k+1)\Phi(\boldsymbol{X}^{(s)}(k+1)))-\sigma(\boldsymbol{W}_s^{(l)}(k)\Phi(\boldsymbol{X}^{(s)}(k)))\right\|_F\right] \\
&\le \sum_{s=0}^{l-1}\left[\boldsymbol{\alpha}_{s,2}^{(l)}\left\|\boldsymbol{X}^{(s)}(k+1)-\boldsymbol{X}^{(s)}(k)\right\|_F + \boldsymbol{\alpha}_{s,3}^{(l)}\tau\mu\left\|\boldsymbol{W}_s^{(l)}(k+1)\Phi(\boldsymbol{X}^{(s)}(k+1))-\boldsymbol{W}_s^{(l)}(k)\Phi(\boldsymbol{X}^{(s)}(k))\right\|_F\right]
\end{aligned}
$$

Then we bound the second term carefully:

$$
\begin{aligned}
&\left\|\boldsymbol{W}_s^{(l)}(k+1)\Phi(\boldsymbol{X}^{(s)}(k+1)) - \boldsymbol{W}_s^{(l)}(k)\Phi(\boldsymbol{X}^{(s)}(k))\right\|_F \\
&= \left\|\boldsymbol{W}_s^{(l)}(k+1)(\Phi(\boldsymbol{X}^{(s)}(k+1)) - \Phi(\boldsymbol{X}^{(s)}(k)))\right\|_F + \left\|(\boldsymbol{W}_s^{(l)}(k+1) - \boldsymbol{W}_s^{(l)}(k))\Phi(\boldsymbol{X}^{(s)}(k))\right\|_F \\
&\le \sqrt{k_c}\left\|\boldsymbol{W}_s^{(l)}(k+1)\right\|_F\left\|\boldsymbol{X}^{(s)}(k+1) - \boldsymbol{X}^{(s)}(k)\right\|_F + \sqrt{k_c}\left\|\boldsymbol{W}_s^{(l)}(k+1) - \boldsymbol{W}_s^{(l)}(k)\right\|_F\left\|\boldsymbol{X}^{(s)}(k)\right\|_F
\end{aligned}
$$

By using Lemma 11 and Lemma 10, we have

$$
\begin{aligned}
\|\boldsymbol{X}^{(s)}(k)\| &\le \|\boldsymbol{X}_i^{(l)}(k) - \boldsymbol{X}_i^{(l)}(0)\|_F + \|\boldsymbol{X}_i^{(l)}(0)\|_F \\
&\le c_{x0} + \left(1+\boldsymbol{\alpha}_2+\boldsymbol{\alpha}_3\mu\sqrt{k_c}(\widetilde{r}+c_{w0})\right)^l\mu\sqrt{k_c}\widetilde{r} \overset{①}{\le} 2c_{x0},
\end{aligned}
$$

where $\alpha_2 = \max_{s,l} \alpha_{s,2}^{(l)}$ and $\alpha_3 = \max_{s,l} \alpha_{s,3}^{(l)}$, and $c_{x0} \geq 1$ is given in Lemma 10. ① holds since in Lemma 13, we set $m$ large enough such that $\widetilde{r}$ is enough small.

Besides, Lemma E.7 shows that

$$\|\boldsymbol{W}_s^{(l)}(k+1) - \boldsymbol{W}_s^{(l)}(k)\|_F \leq \frac{4c\eta\alpha_{s,3}^{(l)}\mu c_{x0}c_{w0}\sqrt{k_c}}{\sqrt{n}} \|\boldsymbol{u}(k) - \boldsymbol{y}\|_F,$$

where $c = \left(1 + \alpha_2 + 2\alpha_3\mu\sqrt{k_c}c_{w0}\right)^l$ with $\alpha_2 = \max_{s,l} \alpha_{s,2}^{(l)}$ and $\alpha_3 = \max_{s,l} \alpha_{s,3}^{(l)}$. Combing all results yields

$$\left\|\boldsymbol{W}_s^{(l)}(k+1)\Phi(\boldsymbol{X}^{(s)}(k+1)) - \boldsymbol{W}_s^{(l)}(k)\Phi(\boldsymbol{X}^{(s)}(k))\right\|_F$$

$$\leq 2\sqrt{k_c m}c_{w0}\left\|\boldsymbol{X}^{(s)}(k+1) - \boldsymbol{X}^{(s)}(k))\right\|_F + \frac{8c\eta\alpha_{s,3}^{(l)}\mu c_{x0}^2 c_{w0}k_c}{\sqrt{n}} \|\boldsymbol{u}(k) - \boldsymbol{y}\|_F.$$

Thus, we can further obtain

$$\left\|\boldsymbol{X}^{(l)}(k+1) - \boldsymbol{X}^{(l)}(k)\right\|_F$$

$$\leq \sum_{s=0}^{l-1} \left[\left(\alpha_{s,2}^{(l)} + 2\sqrt{k_c}c_{w0}\alpha_{s,3}^{(l)}\mu\right)\left\|\boldsymbol{X}^{(s)}(k+1) - \boldsymbol{X}^{(s)}(k))\right\|_F + \frac{8\tau c\eta(\alpha_{s,3}^{(l)})^2\mu^2 c_{x0}^2 c_{w0}k_c}{\sqrt{n}} \|\boldsymbol{u}(k) - \boldsymbol{y}\|_F\right]$$

$$\overset{①}{\leq} \sum_{s=0}^{l-1} \left[\left(\alpha_2 + 2\sqrt{k_c}c_{w0}\alpha_3\mu\right)\left\|\boldsymbol{X}^{(s)}(k+1) - \boldsymbol{X}^{(s)}(k))\right\|_F + \frac{8\tau c\eta(\alpha_3)^2\mu^2 c_{x0}^2 c_{w0}k_c}{\sqrt{n}} \|\boldsymbol{u}(k) - \boldsymbol{y}\|_F\right]$$

$$\leq \left(1 + \alpha_2 + 2\sqrt{k_c}c_{w0}\alpha_3\mu\right)^l \left(\left\|\boldsymbol{X}^{(0)}(k+1) - \boldsymbol{X}^{(0)}(k))\right\|_F + \frac{8\tau c\eta(\alpha_3)^2\mu^2 c_{x0}^2 c_{w0}k_c}{(\alpha_2 + 2\sqrt{k_c}c_{w0}\alpha_3\mu)\sqrt{n}} \|\boldsymbol{u}(k) - \boldsymbol{y}\|_F\right)$$

$$\leq \left(1 + \alpha_2 + 2\sqrt{k_c}c_{w0}\alpha_3\mu\right)^l \left(\frac{4c\tau\eta\mu^2 c_{x0}c_{w0}k_c}{\sqrt{n}} + \frac{8\tau c\eta(\alpha_3)^2\mu^2 c_{x0}^2 c_{w0}k_c}{(\alpha_2 + 2\sqrt{k_c}c_{w0}\alpha_3\mu)\sqrt{n}}\right) \|\boldsymbol{u}(k) - \boldsymbol{y}\|_F$$

$$\leq \left(1 + \alpha_2 + 2\sqrt{k_c}c_{w0}\alpha_3\mu\right)^l \left(1 + \frac{2(\alpha_3)^2 c_{x0}}{(\alpha_2 + 2\sqrt{k_c}c_{w0}\alpha_3\mu)\sqrt{n}}\right) \frac{4c\tau\eta\mu^2 c_{x0}c_{w0}k_c}{\sqrt{n}} \|\boldsymbol{u}(k) - \boldsymbol{y}\|_F.$$

The proof is completed. $\qquad\square$

### E.8 Proof of Lemma 15

*Proof.* In Lemma 13, we have show

$$\max\left(\|\boldsymbol{W}^{(0)}(t) - \boldsymbol{W}^{(0)}(0)\|_F, \|\boldsymbol{W}_s^{(l)}(t) - \boldsymbol{W}_s^{(l)}(0)\|_F, \|\boldsymbol{W}_s(t) - \boldsymbol{W}_s(0)\|_F\right) \leq \sqrt{m}\widetilde{r} \leq \sqrt{m}c_{w0}.$$
(27)

Note $= \frac{1}{\sqrt{m}}$. In this way, from Lemma 13, we have

$$\left\|\boldsymbol{W}^{(0)}(t)\right\|_F \leq \left\|\boldsymbol{W}^{(0)}(t) - \boldsymbol{W}^{(0)}(0)\right\|_F + \left\|\boldsymbol{W}^{(0)}(0)\right\|_F \leq 2\sqrt{m}c_{w0},$$

$$\left\|\boldsymbol{W}_s^{(l)}(t)\right\|_F \leq \left\|\boldsymbol{W}_s^{(l)}(t) - \boldsymbol{W}_s^{(l)}(0)\right\|_F + \left\|\boldsymbol{W}_s^{(l)}(0)\right\|_F \leq 2\sqrt{m}c_{w0},$$

$$\|\boldsymbol{W}_h(t)\|_F \leq \|\boldsymbol{W}_h(t) - \boldsymbol{W}_h(0)\|_F + \|\boldsymbol{W}_h(0)\|_F \leq 2\sqrt{m}c_{w0}$$

In Lemma 10, we show that when Eqn. (27) holds which is proven in Lemma 13, then $\|\boldsymbol{X}_i^{(l)}(0)\|_F \leq c_{x0}$. Under Eqn. (10), Lemma 11 shows

$$\|\boldsymbol{X}_i^{(l)}(k) - \boldsymbol{X}_i^{(l)}(0)\|_F \leq \left(1 + \alpha_2 + 2\alpha_3\mu\sqrt{k_c}c_{w0}\right)^l \mu\sqrt{k_c}\widetilde{r} \overset{①}{\leq} c_{x0},$$

where ① holds since in Lemma 13, we set $m = \mathcal{O}\left(\frac{k_c^2 c_{w0}^2\|\boldsymbol{y}-\boldsymbol{u}(0)\|_2^2}{\lambda^2 n}\left(1 + \alpha_2 + 2\alpha_3\mu\sqrt{k_c}c_{w0}\right)^{4h}\right)$ such that

$$\widetilde{r} = \frac{8c_{x0}\|\boldsymbol{y} - \boldsymbol{u}(0)\|_2}{\lambda\sqrt{mn}} \max\left(1, 2\left(1 + \alpha_2 + 2\alpha_3\mu\sqrt{k_c}c_{w0}\right)^l \alpha_{s,3}^{(l)}\mu\sqrt{k_c}c_{w0}\right)$$

$$\leq \frac{c_{x0}}{\left(1 + \alpha_2 + 2\alpha_3\mu\sqrt{k_c}c_{w0}\right)^l \mu\sqrt{k_c}}.$$

Therefore, we have

$$\left\|\boldsymbol{X}_i^{(l)}(k)\right\|_F \leq \|\boldsymbol{X}_i^{(l)}(k) - \boldsymbol{X}_i^{(l)}(0)\|_F + \left\|\boldsymbol{X}_i^{(l)}(0)\right\|_F \leq 2c_{x0}.$$

The proof is completed. $\qquad\square$

### E.9   Proof of Lemma 16

*Proof.* We first consider $l = 0$. Specifically, we have

$$
\begin{aligned}
\|\boldsymbol{X}_i^{(0)}(k) - \boldsymbol{X}_i^{(0)}(0)\|_F ={} & \tau \left\| \sigma(\boldsymbol{W}^{(0)}(k)\Phi(\boldsymbol{X}_i)) - \sigma(\boldsymbol{W}^{(0)}(0)\Phi(\boldsymbol{X}_i)) \right\|_F \\
\leq{} & \tau\mu \left\| \boldsymbol{W}^{(0)}(k) - \boldsymbol{W}^{(0)}(0) \right\|_F \|\Phi(\boldsymbol{X}_i)\|_F \\
\overset{①}{\leq}{} & \tau\mu\sqrt{k_c} \left\| \boldsymbol{W}^{(0)}(k) - \boldsymbol{W}^{(0)}(0) \right\|_F \\
\overset{②}{\leq}{} & \mu\sqrt{k_c}\widetilde{r},
\end{aligned}
$$

where ① holds since $\|\Phi(\boldsymbol{X}_i)\|_F \leq \sqrt{k_c}\|\boldsymbol{X}_i\|_F \leq \sqrt{k_c}$ and the results in Lemma 13 that $\left\| \boldsymbol{W}^{(0)}(k) - \boldsymbol{W}^{(0)}(0) \right\|_F \leq \sqrt{m}\widetilde{r}.$

Then we consider $l \geq 1$. According to the definition, we have

$$
\begin{aligned}
& \|\boldsymbol{X}_i^{(l)}(k) - \boldsymbol{X}_i^{(l)}(0)\|_F \\
={} & \left\| \sum_{s=0}^{l-1} \left( \boldsymbol{\alpha}_{s,2}^{(l)}(\boldsymbol{X}_i^{(s)}(k) - \boldsymbol{X}_i^{(s)}(0)) + \boldsymbol{\alpha}_{s,3}^{(l)}\tau \left( \sigma(\boldsymbol{W}_s^{(l)}(k)\Phi(\boldsymbol{X}_i^{(s)}(k))) - \sigma(\boldsymbol{W}_s^{(l)}(0)\Phi(\boldsymbol{X}_i^{(s)}(0))) \right) \right) \right\|_F \\
\leq{} & \sum_{s=0}^{l-1} \left[ \boldsymbol{\alpha}_{s,2}^{(l)} \left\| \boldsymbol{X}_i^{(s)}(k) - \boldsymbol{X}_i^{(s)}(0) \right\|_F + \boldsymbol{\alpha}_{s,3}^{(l)}\tau \left\| \sigma(\boldsymbol{W}_s^{(l)}(k)\Phi(\boldsymbol{X}_i^{(s)}(k))) - \sigma(\boldsymbol{W}_s^{(l)}(0)\Phi(\boldsymbol{X}_i^{(s)}(0))) \right\|_F \right] \\
\leq{} & \sum_{s=0}^{l-1} \left[ \boldsymbol{\alpha}_{s,2}^{(l)} \left\| \boldsymbol{X}_i^{(s)}(k) - \boldsymbol{X}_i^{(s)}(0) \right\|_F + \boldsymbol{\alpha}_{s,3}^{(l)}\tau\mu \left\| \boldsymbol{W}_s^{(l)}(k)\Phi(\boldsymbol{X}_i^{(s)}(k)) - \boldsymbol{W}_s^{(l)}(0)\Phi(\boldsymbol{X}_i^{(s)}(0)) \right\|_F \right].
\end{aligned}
$$

Then we bound

$$
\begin{aligned}
& \left\| \boldsymbol{W}_s^{(l)}(k)\Phi(\boldsymbol{X}_i^{(s)}(k)) - \boldsymbol{W}_s^{(l)}(0)\Phi(\boldsymbol{X}_i^{(s)}(0)) \right\|_F \\
\leq{} & \left\| (\boldsymbol{W}_s^{(l)}(k) - \boldsymbol{W}_s^{(l)}(0))\Phi(\boldsymbol{X}_i^{(s)}(k)) \right\|_F + \left\| \boldsymbol{W}_s^{(l)}(0)(\Phi(\boldsymbol{X}_i^{(s)}(k)) - \Phi(\boldsymbol{X}_i^{(s)}(0))) \right\|_F \\
\leq{} & \left\| \boldsymbol{W}_s^{(l)}(k) - \boldsymbol{W}_s^{(l)}(0) \right\|_F \left\| \Phi(\boldsymbol{X}_i^{(s)}(k)) \right\|_F + \left\| \boldsymbol{W}_s^{(l)}(0) \right\|_F \left\| \Phi(\boldsymbol{X}_i^{(s)}(k)) - \Phi(\boldsymbol{X}_i^{(s)}(0)) \right\|_F \\
\overset{①}{\leq}{} & 2\sqrt{k_c m}c_{x0}\widetilde{r} + 2\sqrt{k_c m}c_{w0} \left\| \boldsymbol{X}_i^{(s)}(k) - \boldsymbol{X}_i^{(s)}(0) \right\|_F,
\end{aligned}
$$

where ① holds since Lemma 13 shows $\left\| \boldsymbol{W}^{(0)}(k) - \boldsymbol{W}^{(0)}(0) \right\|_F \leq \sqrt{m}\widetilde{r}$ and Lemma 15 shows $\left\| \boldsymbol{X}_i^{(s)}(k) \right\|_F \leq 2c_{x0}$ and $\left\| \boldsymbol{W}_s^{(l)}(0) \right\|_F \leq 2\sqrt{m}c_{w0}.$

In this way, we have

$$
\begin{aligned}
& \|\boldsymbol{X}_i^{(l)}(k) - \boldsymbol{X}_i^{(l)}(0)\|_F \\
\leq{} & \sum_{s=0}^{l-1} \left[ \left( \boldsymbol{\alpha}_{s,2}^{(l)} + 2\boldsymbol{\alpha}_{s,3}^{(l)}\mu\sqrt{k_c}c_{w0} \right) \left\| \boldsymbol{X}_i^{(s)}(k) - \boldsymbol{X}_i^{(s)}(0) \right\|_F + 2\boldsymbol{\alpha}_{s,3}^{(l)}\mu\sqrt{k_c}c_{x0}\widetilde{r} \right] \\
\overset{①}{\leq}{} & \sum_{s=0}^{l-1} \left[ \left( \boldsymbol{\alpha}_2 + 2\boldsymbol{\alpha}_3\mu\sqrt{k_c}c_{w0} \right) \left\| \boldsymbol{X}_i^{(s)}(k) - \boldsymbol{X}_i^{(s)}(0) \right\|_F + 2\boldsymbol{\alpha}_3\mu\sqrt{k_c}c_{x0}\widetilde{r} \right] \\
\overset{②}{\leq}{} & c \left[ \left\| \boldsymbol{X}_i^{(0)}(k) - \boldsymbol{X}_i^{(s)}(0) \right\|_F + 2\boldsymbol{\alpha}_3\mu\sqrt{k_c}c_{x0}\widetilde{r} \right] \\
={} & c(1 + 2\boldsymbol{\alpha}_3 c_{x0})\mu\sqrt{k_c}\widetilde{r}
\end{aligned}
$$

where ① and ② hold by using $c = \left( 1 + \boldsymbol{\alpha}_2 + 2\boldsymbol{\alpha}_3\mu\sqrt{k_c}c_{w0} \right)^l$ with $\boldsymbol{\alpha}_2 = \max_{s,l} \boldsymbol{\alpha}_{s,2}^{(l)}$ and $\boldsymbol{\alpha}_3 = \max_{s,l} \boldsymbol{\alpha}_{s,3}^{(l)}$. The proof is completed. $\qquad\square$

### E.10   Proof of Lemma 17

*Proof.* For this proof, we need to use the results in other lemmas. Specifically, Lemma 13

$$
\|\boldsymbol{W}^{(0)}(t) - \boldsymbol{W}^{(0)}(0)\|_F \leq \sqrt{m}\widetilde{r}, \ \|\boldsymbol{W}_s^{(l)}(t) - \boldsymbol{W}_s^{(l)}(0)\|_F \leq \sqrt{m}\widetilde{r}, \ \|\boldsymbol{W}_s(t) - \boldsymbol{W}_s(0)\|_F \leq \sqrt{m}\widetilde{r},
\tag{28}
$$

where $c = \left(1 + \boldsymbol{\alpha}_2 + 2\boldsymbol{\alpha}_3\mu\sqrt{k_c}c_{w0}\right)^l$ with $\boldsymbol{\alpha}_2 = \max_{s,l}\boldsymbol{\alpha}_{s,2}^{(l)}$ and $\boldsymbol{\alpha}_3 = \max_{s,l}\boldsymbol{\alpha}_{s,3}^{(l)}$. Based on this, Lemma 15 further shows

$$\left\|\boldsymbol{W}^{(0)}(k)\right\|_F \le 2\sqrt{m}c_{w0}, \quad \left\|\boldsymbol{W}_s^{(l)}(k)\right\|_F \le 2\sqrt{m}c_{w0}, \quad \|\boldsymbol{W}_s(k)\|_F \le 2\sqrt{m}c_{w0}, \quad \left\|\boldsymbol{X}_i^{(l)}(k)\right\|_F \le 2c_{x0}. \tag{29}$$

Next, Lemma 16 also proves

$$\|\boldsymbol{X}_i^{(l)}(k) - \boldsymbol{X}_i^{(l)}(0)\|_F \le c(1 + 2\boldsymbol{\alpha}_3 c_{x0})\mu\sqrt{k_c}\widetilde{r}.$$

Then we can easily obtain our result:

$$
\begin{aligned}
|u_i(k) - u_i(0)| &= \left|\sum_{s=1}^h \langle \boldsymbol{W}_s(k), \boldsymbol{X}_i^{(l)}(k)\rangle - \langle \boldsymbol{W}_s(0), \boldsymbol{X}_i^{(l)}(0)\rangle\right| \\
&\le \sum_{s=1}^h \left|\langle \boldsymbol{W}_s(k) - \boldsymbol{W}_s(0), \boldsymbol{X}_i^{(l)}(k)\rangle + \langle \boldsymbol{W}_s(0), \boldsymbol{X}_i^{(l)}(k) - \boldsymbol{X}_i^{(l)}(0)\rangle\right| \\
&\le \sum_{s=1}^h 2\sqrt{m}\widetilde{r}c_{x0} + 2\sqrt{m}c_{w0}c(1 + 2\boldsymbol{\alpha}_3 c_{x0})\mu\sqrt{k_c}\widetilde{r} \\
&= 2\sqrt{m}h\left(c_{x0} + c_{w0}c(1 + 2\boldsymbol{\alpha}_3 c_{x0})\mu\sqrt{k_c}\right)\widetilde{r}.
\end{aligned}
$$

Then we look at the second part. We first look at $l = h$:

$$
\begin{aligned}
\left\|\frac{\partial\ell}{\partial\boldsymbol{X}_i^{(l)}(k)} - \frac{\partial\ell}{\partial\boldsymbol{X}_i^{(l)}(0)}\right\|_F &= \|(u_i(k) - y_i)\boldsymbol{W}_l(k) - (u_i(0) - y_i)\boldsymbol{W}_l(0)\|_F \\
&= |u_i(k) - y_i|\,\|\boldsymbol{W}_l(k)\|_F + |u_i(0) - y_i|\,\|\boldsymbol{W}_l(0)\|_F \\
&\le \|(u_i(k) - u_i(0))\boldsymbol{W}_l(k)\|_F + \|(u_i(0) - y_i)(\boldsymbol{W}_l(k) - \boldsymbol{W}_l(0))\|_F \\
&\le |u_i(k) - u_i(0)|\,\|\boldsymbol{W}_l(k)\|_F + |u_i(0) - y_i|\,\|(\boldsymbol{W}_l(k) - \boldsymbol{W}_l(0))\|_F \\
&\le 4\sqrt{m}\widetilde{r}\left(c_{w0}\sqrt{m}h\left(c_{x0} + c_{w0}c(1 + 2\boldsymbol{\alpha}_3 c_{x0})\mu\sqrt{k_c}\right) + |u_i(0) - y_i|\right).
\end{aligned} \tag{30}
$$

Then we consider $l < h$. According to the definitions in Lemma 8, we have

$$\frac{\partial\ell}{\partial\boldsymbol{X}^{(l)}} = (u - \boldsymbol{y})\boldsymbol{W}_l + \sum_{s=l+1}^h \left(\boldsymbol{\alpha}_{l,2}^{(s)}\frac{\partial\ell}{\partial\boldsymbol{X}^{(s)}} + \boldsymbol{\alpha}_{l,3}^{(s)}\tau\Psi\left((\boldsymbol{W}_l^{(s)})^\top\left(\sigma'\left(\boldsymbol{W}_l^{(s)}\Phi(\boldsymbol{X}^{(l)})\right)\odot\frac{\partial\ell}{\partial\boldsymbol{X}^{(s)}}\right)\right)\right).$$

In this way, we can upper bound

$$
\left\|\frac{\partial\ell}{\partial\boldsymbol{X}_i^{(l)}(k)} - \frac{\partial\ell}{\partial\boldsymbol{X}_i^{(l)}(0)}\right\|_F
$$
$$
= \|(u_i(k) - y_i)\boldsymbol{W}_l(k) - (u_i(0) - y_i)\boldsymbol{W}_l(0)\|_F + \sum_{s=l+1}^h \boldsymbol{\alpha}_{l,2}^{(s)}\left\|\frac{\partial\ell}{\partial\boldsymbol{X}_i^{(s)}(k)} - \frac{\partial\ell}{\partial\boldsymbol{X}_i^{(s)}(k)}\right\|_F + \sum_{s=l+1}^h \boldsymbol{\alpha}_{l,3}^{(s)}\tau\sqrt{k_c}D,
$$

where $D = \left\|\boldsymbol{A}_k^\top(\boldsymbol{B}_k \odot \boldsymbol{C}_k) - \boldsymbol{A}_0^\top(\boldsymbol{B}_0 \odot \boldsymbol{C}_0)\right\|_F$ in which $\boldsymbol{A}_k = \boldsymbol{W}_l^{(s)}(k), \boldsymbol{B}_k = \sigma'\left(\boldsymbol{W}_l^{(s)}(k)\Phi(\boldsymbol{X}_i^{(l)}(k))\right), \boldsymbol{C}_k = \frac{\partial\ell}{\partial\boldsymbol{X}_i^{(s)}(k)}$. Similar to Eqn. (30), we have

$$
\begin{aligned}
\|(u_i(k) - y_i)\boldsymbol{W}_l(k) &- (u_i(0) - y_i)\boldsymbol{W}_l(0)\|_F \\
&\le 4\sqrt{m}\widetilde{r}\left(c_{w0}\sqrt{m}h\left(c_{x0} + c_{w0}c(1 + 2\boldsymbol{\alpha}_3 c_{x0})\mu\sqrt{k_c}\right) + |u_i(0) - y_i|\right).
\end{aligned}
$$

Then, we can bound $D$ as follows:

$$
\begin{aligned}
D &= \left\|(\boldsymbol{A}_k - \boldsymbol{A}_0)^\top(\boldsymbol{B}_0 \odot \boldsymbol{C}_0)\right\|_F + \left\|\boldsymbol{A}_k^\top(\boldsymbol{B}_k \odot \boldsymbol{C}_k - \boldsymbol{B}_0 \odot \boldsymbol{C}_0)\right\|_F \\
&\le \|\boldsymbol{A}_k - \boldsymbol{A}_0\|_F\|\boldsymbol{B}_0 \odot \boldsymbol{C}_0\|_F + \|\boldsymbol{A}_k\|_F\|\boldsymbol{B}_k \odot \boldsymbol{C}_k - \boldsymbol{B}_0 \odot \boldsymbol{C}_0\|_F \\
&\overset{\text{①}}{\le} \mu\sqrt{m}\widetilde{r}\|\boldsymbol{C}_0\|_2 + 2\sqrt{m}c_{w0}\|\boldsymbol{B}_k \odot \boldsymbol{C}_k - \boldsymbol{B}_0 \odot \boldsymbol{C}_0\|_F
\end{aligned}
$$

where ① uses the results in Eqns. (29) and (28). The remaining work is to bound

$$\|\boldsymbol{B}_k \odot \boldsymbol{C}_k - \boldsymbol{B}_0 \odot \boldsymbol{C}_0\|_F = \|\boldsymbol{B}_k \odot (\boldsymbol{C}_k - \boldsymbol{C}_0)\|_F + \|(\boldsymbol{B}_k - \boldsymbol{B}_0) \odot \boldsymbol{C}_0\|_F$$

$$\overset{①}{\leq} \mu \|\boldsymbol{C}_k - \boldsymbol{C}_0\|_F + \rho \left\|\boldsymbol{W}_l^{(s)}(k)\Phi(\boldsymbol{X}_i^{(l)}(k)) - \boldsymbol{W}_l^{(s)}(0)\Phi(\boldsymbol{X}_i^{(l)}(0))\right\|_F \|\boldsymbol{C}_0\|_\infty$$

where ① uses the assumption that the activation function $\sigma(\cdot)$ is $\mu$-Lipschitz and $\rho$-smooth. Note $\|\boldsymbol{C}_0\|_\infty$ is a constant, since it is the gradient norm at the initialization which does not involves the algorithm updating. Recall Lemma 11 shows

$$\left\|\boldsymbol{W}_s^{(l)}(k)\Phi(\boldsymbol{X}^{(s)}(k)) - \boldsymbol{W}_s^{(l)}(0)\Phi(\boldsymbol{X}^{(s)}(0))\right\|_F \leq \frac{1}{\boldsymbol{\alpha}_3}\left(1 + \boldsymbol{\alpha}_2 + \boldsymbol{\alpha}_3 \mu \sqrt{k_c}(r + c_{w0})\right)^l \sqrt{k_c m}\widetilde{r},$$

where $\boldsymbol{\alpha}_2 = \max_{s,l}\boldsymbol{\alpha}_{s,2}^{(l)}$ and $\boldsymbol{\alpha}_3 = \max_{s,l}\boldsymbol{\alpha}_{s,3}^{(l)}$, and $c_{x0} \geq 1$ is given in Lemma 10. Then we upper bound

$$\left\|\boldsymbol{W}_l^{(s)}(k)\Phi(\boldsymbol{X}_i^{(l)}(k)) - \boldsymbol{W}_l^{(s)}(0)\Phi(\boldsymbol{X}_i^{(l)}(0))\right\|_F \leq \frac{1}{\boldsymbol{\alpha}_3}\left(1 + \boldsymbol{\alpha}_2 + \boldsymbol{\alpha}_3 \mu \sqrt{k_c}(r + c_{w0})\right)^l \sqrt{k_c m}\widetilde{r}.$$

Therefore, we have

$$D \leq \mu\sqrt{m}\widetilde{r}\|\boldsymbol{C}_0\|_2 + 2\sqrt{m}c_{w0}\left(\mu\|\boldsymbol{C}_k - \boldsymbol{C}_0\|_F + \frac{\rho\|\boldsymbol{C}_0\|_\infty}{\boldsymbol{\alpha}_3}\left(1 + \boldsymbol{\alpha}_2 + \boldsymbol{\alpha}_3\mu\sqrt{k_c}(r + c_{w0})\right)^l \sqrt{k_c m}\widetilde{r}\right)$$

By combining the above results, we have

$$\left\|\frac{\partial \ell}{\partial \boldsymbol{X}_i^{(l)}(k)} - \frac{\partial \ell}{\partial \boldsymbol{X}_i^{(l)}(0)}\right\|_F$$

$$\leq c_1 + \sum_{s=l+1}^{h}\left[\left(\boldsymbol{\alpha}_{l,2}^{(s)} + 2\boldsymbol{\alpha}_{l,3}^{(s)}\sqrt{k_c}\mu c_{w0}\right)\left\|\frac{\partial \ell}{\partial \boldsymbol{X}_i^{(s)}(k)} - \frac{\partial \ell}{\partial \boldsymbol{X}_i^{(s)}(k)}\right\|_F + c_2\right]$$

$$\leq c_1 + \sum_{s=l+1}^{h}\left[\left(\boldsymbol{\alpha}_2 + 2\boldsymbol{\alpha}_3\sqrt{k_c}\mu c_{w0}\right)\left\|\frac{\partial \ell}{\partial \boldsymbol{X}_i^{(s)}(k)} - \frac{\partial \ell}{\partial \boldsymbol{X}_i^{(s)}(k)}\right\|_F + c_3\right]$$

$$\leq \left(1 + \boldsymbol{\alpha}_2 + 2\boldsymbol{\alpha}_3\sqrt{k_c}\mu c_{w0}\right)^l\left[\left\|\frac{\partial \ell}{\partial \boldsymbol{X}_i^{(h)}(k)} - \frac{\partial \ell}{\partial \boldsymbol{X}_i^{(h)}(0)}\right\|_F + c_3\right]$$

where $c_1 = 4\sqrt{m}\widetilde{r}\left(c_{w0}\sqrt{m}h\left(c_{x0} + c_{w0}c(1 + 2\boldsymbol{\alpha}_3 c_{x0})\mu\sqrt{k_c}\right) + |u_i(0) - y_i|\right)$, $c_2 = \boldsymbol{\alpha}_{l,3}^{(s)}\left(\mu\widetilde{r}\|\boldsymbol{C}_0\|_2 + 2c_{w0}\frac{\rho\|\boldsymbol{C}_0\|_\infty}{\boldsymbol{\alpha}_3}\left(1 + \boldsymbol{\alpha}_2 + \boldsymbol{\alpha}_3\mu\sqrt{k_c}(r + c_{w0})\right)^l\sqrt{k_c m}\widetilde{r}\right)$ and $c_3 = \boldsymbol{\alpha}_3\left(\mu\widetilde{r}\|\boldsymbol{C}_0\|_2 + 2c_{w0}\frac{\rho\|\boldsymbol{C}_0\|_\infty}{\boldsymbol{\alpha}_3}\left(1 + \boldsymbol{\alpha}_2 + \boldsymbol{\alpha}_3\mu\sqrt{k_c}(r + c_{w0})\right)^l\sqrt{k_c m}\widetilde{r}\right)$. Consider $\|\boldsymbol{C}_0\|_2 = \mathcal{O}\left(\sqrt{m}\right)$, for brevity, we ignore constants and obtain

$$\left\|\frac{\partial \ell}{\partial \boldsymbol{X}_i^{(l)}(k)} - \frac{\partial \ell}{\partial \boldsymbol{X}_i^{(l)}(0)}\right\|_F \leq c_1 c\boldsymbol{\alpha}_3 c_{w0}^2 c_{x0}\rho k_c m\widetilde{r},$$

where $c = \left(1 + \boldsymbol{\alpha}_2 + 2\boldsymbol{\alpha}_3\sqrt{k_c}\mu c_{w0}\right)^l$ and $c_1$ is a constant. The proof is completed. $\qquad\square$

### E.11  Proof of Lemma 18

*Proof.* By Assumption 2, each entry for the initial parameter $\boldsymbol{W}_s^{(l)}(0)$ obeys Gaussian distribution $\mathcal{N}(0,1)$. Then $\|\boldsymbol{W}_s^{(l)}(0)\|_F^2$ is chi-square variable with freedom degree $k_c pm$. In this way, by using Lemma 4, we have

$$\mathbb{P}\left(\|\boldsymbol{W}_s^{(l)}(0)\|_F^2 - k_c pm \geq 2\sqrt{k_c pmt} + 2t\right) \leq \exp(-t).$$

Therefore, with probability at least $1 - \frac{\delta}{2h(h+3)}$, we can obtain

$$\|\boldsymbol{W}_s^{(l)}(0)\|_F \leq \sqrt{k_c pm + 2\sqrt{k_c pm \log(2h(h+3)/\delta)} + 2\log(2h(h+3)/\delta)} \leq \sqrt{m}c_{w0},$$

where $c_{w0} \sim \sqrt{k_c p}$ is a constant. Note here we focus on $m$ more than $p$ and $k_c$, since $m$ is much larger than $p$ and $k_c$ which is introduced in subsequent analysis.

By using the same method, we can prove that with probability at least $1 - \frac{\delta}{2h(h+3)}$,

$$\|\boldsymbol{W}^0(0)\|_F \leq \sqrt{m} c_{w0} \quad \text{and} \quad \|\boldsymbol{W}_s(0)\|_F \leq \sqrt{m} c_{w0}$$

In this way, with probability at least $\left(1 - \frac{\delta}{2h(h+3)}\right)^{\frac{h(h+3)}{2}} \geq 1 - \frac{\delta}{2h(h+3)} \frac{h(h+3)}{2} = 1 - \delta/4$, these results hold at the same time. The proof is completed. $\qquad\square$