[Reviews · NeurIPS 2020]

Review 1

Summary and Contributions: This paper addresses an important problem that existed in the previous gradient-based NAS (e.g., DARTS) that discovered architectures are often dominated by skip connections which usually leads to poor performance. The authors provide the theoretical analysis of the intrinsic reasons that cause such an issue, which reveals that the architectures with more skip connections converge fast in DARTS. Inspired by that, the authors further propose a novel path-regularized method, including a differential group-structured sparse binary gate on each operation and a path-depth-wise regularization, to better train the gradient-based NAS problem. The experiments on CIFAR-10/100 and ImageNet demonstrate the promising performance of the proposed method.

Strengths: This paper provides a theoretical analysis of why DARTS prefers skip connections in the discovered architecture, and then proposes a path-regularized method to solve this issue. This paper is well written and is valuable in terms of both theoretical analysis and algorithm implementation. Specifically, this paper shows theoretically that it is fundamentally because that the convergence rate of the gradient-based NAS algorithm (e.g., DARTS) depends much heavier on the weights of skip connections than other types of operations, this makes the architectures with more skip connections converging faster than other candidates, and this selected by DARTS. The authors address such an issue by avoiding unfair operation competition (between the skip connection and other operations, such as convolutions) and encouraging exploration in the search space, which are then implemented by a differential group-structured sparse binary gate and a path-depth-wise regularization, respectively. The codes are provided.

Weaknesses: 1. Can the authors clarify the chosen of the learning rate /eta? For example, \eta should be at the order of 1/\sqrt{m}/h^3/\lambda, where m denotes the channel number, h denotes the depth, and lambda denotes the smallest eigenvalue of the data matrix. So in most cases, this setting will give a very small learning rate, which is not very consistent with the practice setting. 2. I suggest the authors move more derivations to the supplement to avoid using too much v-spacing. This also helps to better express the key ideas in the main text.

Correctness: Yes.

Clarity: Yes.

Relation to Prior Work: Yes.

Reproducibility: Yes

Additional Feedback: I have read the authors rebuttal, which well addressed my concerns with further clarified technical details, therefore, I remain my initial rating as a strong accept.


Review 2

Summary and Contributions: This paper aims to resolve the dominated skip connections issue in DARTS-based NAS methods. Specifically, the authors first theoretically analyze the intrinsic reasons for the dominated skip connections by analyzing its convergence behaviors in DARTS cell. Based on the theoretical results, the authors propose a Path-regularized DARTS method by regularizing the competition between skip connections and other operations. It is worth noting that the proposed method is theoretically guaranteed. However, the experimental part could be further improved to make it more convincing.

Strengths: 1 The author theoretically prove that DARTS prefers skip connection to other types of operations (i.e., convolution and zero), since more skip connections the faster convergence. 2 Based on theoretical analysis, the authors propose a Path-regularized DRATS method (namely PR-DARTS), which aims to avoid direct competition between skip connection and other types of operations. 3 All components in the proposed PR-DARTS are theory-inspired and have theoretical guarantees. Experimental results also demonstrate the effectiveness of PR-DARTS.

Weaknesses: 1. The authors theoretically prove that “more skip connections the faster convergence” and “shallow cells benefit faster convergence rate than deep cells”. Is there any experimental evidence to verify these claims? 2. In the theoretical analysis of Section 3.2, the authors do not consider pooling operation since it is also dominated by skip connections. However, does pooling operations also have a slower convergence rate than skip connections? 3. In lines 175-178, the authors mentioned that skip connection in shared path and convolution in private path can benefit the Gram matrix singularity of networks. Thus, the convergence rate can be greatly improved. It would be stronger to provide empirical results to verify these claims. 4. Does depth-wise separable convolution and standard convolution share the same convergence results that are described in Theorem 1? 5. In Section 3.2, the authors assume that the activation function should be $\mu$-Lipschitz and $\rho$-smooth. However, in operations.py, the authors take ReLU as the activation function, which is not $\rho$-smooth. As a result, the assumption may not hold. More discussions regarding this issue should be provided. 6. Several state-of-the-art NAS methods should be compared in the paper, including [1] Cars: Continuous evolution for efficient neural architecture search. CVPR 2020. [2] Breaking the Curse of Space Explosion: Towards Efficient NAS with Curriculum Search. ICML 2020. [3] Neural Architecture Search in A Proxy Validation Loss Landscape. ICML 2020.

Correctness: Yes

Clarity: Yes

Relation to Prior Work: Yes

Reproducibility: Yes

Additional Feedback: -------After Rebuttal--------------- The authors have addressed all my concerns. This paper is of good quality and I still recommend to accept it.


Review 3

Summary and Contributions: This paper provides a theoretical perspective towards the degenerated search architectures in DARTS space where skip-connection dominates the connections. Based on that, it proposes to use a differential group-structured sparse binary gate to reduce the direct competition between skip- and non-skip connections. Furthermore, it proposed a regularization on cell depth to favor deeper models. Experiments in DARTS space on CIFAR10/100 and ImageNet confirm the effectiveness of the proposed regularizations. =============Post rebuttal================ After reviewing authors' response, I confirm this is a strong submission and keep my original rating.

Strengths: - It contains a quite thorough derivations of the relationship between Supernet convergence and the architecture parameters of different types of operators (e.g. skip, zero, and conv layers), which shows skip-connection tends to get larger weights. - The use of individual binary gate at each candidate operator is interesting, and Theorem 3 is not only sound in theory, but also provides solid foundations for designing regularization loss on group sparsity and model depth. - Experiments on CIFAR10/100 and ImageNet are properly designed, and results are quite competitive.

Weaknesses: I am mostly happy with both theoretic- and experimental aspects of this work, and did not find any obvious weakness.

Correctness: Yes.

Clarity: The paper is well written and easy to follow.

Relation to Prior Work: Yes.

Reproducibility: Yes

Additional Feedback:


Review 4

Summary and Contributions: This paper deals with the problem of network architecture search. Based on previous work DARTS, this paper proposes some theoretical analyses and corresponding regularization methods to solve the dominated skip-connection problem in DARTS and improve its performance. The contributions of this paper are as follows: (1) provide rigorous theoretical analyses and explanations as to the reasons why skip-connection dominates in DARTS. (2) Based on the theory, the paper proposes regularized group-structured sparse binary gate method and corresponding theoretical analyses to alleviate the dominated skip-connection problem. (3) the paper then proposes a path-depth-wise regularizer and corresponding theoretical analyses to encourage deep architecture search.

Strengths: The strengeths of this paper are as follows: (1) Rigorous theoretical analysis which hasn't rarely been investigated before, offering theoretical reasons for dominated skip-connection behaviors. (2) Convincing modification of DARTS based on great combination of intuitive explanation, theoretical analysis and detailed empirical results, evaluations. (3) The contribution of theoretical analysis is novel and fundamental. (4) This paper is relevant to the NeurIPS community since NAS is a key issue in artificial intelligence.

Weaknesses: The weaknesses of this paper are as follows: (1)The explanation for theorem 1 is kind of intuitive and confusing. some points are confusing: 1. The illustration of lambda_s by connection path X(0)→X(1)→· · ·→X(s)→X(h−1). From my understanding, alpha_(t,2)^(s) should represent the arrow from X(t) to X(s) while alpha_(s,3)^(h-1) should represnet the arrow from X(s) to X(h-1). So I believe for a single lambda_s, the connection path should be like X(0)→X(s), X(1)→X(s)... and X(s)→X(h−1). So the the illustration of lambda_s by connection path is unclear. 2. The explanation about residual information learned by convolution layers is unclear too. (2)The main theoretical analysis is based on the Mean square error loss assumption, which might not be practical in reality.

Correctness: The claims and methods are mostly correct. However, there might exist some over-claims in some parts. In most parts, this paper conduct theoretical analysis to prove that some settings(like dominated skip-connection and shallow network) have bigger convergence rates, but it doesn't rigourously show that the phenomenon mentioned by the papers “as some cells that fast decay the loss Fval(W, β) at the current iteration have competitive advantage over other cells that reduce Fval(W, β) slowly currently but can achieve superior final performan”. It's an intuitive and reasonable explanatoin for that since the previous setting might lead the network to be trapped in some local minima but the descriptions in the paper appears to be over-claimed.

Clarity: The paper is mostly well written. However, there exists room to improve: (1) Mentioned in the weakness, it might be helpful to detail the explanation from theorem 1 to the reasons for dominated skip-connection.Some explanations in the paper might be confusing for audiences. (2) It might be useful for audience to understand the theorem if some brief summaries (or key steps) of the proof can be discussed in the main paper rather than the appendix since the proof in appendix is relatively long and audiences might lose patience checking all of it.

Relation to Prior Work: The paper clearly discusses the differences between itself and the previous contributions. The theoretical analysis about DART's problem is novel.

Reproducibility: Yes

Additional Feedback: Some questions/suggestions: (1)Is the mean square error loss assumption necessary for the whole theoretical analysis? Can any other loss assumption satisfy the theorem in the paper?(like cross entropy) (2)How do you choose the hyperparameters for lambda_1, lambda_2, lambda_3 in the experiments, which are important for the experiments. (3)Some other baseline experiments can also be conducted, which might detail the ablation test, such as explicit regularization function without independent stochastic gate, which might show the importance of competition or cooperation among operatiions. (4)It might be clear if you can bolden the best experimental results in all tables to show the performance results.

[Author Response · NeurIPS 2020]

We thank all the reviewers for their insightful and encouraging comments, and will update revision to solve the issues.

**To Reviewer #1.** Consider channel number $m = \mathcal{O}(n^2)$ and sample number $n$ is much larger than depth $h$ in NAS,
our learning rate (LR) is $\eta = \mathcal{O}(\lambda/\sqrt{m}/h^3) = \mathcal{O}(\lambda/n)$. It indeed improves LR requirement in [18-20] which analyze
convergence of ResNet, e.g. $\eta = \mathcal{O}(\lambda/n^2)$ in [18,19] and $\eta = \mathcal{O}(\lambda/\mathrm{poly}(n))$ in [20]. As NAS has much dense connections
than ResNet, it allows larger LR. So our work makes towards the practice setting, and will continue to improve it later.

**To Reviewer #2.** 1) We empirically investigate i) more skip connections gives faster convergence and ii) shallow cells
have faster convergence rate than deep cells. We first set all operations in NAS cell (normal and reduction cells) as
convolution ($3\times3$), and randomly select $0\%$, $37.5\%$ and $62.5\%$ operations as skip connections. Next, we stack 8 NAS cells
to build a network and train on CIFAR10 with same settings. Fig. (a) demonstrates our result i). Moreover, Fig. 3 in [9]
also testifies our result i). For result ii), to simply construction, we let each
node in NAS cell only has one connection. To construct deep network, we use
convolution to connect the current node with its previous node, i.e. $0 \to 1 \to 2 \cdots$
$\to 5$. For shallow network, we connect the $i$-th node ($i = 1, \cdots, 5$) to the $0$-th
node with the same convolution. We also stack 8 cells and train them with same
setting. Fig. (b) demonstrates our result ii). We will update it into revision.

2) Pooling operations also converge more slowly than skip connections. We consider function $h(p(g(x)))$, where $g$ are
layers before pooling $p$, $h$ are subsequent layers and loss. Then we can prove ① $\|\frac{\partial h}{\partial p(g(x))} \frac{\partial p(g(x))}{\partial g(x)}\|_F^2 < \|\frac{\partial h}{\partial g(x)}\|_F^2$, where
the later denotes network $h$ using skip connection. So pooling $p$ reduces gradient and gives slower convergence. For max
pooling, it only considers the maximum pixels and ignores others, directly giving ①. For average pooling, by derivation
we have $\frac{\partial h}{\partial p(g(x))} \frac{\partial p(g(x))}{\partial g_{ij}(x)} = \frac{s_{ij}}{o^2} \frac{\partial h}{\partial g_{ij}(x)}$ with pooling size $o \times o$, where $s_{ij}(\leq o^2)$ denotes how many times $g_{ij}(x)$ attends
convolution. If pooling stride $s > 1$, then $s_{ij} < o^2$. If $s = 1$, for pixels near the edges, their $s_{ij} < o^2$. So ① always holds.
Besides, we are sure that with pooling, Theorem 1 still holds for two-layered network and shows that convergence rate
depends on skip connection heavier. For deeper networks, more efforts and time are needed for further derivation.

3) For Gram matrix singularity, we set all operations in NAS cell as convolution ($3\times3$), and randomly select $0\%$, $40\%$,
$80\%$ operations in the shared path as skip connections to obtain cells $A$, $B$ and $C$. Due to memory limitation, we use
one NAS cell and find that smallest eigenvalues of Gram matrix in $A$, $B$ and $C$ on CIFAR10 are respectively $1.1\times10^{-4}$,
$3.4\times10^{-4}$ and $5.9\times10^{-4}$, showing benefits of skip connect to singularity. Then we fix the shared path with $40\%$ skip
connections, and randomly replace $0\%$, $40\%$ and $80\%$ convolutions in private path with zero operations. Then smallest
eigenvalues become $3.4\times10^{-4}$, $1.3\times10^{-4}$ and $8.7\times10^{-5}$, showing important of convolution in private path to singularity.

4) For depth-wise separable convolution (DSC), we can expect the same convergence rate as standard convolution
(SC). Similar to SC, we formulate DSC as $D(W, X) = \sigma(W_p \Phi_p(\Phi_d(X)W_d))$. Similar to $\Phi$ in manuscript, $\Phi_d(X)$ ($\Phi_p(X)$)
rearranges features in $X$ along channel (feature) direction for depthwise convolution $\Phi_d(X)W_d$ (pointwise convolution
$W_p\Phi_p(X)$). Then we replace convolution $\mathrm{Conv}(W, X)$ in manuscript with $D(W, X)$, and follow our proof framework to
prove same results: the convergence rate replies on skip connections heavier than other types of operations.

5) ReLU is not smooth at only one point, i.e. zero. But the measure of one point is zero. So almost sure, our smoothness
assumption holds [ arXiv:1706.03175, ICML'17]. Error (%) on CIFAR10 (ImageNet) of the mentioned references [1-3]
are respectively 2.62 (24.8), 2.6 (24.6) and 2.7 (25.6). Ours is 2.31 (24.3) and thus is better. We will cite them.

**To Reviewer #4.** 1) Per your suggestion, we will use $\boldsymbol{X}^{(t)} \to \boldsymbol{X}^{(s)}$ ($0 \leq t \leq s - 1$) to better illustrate the path of $\lambda_s$. As
$\boldsymbol{X}^{(t)}(1 \leq t \leq h - 2)$ are shared by $\boldsymbol{X}^{(s)}(s \geq t)$, our subsequent explanation to Theorem 1 does not need to change.

Skip connection (SC) has formulation $X_{s+1} = X_s + F(X_s)$ where $F$ is a function, e.g. convolution. So to fit ground truth
$Y$ of $X_s$, $F$ only fits the residual $Y - X_s$ instead of $Y$. Recursively, we have $X_l = X_s + \sum_{t=s}^{l-1} F(X_t)$. Then the gradient of $X_s$
is $\nabla_{X_s}\mathcal{E} = \nabla_{X_l}\mathcal{E} \cdot (1 + \nabla_{X_s}\sum_{t=s}^{l-1} F(X_t))$ where $\mathcal{E}$ is loss. In most cases, $\nabla_{X_s}\sum_{t=s}^{l-1} F(X_t)$ is much smaller than 1, especially
for along more training iterations, which means that SC propagates the main gradient flow and thus information flow.

2) Our theory could be extended to other losses, e.g. cross entropy. Here we choose square error loss because of its much
simpler gradient computation compared with cross entropy. But with more extra efforts, we can follow our framework
to establish similar results. Indeed, this is also one reason why recent works [18-21] on network convergence analysis
focus on square error loss, as different losses reveal similar results but square error loss gives simpler derivation.

3) Appendix A.1 investigates the effects of $\lambda_1 \sim \lambda_3$ to the performance of PR-DARTS. The results show the stable
performance of PR-DARTS on CIAFR10 when tuning $\lambda_1 \sim \lambda_3$ in relatively large ranges, e.g. $\lambda_1 \in [10^{-2}, 1]$, $\lambda_2 \in$
$[10^{-4.5}, 10^{-2.5}]$ and $\lambda_3 \in [10^{-4}, 10^{-1.5}]$. This is mentioned in line 341. So one can choose $\lambda_1 \sim \lambda_3$ from the above ranges.

4) We divide operations in DARTS into skip-connection group and non-skip-connection group, and penalty their average
active probabilities/weights. The error on CIFAR10 is $2.69\%$ which is slightly better than $2.76\%$ of vanilla DARTS but is
worse than $2.58\%$ of ours without path-depth-wise regularizer, showing the importance of independent stochastic gates.

[Meta-Review · NeurIPS 2020]

The reviewers are overwhelmly positive on the paper. The paper provides a theoretical analysis on why DARTS tends to bias towards architectures with skip connections and then leverage the theoretical insight to introduce a regularization term to reduce the bias. The paper is well written and has valuable contribution on both theoretical analysis and practical algorithm.